# ItDPDM: Information-Theoretic Discrete Poisson Diffusion Model

**Sagnik Bhattacharya**[1]*, **Abhiram R. Gorle**[1]*, **Ahsan Bilal**[2], **Connor Ding**[1],
**Amit Kumar Singh Yadav**[3], **Tsachy Weissman**[1]
[1]Department of Electrical Engineering, Stanford University
[2]Department of Computer Science, Oklahoma University
[3]School of Electrical and Computer Engineering, Purdue University, West Lafayette, IN, USA

## Abstract

Generative modeling of non-negative, discrete data, such as symbolic music, remains challenging due to two persistent limitations in existing methods. First, most approaches rely on modeling continuous embeddings, which are not well-suited for inherently discrete data distributions. Second, they typically optimize variational lower bounds instead of the true data likelihood, leading to inaccurate likelihood estimates and degraded sampling quality. While recent diffusion-based models have addressed these issues individually, we tackle them jointly. In this work, we introduce the **Information-Theoretic Discrete Poisson Diffusion Model (ItDPDM)**, inspired by photon arrival processes, unifying exact likelihood estimation with discrete-state generative modeling. Central to our approach is an information-theoretic Poisson Reconstruction Loss (PRL) that admits a provable, exact relationship with the true data likelihood. ItDPDM achieves improved likelihood and sampling performance over prior discrete and continuous diffusion models on a variety of synthetic discrete datasets. Furthermore, on real-world datasets such as symbolic music and images, ItDPDM attains superior likelihood estimates and competitive generation quality, demonstrating a proof of concept for principled, distribution-robust discrete generative modeling.

## 1 Introduction and Background

Denoising diffusion models have advanced generative modeling, outperforming GANs in image synthesis [1] and autoregressive models in likelihood-based tasks [2]. Their flexibility enables broad industrial use—from open-ended text-to-image generation [3–5], to audio [6] and medical imaging [7]. Diffusion has also been extended to multimodal and structured tasks, including video synthesis [8], cross-modal retrieval [9], and molecular modeling [10, 11].

**Limitations of Existing Works:** Diffusion models can be classified by timestep type: discrete (DT) or continuous (CT) and latent space: discrete (DS) or continuous (CS), forming four classes: DTDS, DTCS, CTDS, and CTCS, as shown in Figure 1. DTCS (e.g., VDM [2]) and CTCS (e.g., IT-Gaussian diffusion [12]) are effective in continuous domains [2, 13], but suboptimal for inherently discrete non-Gaussian data distributions. As shown in Figure 1, the continuous-state models map discrete data to continuous state space via z-scoring [14], tail normalization [15], or uniform dequantization [12]. However, these fail to close the discretization gap (e.g., $\frac{1}{127.5}$ for images), and lead to learning suboptimal probability density functions (pdf) instead of probability mass functions (pmf) [12]. Figure 3 shows how continuous DDPMs miss the second mode in the evidently bimodal NYC Taxi distribution [16]. Moreover, discretizing outputs during post-processing introduces train-test mismatch [12, 17, 18]. Recent discrete-state models directly operate in the discrete domain, addressing these limitations by avoiding embedding into continuous spaces altogether.

---

*Equal contribution; Correspondence to `sagnikb@stanford.edu`. Our implementation is available here.

39th Conference on Neural Information Processing Systems (NeurIPS 2025).

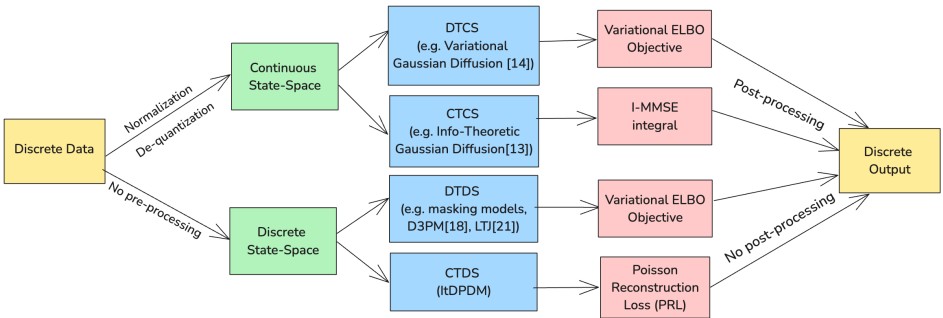

Figure 1: Classification of diffusion models based on latent state-space (DS/CS) and timesteps (DT/CT), resulting in 4 combinations - DTCS, CTCS, DTDS, and CTDS

Discrete-time discrete-state (DTDS) models [15, 17, 19] operate natively in the discrete domain and outperform variational Gaussian-based methods, but often ignore ordinal structure of integer-valued data and need post-processing. Learning-to-Jump (LTJ) [20], a recent DTDS method using binomial thinning and a variational objective, improves generation on non-negative, skewed data. However, LTJ has two drawbacks: (1) its evidence lower bound (ELBO)-based training uses a variational relative entropy loss, which lacks an exact relation to the true data likelihood, yielding suboptimal likelihood and degraded generation quality; (2) denoising requires careful calibration of $T$ (e.g., 1000), the number of discrete denoising timesteps, without any flexibility to skip or subsample.

**Main Contributions.** To address these limitations, we introduce a novel information-theoretic **Discrete Poisson Diffusion Model (ItDPDM)**. As shown in Figure 1, contrary to Gaussian diffusion, ItDPDM directly models discrete non-negative data using a Poisson process, avoiding the necessity for any soft discretization or dequantization. Contrary to variational DTDS models like LTJ [20], ItDPDM provides improved, closed-form likelihood estimates while maintaining competitive generation quality. Our main contributions are summarized as follows:

- We propose **ItDPDM**, a novel generative framework based on a Poisson diffusion process for modeling non-negative discrete data. Unlike prior approaches relying on variational ELBO objectives, ItDPDM enables a likelihood-consistent objective with tractable 1-D quadrature, bypassing the limitations of variational inference.

- We introduce the information-theoretic *Poisson Reconstruction Loss (PRL)*, a Bregman divergence [21] tailored to Poisson processes and establish its exact relation to negative log-likelihood (NLL) via the **I-MPRL identity** in Eq. (17), enabling non-variational optimization of discrete probability mass functions (PMFs).

- Experiments on synthetic datasets with varied data distributions show that ItDPDM outperforms earlier baselines in **Wasserstein-1** distance and log-likelihood (NLL). ItDPDM's discrete Poisson-based diffusion generalizes well beyond Poisson distributed data.

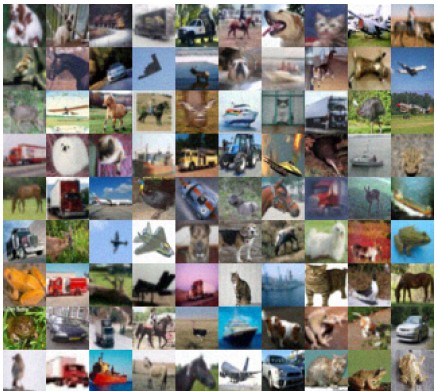

Figure 2: Unconditional image samples generated by ItDPDM

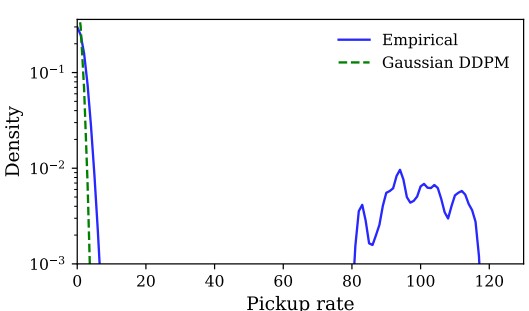

Figure 3: Gaussian diffusion fails to accurately learn the discrete probability density

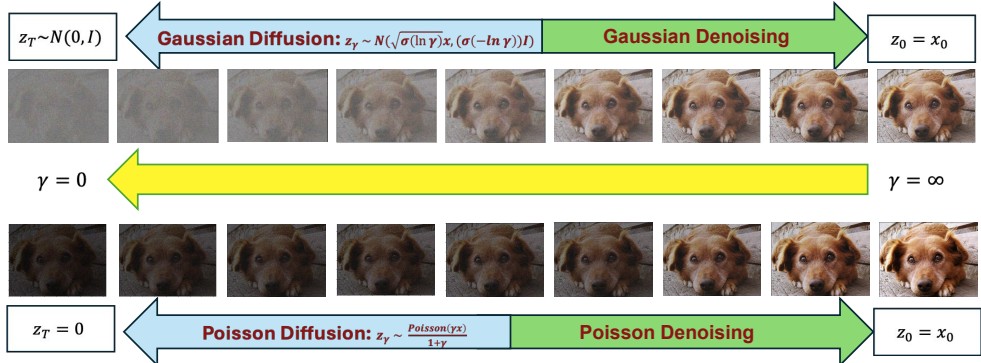

Figure 4: Comparison of Gaussian (top) and Poisson diffusion processes (bottom).

- We also provide closed-form upper bounds on the negative log-likelihood (NLL) and an importance-sampling estimator for efficient training, ensuring scalability to high-dimensional settings. Empirically, ItDPDM achieves significantly lower NLL estimates on CIFAR-10 and Lakh MIDI datasets while maintaining competitive generation quality.

This work presents a proof-of-concept for information-theoretic discrete Poisson diffusion models, showing initial gains over baselines in modeling discrete, positive-valued data. It serves as a first step toward principled diffusion modeling in discrete domains, not a state-of-the-art solution.

## 2 Information-Theoretic Diffusion

We briefly revisit the Information-Theoretic Gaussian Diffusion (ITDiff) framework from [12], which helps us draw parallels to ItDPDM in Sec 3. The Gaussian noise channel is defined as

$$z_\gamma = \sqrt{\gamma}x + \epsilon, \ \epsilon \sim \mathcal{N}(0, I),$$

with signal-to-noise ratio (SNR) parameter '$\gamma$' and data distribution $p(x)$.

**Relating Minimum Mean Square Error (MMSE) to Mutual Information**

The "I-MMSE" relation [22] links mutual information $I$ with minimum mean square error (MMSE):

$$\frac{d}{d\gamma}I(x; z_\gamma) = \frac{1}{2}\mathrm{mmse}(\gamma),\tag{1}$$

where the MMSE is defined as: $\mathrm{mmse}(\gamma) = \min_{\hat{x}(z_\gamma, \gamma)} \mathbb{E}_{p(z_\gamma, x)}\left[\|x - \hat{x}(z_\gamma, \gamma)\|_2^2\right]$. A pointwise generalization of Eq. (1) to the KL divergence is as follows:

$$\frac{d}{d\gamma}D_{KL}\big(p(z_\gamma|x) \parallel p(z_\gamma)\big) = \frac{1}{2}\mathrm{mmse}(x, \gamma)\tag{2}$$

From here, the following discrete probability estimator is derived as through an exact formulation of the variational lower bound (VLB) for diffusion models [2] :

$$-\log P(x) = \frac{1}{2}\int_0^\infty \mathrm{mmse}(x, \gamma)\, d\gamma\tag{3}$$

## 3 ItDPDM: Information-Theoretic Poisson Diffusion

**Poisson Noise Channel:** We define the canonical Poisson noise channel: given a non-negative input $x \geq 0$, the output $z_\gamma$ is drawn from $\mathcal{P}(\gamma x)$, where $\gamma$ denotes the SNR. The conditional PMF is

$$P(z_\gamma|x) = \frac{(\gamma x)^{z_\gamma} e^{-\gamma x}}{z_\gamma!}, \ z_\gamma \in \mathbb{N}_0,\tag{4}$$

where $\mathcal{P}(\cdot)$ denotes the Poisson distribution. This setup is motivated by Poisson channels arising in direct-detection optical systems [23, 24], where photon counts follow a Poisson process with rate determined by a combination of signal intensity and device-induced dark current [25].

**Diffusion with Poisson Noise:** We propose an information-theoretic Poisson diffusion process, where a source $x \sim p(x)$ is corrupted at SNR $\gamma$ via $z_\gamma \sim \mathcal{P}(\gamma x)$, producing discrete, non-negative integers at each step. Unlike Gaussian noise, Poisson corruption is non-additive and not source-separable, making denoising more challenging. Figure 4 contrasts Gaussian and Poisson diffusion: Gaussian begins from white noise, whereas Poisson diffusion starts from a black image with zero photons.

**Poisson Reconstruction Loss (PRL):** The function $l_0(x) = x \log x - x + 1$, $x > 0$ (where $\log$ denotes the natural logarithm) is the convex conjugate of the Poisson distribution's log moment generating function (proof in App. D.1) and often arises naturally in the analysis of continuous and discrete-time jump Markov processes [20, 26] and for mutual information estimation in the Poisson channel [27]. Building on this, we define the **poisson reconstruction loss** $l(x, \hat{x})$ as:

$$l(x, \hat{x}) = \hat{x} l_0(x/\hat{x}) = x \log\left(\frac{x}{\hat{x}}\right) - x + \hat{x}, \tag{5}$$

Analogous to the MMSE, we also define the minimum poisson reconstruction loss (MPRL) as:

$$\mathrm{mprl}(\gamma) \equiv \min_{\hat{\boldsymbol{x}}(\boldsymbol{z}_\gamma, \gamma)} E_{P(\boldsymbol{z}_\gamma, \boldsymbol{x})}\left[l(\boldsymbol{x}, \hat{\boldsymbol{x}}(\boldsymbol{z}_\gamma, \gamma))\right], \tag{6}$$

where $\hat{\boldsymbol{x}}(\boldsymbol{z}_\gamma, \gamma)$ denotes the denoiser. The optimal denoiser $\hat{\boldsymbol{x}}^*$ is the conditional expectation $E[X|Z_\gamma]$ using the fact that the Poisson reconstruction loss is a Bregman divergence [28] (proof in App.D.1).

$$\hat{\boldsymbol{x}}^*(z_\gamma, \gamma) \equiv \arg\min \mathrm{mprl}(\gamma) = E_{\boldsymbol{x} \sim P(\boldsymbol{x}|z_\gamma)}[\boldsymbol{x}] \tag{7}$$

The analytical solution is typically intractable due to the need to sample from the Poisson noise channel's posterior. We next highlight key properties [29] of this loss, showing it is a natural fit for evaluating reconstruction of non-negative data, analogous to squared error in the Gaussian case.

**Lemma 1** (Poisson Reconstruction Loss). *The loss function $l(x, \hat{x})$ satisfies the following properties:*

1. ***Non-negativity:*** *$l(x, \hat{x}) \geq 0$, with equality if and only if $x = \hat{x}$.*

2. ***Convexity:*** *$l(x, \hat{x})$ is convex in $\hat{x}$ for each fixed $x$, and in $x$ for each fixed $\hat{x}$.*

3. ***Scaling:*** *For any $\alpha > 0$, $l(\alpha x, \alpha \hat{x}) = \alpha l(x, \hat{x})$.*

4. ***Unboundedness for underestimation:*** *For any $x > 0$, $\lim_{\hat{x} \to 0^+} l(x, \hat{x}) = \infty$.*

5. ***Optimality of Conditional Expectation:*** *For any non-negative random variable $X$ with $E[X \log^+ X] < \infty$, the conditional expectation $E[X|Y]$ uniquely minimizes the expected loss $E[l(X, \hat{x})]$.*

Convexity makes the loss amenable to gradient-based methods. Property 4 penalizes underestimation, making $l(x, \hat{x})$ well-suited for non-negative data, unlike common loss functions (absolute/squared error). Figure 5 illustrates the behavior of the proposed PRL. As per Lemma 1, the conditional expectation $E[X|Y]$ uniquely minimizes the expected "mprl" loss function.

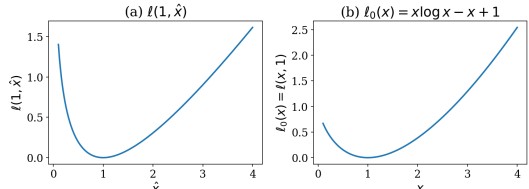

Figure 5: Poisson Reconstruction Loss (PRL): (a) vs. denoised pixel $\hat{x}$, for fixed ground truth pixel 1; (b) vs. ground truth pixel $x$, for fixed denoised output 1.

**Conditional Expectation for Poisson Channel** We define the angle bracket operator $X$ as conditional expectation given $Z_\gamma$: $\langle X \rangle = E[X|Z_\gamma]$ Unlike the linear Gaussian case, Poisson has a non-linear $\langle X \rangle$, making Poisson-based denoising fundamentally more complex. Nevertheless, it becomes linear under certain conditions, let $\langle X \rangle_z$ denote $\langle X \rangle$ evaluated at $Z_\gamma = z$ then:

**Lemma 2** (Linearity in Poisson Channel). *Let $Z_\gamma = \mathcal{P}(\gamma X)$. Then, $\langle X \rangle_z = az + b$, if and only if $X \sim Gam\left(\frac{1-\gamma a}{a}, \frac{b}{a}\right)$ for any $0 < a < \frac{1}{\gamma}$ and $b > 0$.*

Though Poisson analysis is complex, it simplifies here since the Gamma distribution is its conjugate prior [30], yielding a linear conditional variance (see App. D.2). This contrasts with the Gaussian case, where conditional variance is constant. We now revisit the squared error loss $\ell_{\mathrm{SE}}(x, \hat{x}) = (x - \hat{x})^2$, which satisfies for any finite-variance $X$:

$$E[\ell_{\mathrm{SE}}(X, \hat{x})] = E[\ell_{\mathrm{SE}}(X, E[X])] + \ell_{\mathrm{SE}}(E[X], \hat{x}) \tag{8}$$

Our Poisson reconstruction loss (PRL) has a similar property stated below ([29]).

**Lemma 3.** *For any non-negative random variable $X$ with $E[X \log^+ X] < \infty$, and any $\hat{x} \in [0, \infty)$,*

$$E\left[l(X, \hat{x})\right] = E\left[l\left(X, E[X]\right)\right] + l\left(E[X], \hat{x}\right) \tag{9}$$

A result that immediately follows from Lemma (3), when combined with the non-negativity property in Lemma 1, is that $E[X]$ uniquely minimizes $E\left[l(X, \hat{x})\right]$ over all $\hat{x}$:

$$\min_{\hat{x}} E\left[l(X, \hat{x})\right] = E\left[l\left(X, E[X]\right)\right] = E[X \log X] - E[X] \log E[X]. \tag{10}$$

Interestingly, in Poisson channels, this estimator depends only on the marginal distribution of $Z_\gamma$, a property formalized by the *Turing-Good-Robbins* formula [31, 32]. This result closely relates to the Discrete Universal Denoiser (DUDE) [33], which estimates discrete signals from noisy observations.

**Lemma 4** (Optimal Estimator in Poisson Channel). *Let $Z_\gamma = \mathcal{P}(\gamma X)$. Then, for every $\gamma > 0$,*

$$\langle X \rangle_z = \frac{1}{\gamma} \frac{(z+1) P_{Z_\gamma}(z+1)}{P_{Z_\gamma}(z)}, \quad z = 0, 1, \dots \tag{11}$$

The PRL objective provides a principled objective for modeling non-negative discrete data, directly modeling PMFs and avoiding quantization artifacts inherent in squared error loss, which assumes continuous outputs. The pointwise denoising relation for the Poisson channel is: (proof in App.F)

$$\frac{d}{d\gamma} D_{KL}\left[P(z_\gamma|x) \parallel P(z_\gamma)\right] = \mathrm{mprl}(x, \gamma), \tag{12}$$

where $p(z_\gamma) = \int p(z_\gamma|x)p(x)dx$ is the marginal distribution, and **pointwise MPRL** is defined as:

$$\mathrm{mprl}(x, \gamma) \equiv E_{P(z_\gamma|x)}\left[l\left(x, \hat{x}^*(z_\gamma, \gamma)\right)\right] \tag{13}$$

The **pointwise MPRL** is the MPRL evaluated at a fixed $x$, and its expectation over $p(x)$ recovers the total MPRL. Taking expectation wrt $x$ in Eq. (12) recovers the I-MPRL relation in Eq. (14). Moreover, for a mismatched denoiser [29], integrating the excess pointwise loss over $\gamma$ equals the KL divergence between the true and mismatched channel outputs (via Eq. (12)).

**I-MPRL identity:** Following the foundational result of [22], which relates the mutual information derivative to MMSE in Gaussian channels, [12] leverages this identity in generative modeling. Analogously, we establish the I-MPRL identity for the Poisson channel, as follows:

$$\frac{d}{d\gamma} I(x; z_\gamma) = \mathrm{mprl}(\gamma). \tag{14}$$

A similar result holds for the derivative with respect to the *dark current* in a general Poisson channel. Using the **incremental channel** technique from [22], we derive both results in App. D.3. This enables exact relations between the proposed PRL objective and the likelihood, offering an *information-theoretic justification for Poisson diffusion*. Detailed proofs of Lemmas 1–4 are provided in App. D.2, along with Lemmas 5 and 6, stated and proved therein.

**Thermodynamic Integration for Variational Bound:** The pointwise denoiser yields a log-likelihood expression akin to the variational bound. Unlike traditional methods that rely on expensive sampling, diffusion models leverage the structure of the noise for efficient sampling at arbitrary noise levels [34]. Letting $P(z_\gamma|x) \sim \mathcal{P}(\gamma x)$, using thermodynamic integration method from [35, 36] yields:

$$\int_{\gamma_0}^{\gamma_1} \frac{d}{d\gamma} D_{KL}[P(z_\gamma|x) \parallel P(z_\gamma)] \, d\gamma = -\int_{\gamma_0}^{\gamma_1} \mathrm{mprl}(x, \gamma) \, d\gamma, \tag{15}$$

where $\mathrm{mprl}(x, \gamma) \equiv \mathbb{E}_{p(z_\gamma|x)}\left[l\left(x, \hat{x}^*(z_\gamma, \gamma)\right)\right]$ is the pointwise MPRL for Poisson denoising. The true (exact) log-likelihood is given by:

$$-\log P(x) = \underbrace{D_{KL}[P(z_{\gamma_1}|x) \parallel P(z_{\gamma_1})]}_{\text{Prior loss}} + \underbrace{E_P[-\log P(x|z_{\gamma_0})]}_{\text{Reconstruction loss}} - \underbrace{\int_{\gamma_0}^{\gamma_1} \mathrm{mprl}(x, \gamma) \, d\gamma}_{\text{Diffusion loss}} \tag{16}$$

We also outline a possible extension of the proposed Continuous-Time Discrete-State Poisson Diffusion to a Continuous-Time Continuous-State equivalent of [12] in App. G.

**Discrete Probability Estimation via MPRL:** We derive a novel discrete probability estimator in the Poisson channel setting, where $x \sim P(x)$ and $Z_\gamma \sim \mathcal{P}(\gamma x)$. In the limits $\gamma_0 \to \infty$ and $\gamma_1 \to 0$, both the prior and reconstruction loss vanish, which yields the following tractable expression:

$$D_{KL}[P(z_{\gamma_1}|x)||P(z_{\gamma_1})] + E_{P(z_{\gamma_0}|x)}[-\log P(x|z_{\gamma_0})] = 0.$$

and, therefore Eq. 16 yields an *exact likelihood relation*:

$$-\log P(x) = \int_0^\infty \mathrm{mprl}(x,\gamma)\,d\gamma \implies \boxed{E[-\log P(x)] = \int_0^\infty \mathrm{mprl}(\gamma)\,d\gamma} \tag{17}$$

In practice, we use importance-sampling quadrature for this, yielding an unbiased estimate. Conventional diffusion incurs two levels of approximation: a) the ELBO (surrogate) loss replaces the true log-likelihood, and b) Monte-Carlo quadrature (or the integral). Our formulation eliminates the first level to work with the true data likelihood.

To obtain an expression resembling the variational bound, taking an expectation in $x \sim P(x)$ gives:

$$E[-\log P(x)] = E\left[\int_0^\infty \mathrm{mprl}(x,\gamma)\,d\gamma\right] = \int_0^\infty E[\mathrm{mprl}(x,\gamma)]d\gamma = \int_0^\infty E[l(X,\hat{X}^*)]\,d\gamma$$

$$= \int_0^\infty E\left[X\log X - E[X|Z_\gamma]\log E[X|Z_\gamma]\right]d\gamma = \int_0^\infty E\left[X\log \frac{X}{E[X|Z_\gamma]}\right]d\gamma,$$

here, $\hat{X}^*(X,\gamma) = E[X|Z_\gamma]$ denotes the optimal estimator. This section establishes a tractable, non-variational estimator for discrete distributions in the Poisson channel by connecting the MPRL objective to the true data likelihood. We also present in App. F.1 an equivalent score matching formulation using a Poisson-adapted version of *Tweedie's formula* denoising. Additionally, App. H provides a comprehensive comparison between ItDPDM (ours) and LTJ [20].

**Extension to Multivariate settings:** While the current framework considers a univariate setting, this can naturally be extended to the vector Poisson channel under mild regularity. Let $X, \lambda \in \mathbb{R}_+^d$, $\gamma \in \mathbb{R}_+^{d \times d}$, and $Z|X \sim \Pi_{i=1}^d \mathrm{Pois}((\gamma X)_i + \lambda_i)$. In this case, the I-MPRL identities (Eq. 14) hold component-wise [37]. Consequently, all results in this work effectively carry over to the multivariate setting by replacing scalars with vectors and sums with inner products (or traces). For empirical validation, we also provide toy 2D experiments in App. C.5.

## 4 Numerical Details

**MPRL Upper Bound:** A key challenge is the inaccessibility of the posterior distribution in Eq. (7). To bound the intractable marginal likelihood, we compare our (suboptimal) neural denoiser $\hat{X}(Z_\gamma, \gamma)$ with the intractable optimal conditional expectation $\hat{X}^*$. This reformulates the expected loss entirely in terms of $\hat{X}(Z_\gamma, \gamma)$ from the Poisson diffusion model: (proof in App. A.1)

$$E[-\log P(x)] = \int_0^\infty \mathrm{mprl}(\gamma)\,d\gamma \leq \int_0^\infty \mathbb{E}[\ell(X,\hat{X})]\,d\gamma \tag{18}$$

It is important to note that the likelihood (NLL) upper bound in Eq. (18) is empirical, capturing the suboptimality of the learned neural denoiser. Eq. (17) yields an exact theoretical expression for the likelihood (NLL), unlike variational diffusion models, which introduce two layers of approximation: first via the ELBO, and then through an upper bound on the denoiser.

**Parametrization:** To ensure stability across SNR levels, we reparameterize the Poisson observation $Z_\gamma \sim \mathcal{P}(\gamma X)$ to mitigate mean and variance explosion. Instead of feeding $Z_\gamma$ directly into the neural network, we define the normalized $\tilde{Z}_\gamma = Z_\gamma/(1+\gamma)$, keeping it within $[0, X]$ with high probability. This transformation preserves interpretability: at high SNR, $E[\tilde{Z}_\gamma] \approx X$, while at low SNR ($\gamma \to 0$), it tends to zero, aligning with Poisson behavior. We input $(\tilde{Z}_\gamma, \gamma)$ into the network in place of $(Z_\gamma, \gamma)$. Adopting the log-SNR parameterization $\alpha = \log \gamma$, we get:

$$E[-\log P(x)] = \int_{-\infty}^\infty e^\alpha \mathrm{mprl}(\alpha)\,d\alpha \leq \int_{-\infty}^\infty e^\alpha E[l(X,\hat{X})]\,d\alpha. \tag{19}$$

For details on efficient numerical integration of this expression, see App. A.2.

**MPRL Tail Bounds:** Since the integration on the RHS of Eq.(19) is intractable, we identify a finite integration range $(\alpha_0, \alpha_1)$ beyond which the contribution becomes negligible. The RHS of Eq. (19) can thus be written in terms of '$\alpha$' as:

$$= \int_{\alpha_0}^{\alpha_1} e^\alpha \mathrm{mprl}(\alpha)d\alpha + \left(\int_{-\infty}^{\alpha_0} + \int_{\alpha_1}^\infty\right) e^\alpha \mathrm{mprl}(\alpha)d\alpha \leq \int_{\alpha_0}^{\alpha_1} e^\alpha E[l(X,\hat{X})]\,d\alpha + f(\alpha_0,\alpha_1)$$

**Algorithm 1** ItDPDM Training

**Require:** Dataset $\{x_i\}_{i=1}^N$, # log-SNR samples $S$, SNR range $[\gamma_{\min}, \gamma_{\max}]$, denoiser $f_\theta$
1: **for** $s = 1, \ldots, S$ **do**
2:     Sample mini-batch $B$ from $\{x_i\}$
3:     Sample $\alpha \sim$ Logistic, $\gamma \leftarrow \exp(\alpha)$
4:     Sample $z_\gamma \sim \frac{\text{Poisson}(\gamma\, x_B)}{1+\gamma}$
5:     $\hat{x}_B \leftarrow f_\theta\big(\text{data\_transform}(z_\gamma), \gamma\big)$
6:     $\ell \leftarrow \sum_{i \in B} \text{PRL}(x_i, \hat{x}_i)$, $L \leftarrow \ell \,/\, q(\alpha)$
7:     Update $\theta$ by gradient descent on $L$
8: **end for**
9: **return** $\theta$

**Algorithm 2** ItDPDM Sampling

**Require:** Trained model $f_\theta$, # reverse steps $T$
1: Compute $\{\gamma_t\}$ (e.g. spaced in log-SNR)
2: Initialize $z_{\gamma_T} \leftarrow \mathbf{0}$
3: **for** $t = T, T-1, \ldots, 1$ **do**
4:     $\hat{x}_0 \leftarrow f_\theta\big(\text{data\_transform}(z_{\gamma_t}), \gamma_t\big)$
5:     Sample $z_{\gamma_{t-1}} \sim \text{Poisson}\big(\gamma_{t-1}\, \hat{x}_0\big)$
6: **end for**
7: **return** $\hat{x}_0$

We analytically derive upper bounds for the left and right tail integrals, denoted by $f(\alpha_0, \alpha_1)$ above in App. E, and show that their contributions decay rapidly outside the relevant integration range. Algorithm 1 and 2 represent the pseudocode used for ItDPDM training and generation respectively.

**Gap between the exact and Monte Carlo (MC) estimators:** Consider the canonical binary-input Gaussian channel $Y = \sqrt{\gamma}X + N$, where $X \in \{\pm 1\}$ with equal probability and $N \sim \mathcal{N}(0, 1)$. The I-MMSE identity gives (although intractable in most practical cases)

$$\frac{d}{d\gamma} I(X;Y) = \tfrac{1}{2}\text{mmse}(\gamma), \text{ so that } I(X;Y) = \frac{1}{2}\int_0^\infty \text{mmse}(\gamma)\, d\gamma = \ln 2.$$

For this binary setting, the MMSE admits a closed form,

$$\text{mmse}(\gamma) = 1 - \mathbb{E}\big[\tanh^2(\sqrt{\gamma}Y)\big],$$

since the posterior mean is $\mathbb{E}[X|Y] = \tanh(\sqrt{\gamma}Y)$. When approximated using an $n$-sample Monte Carlo (MC) estimator, the resulting error decays at the rate $\mathcal{O}(1/\sqrt{n})$, consistent with the central limit theorem (CLT). In practice, this corresponds to an error of order $10^{-3}$ for $n > 1000$. A similar behavior holds for MPRL estimators: for instance, when $X \sim \text{Gamma}(2, 3)$, its MC estimator exhibits the same $\mathcal{O}(1/\sqrt{n})$ convergence rate, with an error magnitude of order $10^{-2}$ for $n > 1000$. These results help illustrate that empirical estimators closely track theoretical values even for finite, moderate sample sizes, motivating our design choice of using an importance-sampling quadrature.

## 5 Experiments

We begin by evaluating on synthetic datasets exhibiting sparsity, skewness, and overdispersion: settings where Gaussian diffusion models often underperform, along with extreme distributions like Zipf where **LTJ** [20] underperforms. These experiments help empirically validate ItDPDM by i) recovering ground-truth likelihood (NLL) and ii) improving modeling of discrete, non-negative data. We then evaluate ItDPDM on real-world domains like **CIFAR10** (images) and **Lakh MIDI** (symbolic music), where discrete structure is inherent. ItDPDM consistently achieves superior likelihood estimates and competitive generation quality, as evidenced by domain-specific metrics.

### 5.1 Synthetic Data

We consider various synthetic distributions containing univariate non-negative data $x$ grouped into two broad categories: discrete $x \in \mathbb{N}$, and continuous $x \in [0, \infty]$ to mimic distributions exhibiting either sparse, heavy-tailed, skewed, zero-inflated or overdispersed behaviour.

**Discrete counts ($x \in \mathbb{N}$):** We generate six synthetic distributions capturing key real-world behaviors: PoissMix (airport arrivals), ZIP, NBinomMix (forum activity), BNB, and two heavy-tailed laws: Zipf and Yule-Simon (word frequencies). These cover bimodality, overdispersion, and long tails. Distribution parameters are listed in Table 5 in Appendix, with design details in App. C.1.

**Continuous non-negative ($x \in [0, \infty)$):** We also include six skewed continuous densities—Gamma, Log-Normal, Lomax, Half-Cauchy, Half-t, and Weibull—described in App. C.4.

Table 1: Metrics for synthetic datasets (↓ lower is better). Bold indicates best.

| | WD | | | NLL$^2$ | | | |
| --- | --- | --- | --- | --- | --- | --- | --- |
| **Distribution** | **DDPM** | **ItDPDM** | **LTJ** | **True NLL** | **DDPM** | **ItDPDM** | **LTJ** |
| PoissMix | $3.76 \pm 0.32$ | $\mathbf{0.99 \pm 0.15}$ | $1.21 \pm 0.30$ | 3.80 | 4.24 | **3.72** | 3.69 |
| ZIP | $2.31 \pm 0.66$ | $\mathbf{0.56 \pm 0.43}$ | $0.69 \pm 0.24$ | 2.13 | 1.67 | **2.22** | 2.30 |
| NBinomMix | $4.89 \pm 0.59$ | $1.39 \pm 0.37$ | $\mathbf{1.15 \pm 0.41}$ | 0.87 | 1.84 | 1.43 | **1.30** |
| BNB | $1.89 \pm 0.45$ | $0.67 \pm 0.23$ | $\mathbf{0.65 \pm 0.32}$ | 2.06 | 2.56 | 1.87 | **2.01** |
| Zipf | $1.51 \pm 0.53$ | $\mathbf{0.48 \pm 0.13}$ | $0.73 \pm 0.25$ | 1.57 | 1.34 | **1.70** | 1.77 |
| YS | $0.32 \pm 0.12$ | $\mathbf{0.14 \pm 0.03}$ | $0.17 \pm 0.06$ | 0.94 | 1.39 | **0.79** | 0.76 |

**Model Architecture:** The neural denoiser model for all (discrete, continuous) cases uses a similar architecture (`ConditionalMLP`) to ensure fair comparison: a 3-layer MLP with 64 hidden units, LayerNorm, Leaky-ReLU activations (slope = 0.2). Further training details can be found in App. C.2. To maintain computational tractability, most distributions are truncated at 50. For each distribution, we draw 50,000 i.i.d. samples to form the training data and generate 50,000 samples for each run.

**Metrics and results:** We report Wasserstein-1 distance (WD) and negative log-likelihood (NLL) between empirical distributions of generated and test samples (see App. C.2). Table 1 summarizes these metrics for **ItDPDM** and all baselines. To illustrate the quality of PMF modeling, Figure 6 overlays the true and generated PMFs across all discrete datasets. As shown, **ItDPDM** consistently outperforms DDPM (trained with MSE) across all datasets, achieving lower WD and NLL estimates that closely align with the true values. It further outperforms **LTJ** in 4 out of 6 datasets, demonstrating strong generalization of ItDPDM across diverse distributions, beyond just Poisson-mixture datasets. In contrast, LTJ performs well primarily on binomial-related datasets, which are well-suited to its variational count-thickening loss. More details on PMF estimation are in App. C.3.

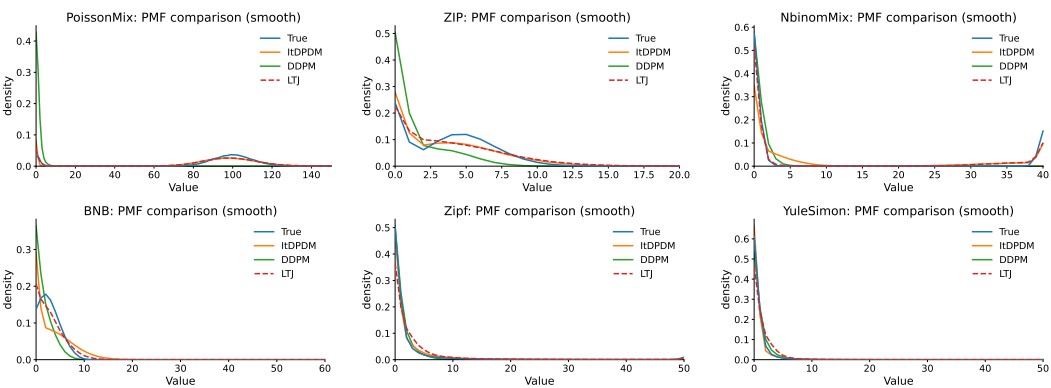

Figure 6: Comparison of true and generated probability distributions

## 5.2 Real-World Data

We evaluate ItDPDM on two discrete datasets: CIFAR-10 images and Lakh MIDI (LMD) symbolic music and compare against existing baselines: Improved DDPM (IDDPM) [38], information-theoretic Gaussian diffusion (ITDiff) [12], discrete masking-based (D3PM) [17], and learning-to-jump (LTJ) [20]. CIFAR-10 comprises 60,000 color images ($32 \times 32$) across 10 classes [39]. LMD contains 648,574 symbolic music sequences of 1024 integers: 0 (rest), 1 (continuation), and $2-89$ representing note pitches [40]. Unlike [12], which fine-tunes pre-trained models, the absence of pre-trained models in our setting necessitates training from scratch. Denoiser architectures (U-Net [41], ConvTransformer [17], DenseDDPM [42]) are discussed in App. B.3.

## 5.3 Performance Comparison: Negative Log Likelihood (NLL)

Two architectural variants from **DDPM** [13] and **IDDPM** [38] are used. Table 2 reports test-set NLLs on a CIFAR-10 subset, comparing **ItDPDM** to relevant baselines: (1) ITDiff [12], which fine-tunes pretrained Gaussian DDPM/IDDPM models, and (2) Gaussian + MSE, where DDPM/IDDPM models are trained from scratch using the ITDiff objective, ensuring a fair comparison. ItDPDM (Poisson + PRL) consistently achieves the lowest NLL across both backbones, with IDDPM slightly outperforming DDPM. These results underscore the effectiveness of Poisson diffusion and PRL for modeling discrete, non-negative data without requiring dequantization. Figure 7 shows denoising

| Noising + Objective | DDPM | IDDPM |
|---|---|---|
| ITDiff[a] | 2.97 | 0.86 |
| Gaussian + MSE | 0.44 | 0.48 |
| Gaussian + PRL | **0.27** | **0.32** |
| Poisson + MSE | 0.23 | 0.22 |
| **ItDPDM**: Poisson + PRL | **0.18** | **0.17** |

| Noising + Objective | NLL (total data) |
|---|---|
| Gaussian + MSE | 0.51 |
| **ItDPDM**: Poisson + PRL | $\mathbf{4.61 \times 10^{-5}}$ |
| **Noising + Objective** | **NLL (without rests)** |
| Gaussian + MSE | 1.41 |
| **ItDPDM**: Poisson + PRL | **0.23** |

[a]checkpoint models provided by [12] directly used

Table 2: (a) (Left) CIFAR10 (image) test-set NLL; (b) (Right) LMD (music) test-set NLL.

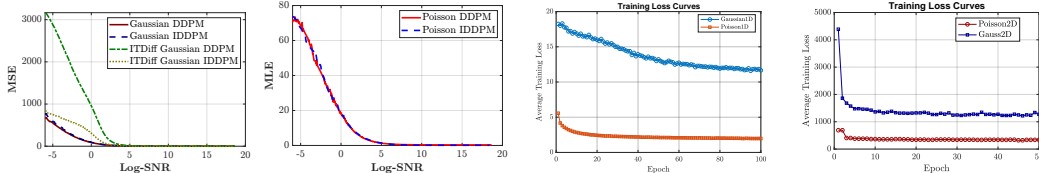

Figure 7: (a) Test MSE vs. logSNR for Gaussian diffusion; (b) Test PRL vs. logSNR for ItDPDM; (c) Training loss under Gaussian noise vs. PRL for 1D music; (d) Training loss under Gaussian noise vs. PRL for 2D images

loss curves across SNRs: MSE for ITDiff and Gaussian + MSE (Figure 7a), and PRL for ItDPDM (Figure 7b). PRL remains lower at low SNRs, consistent with the NLL improvements observed in Table 2. Similar trends are seen on symbolic music (Table 2b), where ItDPDM achieves even larger NLL reductions, further demonstrating its suitability for discrete generative modeling.

## 5.4 Performance Comparison: Generation Quality

Next, for evaluating generation quality of the generated images and music, we use domain-specific metrics: Structural Similarity Index Measure [43], and Fréchet Inception distance (FID) [44] for generated images; Fréchet Audio distance (FAD) [45], Consistency (C) [46], Mel-Spectrogram Inception Distance (MSID) [47] and Wasserstein Distance (WD) [48] for generated music. As shown in Figure 2, ItDPDM can generate realistic-looking natural images. Due to the limited computational budget available for training, the raw metrics for all models are lower than their reported values in IDDPM [38] and LTJ [20]. The relative performance of the models gives us the necessary insights: for image generation, IDDPM[38] achieves the best FID, with ItDPDM ranking second. In symbolic music, D3PM with categorical masking obtains the lowest FAD. ItDPDM outperforms LTJ for both image and symbolic music cases, by virtue of our exact likelihood estimation, as opposed to LTJ's variational relative entropy loss. Further details along with generated piano rolls are in App. B.

Table 3: Domain-specific generative quality metrics. Image: FID, SSIM; Audio: FAD, C, MSID, WD. FID values indicate dB increase (worse) from DDPM[13] baseline

| Baseline | Image | | Audio | | | |
|---|---|---|---|---|---|---|
| | FID (dB) | SSIM | FAD | C | MSID | WD |
| DDPM [13] | **0** | **0.93** | 0.89 | 0.91 | 0.82 | 2.83 |
| LTJ [20] | 0.30 | 0.90 | 0.66 | 0.92 | 0.71 | 2.23 |
| D3PM [17] | 2.93 | 0.86 | **0.61** | **0.98** | **0.59** | **1.99** |
| ItDPDM | 0.18 | 0.91 | 0.64 | 0.94 | 0.67 | 2.14 |

## 5.5 Cross Training Paradigm

To isolate the benefits of Poisson diffusion and PRL objective, we perform cross-training: Gaussian + PRL and Poisson + MSE. As shown in Table 2(a), ItDPDM (Poisson + PRL) yields the best NLL, confirming PRL's suitability for Poisson diffusion. Notably, Gaussian + PRL also outperforms Gaussian + MSE, suggesting PRL's broader effectiveness on discrete data. Moreover, ItDPDM converges faster and reaches lower loss than its Gaussian counterpart, as shown in ( Figure 7c–d). We further validate the I–MPRL identity (Eq. 14) by comparing area-under-loss curves and final losses, finding close numerical similarities between the Poisson and Gaussian models, and aligning with the theoretical formulation.

# 6 Limitations and Future Work

Despite its theoretical strengths, performance benefits on diverse discrete distributions, and competitive empirical results, **ItDPDM** remains a proof of concept and does not yet match state-of-the-art performance in real-world generative tasks. As discussed in 5.4, these performance gaps are partly due to limited training and architectural tuning, with details provided in App. B. Additionally, logistic sampling parameters are fixed a priori without extensive hyperparameter tuning. We believe that longer training schedules (3000+ epochs), systematic hyperparameter sweeps (e.g., number of log-SNR steps), and targeted ablations could substantially improve ItDPDM's performance. Following prior (related) work [12, 13, 17, 49], we also limit evaluation to unconditional generation on the training distribution; exploring conditional generation and robustness under prior misspecification or OOD generation remains a promising direction for future work.

# 7 Related Work

Diffusion models are widely used in generative and restoration tasks [38, 50], grounded in denoising autoencoders [51], variational inference [2], and score-based SDEs [49]. Recent works add information-theoretic insights [12], linking mutual information and MMSE [22] to likelihood bounds. Non-Gaussian extensions via annealed score matching [52, 53] and score-based SDEs [54] enhance theoretical rigor. In the discrete setting, **Blackout Diffusion**[55] and **Beta Diffusion**[56] use irreversible priors without tractable likelihoods. **SEDD**[57] uses score-entropy loss for token-level modeling but inherits ELBO-based approximations and lacks exact likelihood. **LTJ**[20] employs binomial thinning but is non-reversible and discrete-time. Our method overcomes these by using a reversible Poisson process, enabling bidirectional corruption, exact likelihood, and efficient continuous-time sampling. A more detailed discussion is provided in App. K.

# 8 Conclusion

We introduce **ItDPDM**, a diffusion framework for non-negative discrete data that combines a Poisson noising process with a principled Poisson Reconstruction Loss (PRL), enabling exact likelihood estimation and discrete sampling without dequantization. ItDPDM achieves lower NLL on both synthetic and real data, enhances modeling quality on varied synthetic distributions, and delivers competitive results in image and symbolic music generation. Though a proof-of-concept, ItDPDM lays a strong foundation for distribution-robust discrete generative modeling, with applications in symbolic music, low-light imaging, and other count-based domains.

# 9 Broader Impact

**ItDPDM** provides a principled framework for modeling a wide range of discrete, non-negative data distributions, including symbolic music, images, and skewed count data, without assuming any Gaussian or Binomial-related structure. Its grounding in a reversible Poisson process enables tractable likelihood estimation and efficient sampling, offering a robust alternative for domains with inherently discrete structure. This work may inspire future applications in low-light image reconstruction, scientific count data modeling, and generative modeling in constrained-data regimes.

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

# Appendix

## A Numerical Details

### A.1 MPRL Upper Bound

This part delves into the derivation of Eq. (18). We first express the expected loss in terms of the optimal estimator (using shorthand notation subsequently):

$$E[l(X, \hat{X}(X, \gamma))] = E\left[X \log \frac{X}{\hat{X}(X, \gamma)} - X + \hat{X}(X, \gamma)\right] = E\left[X \log \frac{X}{\hat{X}^*}\right] + E\left[X \log \frac{\hat{X}^*}{\hat{X}} - X + \hat{X}\right].$$
(20)

Using the law of iterated expectation gives:

$$E[l(X, \hat{X})] = \mathrm{mprl}(\gamma) + E[l(\hat{X}^*, \hat{X})].$$

The second term above denotes the estimation gap, and rearranging the terms, we get:

$$E[-\log P(x)] = \int_0^\infty \left(E[l(X, \hat{X})] - E[l(\hat{X}^*, \hat{X})]\right) d\gamma.$$

Using Jensen's inequality here (based on the properties mentioned in Lemma 1), we have:

$$-E[l(\hat{X}^*, \hat{X})] \leq -E[\hat{X}^*] \log \frac{E[\hat{X}^*]}{\hat{X}} - E[\hat{X}^*] + \hat{X} = -l(E[X], \hat{X}).$$
(21)

Now, using the relation from Lemma 2 gives us:

$$\int_0^\infty \mathrm{mprl}(\gamma) \, d\gamma \leq \int_0^\infty E[l(X, E[X])] \, d\gamma.$$
(22)

We obtain a more elegant bound in terms of our suboptimal neural denoiser by dropping the negative term:

$$E[-\log P(x)] = \int_0^\infty \mathrm{mprl}(\gamma) \, d\gamma \leq \int_0^\infty E[l(X, \hat{X})] \, d\gamma.$$
(23)

### A.2 Numerical Integration.

This section outlines the effective computation of integral from (19). We first use importance sampling to rewrite the integral as an expectation over a distribution, $q(\gamma)$, allowing for unbiased Monte Carlo estimation. This leads to our final numerical approximation of the loss function $E_{p(x)}\left[-\log p(x)\right] \leq \mathcal{L}$, where

$$\mathcal{L} \equiv E_{q(\alpha)}\left[\frac{1}{q(\alpha)} E_{(x, z_\gamma)}[l(X, \hat{x})]\right].$$

We propose two paradigms for numerical integration: **Logistic** and **Uniform** Integration, respectively.

**Logistic Integration.** In Gaussian diffusion models, the log-SNR integral is approximated via importance sampling with a truncated logistic distribution. The integrand, shaped by a mixture of logistic CDFs influenced by data covariance eigenvalues $\lambda_i$, is captured by matching the empirical mean $\mu$ and variance $s$ of $-\log \lambda_i$, with integration bounds $[\mu - 4s, \mu + 4s]$. Samples drawn via the logistic quantile function are weighted by $1/q(\alpha)$ to prioritize critical regions, reducing variance.

**Uniform Integration**. This simpler numerical method discretizes the log-SNR range $[\alpha_1, \alpha_2]$ into a uniform grid, applying trapezoidal or Riemann-sum integration without assuming an underlying distribution. While simple, efficiency depends on grid density for broad ranges, favoring ease over optimal sampling. The predefined range is $[-28, 37]$ with uniform sampling.

## B Experimental Details

### B.1 Training Details (contd.)

For a fair comparison, we train both CIFAR and LMD models from scratch for 600 epochs. The training starts with a learning rate of $2 \times 10^{-5}$ using the Adam optimizer. We adopt an 80-20 train-test

split for evaluating likelihoods. For image generation, we use a UNet-based model[41], while for music generation, we employ the DenseDDPM[42] and convolutional-transformer[17]-based models for the continuous embeddings (DDPM-style) and discrete domain (D3PM[17]) respectively. The training procedure ensures consistency across both domains, facilitating a meaningful comparison of their performance. It is to be noted that we train all of the models from scratch, owing to a lack of pre-trained Poisson diffusion baselines, to ensure fair comparison. Because of compute resource constraints, we train the models upto 600 epochs, which falls short of the usual amount of training required to achieve peak performance (e.g., LTJ[20] trains their models for 3600 epochs). We also restrict ourselves to 100 logSNR values per image / music sample, and restrict the number of denoising steps used in the DDPM / D3PM baselines to 100 as well (instead of 1000), to ensure fair comparison. Thus, although the relative performance of the models is preserved, the absolute values of the metrics underperform those presented in DDPM[13] and LTJ[20].

## B.2 Data and Model Normalization

We experimented with various schemes for data (Dn) before passing it through the noisy channel and for model inputs (Mn) post-noising. CIFAR-10 data is normalized to $[0, 1], [1, 2], [0, 255], [-1, 1]$; Lakh MIDI to $[0, 1], [1, 2], [0, 90], [-1, 1]$. Poisson channels cannot handle negatives and since zero inputs yield zeros, we shift inputs by $\epsilon = 10^{-6}$. For Gaussian noising, model normalization used $[0, 1]$ or $[-1, 1]$, while Poisson noising used only $[0, 1]$. The best results were achieved with $[-1, 1]$ (Gaussian) and $[1, 2]$ (Poisson) for Dn, and $[-1, 1]$ (Gaussian) and $[0, 1]$ (Poisson) for Mn. Among the integration paradigms used, logistic integrate yielded the best empirical results, and the `loc` and `scale` parameters obtained for the mid-integral range were $(6, 3)$ for Gaussian noising and $(-1, 5)$ for Poisson noising.

## B.3 Denoiser Architecture

For CIFAR-10 images, we employ a U-Net architecture [41] with residual blocks and self-attention layers. The encoder comprises four downsampling blocks (convolution $\rightarrow$ GroupNorm $\rightarrow$ SiLU) that reduce spatial resolution from $32 \times 32$ to $4 \times 4$, followed by a bottleneck with self-attention at $8 \times 8$ resolution. The decoder mirrors the encoder via transposed convolutions and skip connections. For symbolic music synthesis on Lakh MIDI, we use a DenseDDPM[53]-based architecture and a convolutional transformer[19]-based model, for the continuous-state DDPM modeling and the discrete D3PM[19] modeling respectively. For the continuous modeling, we adapt the DenseDDPM architecture from [42]. It first projects the input latent vector to an MLP hidden size (default 2048) with a single Dense layer, then runs it through 3 residual MLP blocks whose weights are modulated by a 128-dimensional sinusoidal embedding of the diffusion timestep t. After these conditioned residual blocks, it applies a LayerNorm and a final Dense layer that maps back to the original latent dimensionality, yielding the denoised output. For the discrete modeling, we adapt an NCSN++ backbone [53] with a Convolutional Transformer encoder [19]. The architecture includes a 512-dimensional embedding layer, six transformer layers with multi-head attention (8 heads) and positional encodings, and time-dependent noise conditioning.

## B.4 Symbolic Music Dataset Cleanup

We utilize the cleaned Lakh MIDI dataset [40], loading note sequences from `.npy` files with original shape $(x, 1024)$. For training, sequences are partitioned into individual 1D vectors of shape (1,1024), representing discrete musical events. So, our method directly models symbolic music as discrete 1D note sequences using Poisson diffusion, avoiding hybrid architectures or preprocessing.

## B.5 Domain-Specific Metrics

To evaluate the generation quality of our model across image and audio domains, we utilize established domain-specific metrics that quantify fidelity, diversity, and structural realism. Below, we provide descriptions and implementation details for each metric employed in our evaluation.

**Image Metrics** All image-generation metrics were computed on 40,000 randomly selected ground-truth images from the CIFAR-10 test split and 40,000 model-generated samples. Fréchet Inception Distance (FID) was evaluated with the PyTorch `torch-fidelity` package (Inception-v3 network, 2048-dimensional pool3 activations).

- **Structural Similarity Index Measure (SSIM)** [43]: SSIM measures the similarity between two images by comparing their luminance, contrast, and structure. It is defined as:

$$\text{SSIM}(x, y) = \frac{(2\mu_x\mu_y + c_1)(2\sigma_{xy} + c_2)}{(\mu_x^2 + \mu_y^2 + c_1)(\sigma_x^2 + \sigma_y^2 + c_2)}$$

  where $\mu$ and $\sigma$ denote mean and standard deviation over local image patches. Higher SSIM indicates better perceptual similarity.

- **Fréchet Inception Distance (FID)** [44]: FID evaluates the distance between real and generated image distributions in the feature space of a pretrained Inception network. It is calculated as:

$$\text{FID} = \|\mu_r - \mu_g\|^2 + \text{Tr}\left(\Sigma_r + \Sigma_g - 2(\Sigma_r\Sigma_g)^{1/2}\right)$$

  where $(\mu_r, \Sigma_r)$ and $(\mu_g, \Sigma_g)$ are the means and covariances of the feature embeddings of real and generated samples.

**Audio Metrics.** All audio-based metrics are computed using 10,000 ground-truth samples and 10,000 generated samples per model. To enable consistent audio evaluation, we first convert model-generated `.npy` files to MIDI format using the `pretty_midi` library. These MIDI files are then rendered to WAV audio using `FluidSynth` [58] with the `FluidR3_GM` soundfont, ensuring uniform timbre across all samples. All tools and dependencies are managed within an automated evaluation pipeline. This standardized conversion procedure ensures reproducibility and fair comparison of audio metrics across all models.

- **Fréchet Audio Distance (FAD)** [45]: Analogous to FID, FAD computes the Fréchet distance between embeddings of real and generated audio, extracted via a VGGish model pretrained for audio classification. It reflects perceptual similarity in the feature space and is calculated similarly to FID.

- **Consistency (C)**: To evaluate sequence-level realism, we employ framewise self-similarity based on overlapping Gaussian approximations of pitch histograms. Specifically, we use the overlapping area (OA) from [46], applied to pitch only (since duration is fixed in our setup). For sliding 4-measure windows with 2-measure hop:

$$\text{OA}(k, k+1) = 1 - \text{erf}\left(\frac{c - \mu_1}{\sqrt{2}\,\sigma_1}\right) + \text{erf}\left(\frac{c - \mu_2}{\sqrt{2}\,\sigma_2}\right)$$

  The resulting pitch OA values are compared to ground-truth sequences via:

$$\text{C} = \max\left(0, 1 - \frac{|\mu_{\text{OA}} - \mu_{\text{GT}}|}{\mu_{\text{GT}}}\right)$$

$$\text{Var} = \max\left(0, 1 - \frac{|\sigma_{\text{OA}}^2 - \sigma_{\text{GT}}^2|}{\sigma_{\text{GT}}^2}\right)$$

  Consistency (C) measures global similarity to ground truth, while variance (Var) captures generation diversity. High C implies structured, music-like pitch transitions.

- **Mel-Spectrogram Inception Distance (MSID)** [47]: MSID adapts FID for audio by computing the Fréchet distance over features extracted from Mel spectrograms. The key steps include:

  - Convert generated `.npy` files to MIDI and synthesize audio using `FluidSynth`.
  - Compute 128-band Mel spectrograms (16kHz, FFT=2048, hop=512), as outlined in B.5.
  - Extract features using a VGG16-based architecture trained on audio (VGGish).
  - Compute MSID using: $\text{MSID} = \|\mu_r - \mu_g\|^2 + \text{Tr}(\Sigma_r + \Sigma_g - 2(\Sigma_r\Sigma_g)^{1/2})$

  MSID captures both spectral and perceptual differences, correlating with human audio quality judgments.

- **Wasserstein Distance (WD)** [48]: WD quantifies the distance between the token distributions of real and generated symbolic music. We compute a *weighted Wasserstein distance* that prioritizes important token types (e.g., binary onsets or active pitches):

$$W_w(p, q) = \inf_{\gamma \in \Pi(p,q)} \mathbb{E}_{(x,y) \sim \gamma}[c(x, y) \cdot w(x, y)]$$

Weights are assigned based on token values: 0.2 for 0s, 0.5 for 1s, 1.0 for others. Tokens are normalized and reshaped as needed. Lower WD values indicate better alignment of pitch activation distributions.

In addition to the core domain-specific metrics described in Appendix B.5, we include the following complementary metrics used for additional analysis presented in Table 4. These metrics help analyze fine-grained perceptual and structural properties of the generated data.

**Images:**

- **Learned Perceptual Image Patch Similarity (LPIPS)** [59]: LPIPS measures perceptual similarity by computing the distance between deep features extracted from pretrained vision networks (e.g., VGG, AlexNet). It is defined as:

$$\text{LPIPS}(x, y) = \sum_l \frac{1}{H_l W_l} \sum_{h,w} \| w_l \odot (\phi_l^x(h, w) - \phi_l^y(h, w)) \|_2^2$$

where $\phi_l^x$ and $\phi_l^y$ are feature activations from layer $l$, and $w_l$ are learned weights. Lower LPIPS values indicate higher perceptual similarity between generated and reference images.

**Audio:**

- **Spectral Convergence (SC)**: SC quantifies the relative difference between the magnitude spectra of real and generated audio:

$$\text{SC} = \frac{\| |S_{\text{gen}}| - |S_{\text{ref}}| \|_F}{\| |S_{\text{ref}}| \|_F}$$

where $S_{\text{gen}}$ and $S_{\text{ref}}$ are the STFTs (Short-Time Fourier Transforms) of generated and reference audio, and $\| \cdot \|_F$ denotes the Frobenius norm. Lower SC suggests higher spectral alignment.

- **Log Mean Spectral Distance (LMSD)**: LMSD captures differences in log-scaled spectral magnitudes and is defined as:

$$\text{LMSD} = \frac{1}{T} \sum_t \| \log(\epsilon + |S_{\text{gen}}(t)|) - \log(\epsilon + |S_{\text{ref}}(t)|) \|_1$$

where $\epsilon$ is a small constant to ensure numerical stability, and the summation is over time frames $t$. Lower LMSD implies improved perceptual quality in frequency response.

- **Variance (Pitch Histogram Diversity)**: [42] As described in Appendix B.5, we also compute the pitch variance metric (*Var*) to measure structural diversity in symbolic music:

$$\text{Var} = \max \left( 0, 1 - \frac{|\sigma_{\text{OA}}^2 - \sigma_{\text{GT}}^2|}{\sigma_{\text{GT}}^2} \right)$$

Higher variance indicates greater distributional diversity while maintaining similarity to ground truth statistics. Together, these metrics offer a comprehensive, multi-faceted evaluation of image and audio generation quality, balancing fidelity, diversity, and perceptual structure.

**Mel Spectrogram Computation Parameters:**

For the listed audio-based metrics (FAD, MSID, SC, LMSD), we first convert generated symbolic music into waveform as discussed earlier [58] and compute Mel spectrograms with the following parameters:

- **Sampling rate:** 16 kHz — chosen to balance temporal resolution and frequency coverage for symbolic music.
- **FFT size:** 2048 — defines the window size for frequency analysis. This size gives sufficient frequency granularity ($\approx$7.8 Hz per bin at 16 kHz).
- **Hop length:** 512 — determines the stride between successive STFT windows, corresponding to 32 ms hop (suitable for music temporal structure).
- **Mel bands:** 128 — provides a perceptually motivated representation of frequency, emphasizing resolution in lower frequency ranges where musical structure is denser.

These parameters are consistent with best practices in neural audio synthesis [60],[47] and ensure compatibility with pretrained perceptual models like VGGish.

**Additional Metrics:**

Table 4: Auxiliary generative quality metrics. Image: LPIPS; Audio: SC, LMSD, LPIPS (Mel), Var

| Baseline | LPIPS (Img) | SC | LMSD | LPIPS (Mel) | Var |
|---|---|---|---|---|---|
| IDDPM [38] | $0.17 \pm 0.05$ | 1.56 | 9.99 | $0.38 \pm 0.10$ | 0.81 |
| LTJ [20] | $0.18 \pm 0.06$ | 1.51 | 9.81 | $0.33 \pm 0.10$ | 0.87 |
| D3PM [17] | $0.29 \pm 0.09$ | **1.41** | **9.63** | **$0.28 \pm 0.09$** | **0.90** |
| ItDPDM | $0.18 \pm 0.08$ | 1.49 | 9.71 | $0.30 \pm 0.09$ | 0.85 |

### B.5.1    Visualizing generated music samples:

**Individual ItDPDM Samples:** To examine local model behavior, we present isolated piano roll visualizations of individual samples (see Figure 8). Each plot shows the temporal and pitch structure of a single sequence, with color indicating note velocity. These visualizations enable detailed inspection of rhythmic patterns, pitch range, note density, and artifacts.

For example, ItDPDM-generated samples exhibit consistent pitch contours and relatively uniform spacing, occasionally disrupted by outlier notes or sparse regions. Such plots help diagnose issues like over/under-generation, discontinuities, or anomalies, and complement the broader comparisons across models.

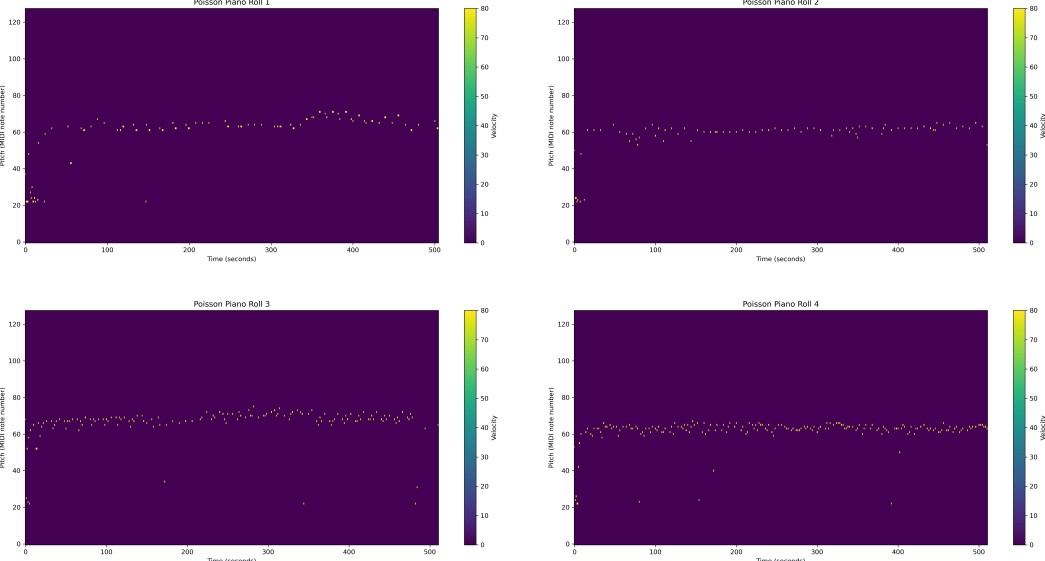

Figure 8: Isolated piano roll visualizations of four ItDPDM-generated samples. Each plot shows pitch over time, with note velocity indicated by color intensity.

**Qualitative comparison:** To qualitatively observe the generative performance of our models, we visualize representative samples as piano rolls in Figure 9. Each row presents a different generated

sequence, with columns corresponding to different models: DDPM (left), ASD3PM (center), and ItDPDM (right). Each piano roll plot depicts note pitch (vertical axis) over time (horizontal axis), with intensity indicating note onset.

**DDPM (left):** Samples from DDPM display high variability in pitch and rhythm, with note events appearing scattered and less structured. While diverse, these outputs often lack recognizable musical motifs or rhythmic regularity, indicating that the model struggles to capture long-range musical structure.

**ASD3PM (center):** ASD3PM outputs, derived from perturbed ground truth MIDI sequences, exhibit strong rhythmic and melodic coherence. These samples closely mirror the structure of real music, featuring sustained motifs, consistent phrasing, and regular timing. This visual consistency aligns with the model's design, which prioritizes fidelity to the data manifold.

**ItDPDM (right):** Samples from ItDPDM demonstrate improved musical structure over DDPM. While some randomness remains, many outputs show rhythmic grouping, pitch contours, and repeating patterns, suggesting the model's ability to learn and replicate fundamental elements of musical organization. Overall, the visualizations highlight key differences in generative behavior. ASD3PM achieves the highest structural fidelity, followed by ItDPDM, which balances diversity with coherence. DDPM produces varied outputs but lacks the structured rhythmic and melodic features observed in the other methods. These qualitative findings complement our quantitative results, offering insight into how each model captures musical dependencies in time and pitch.

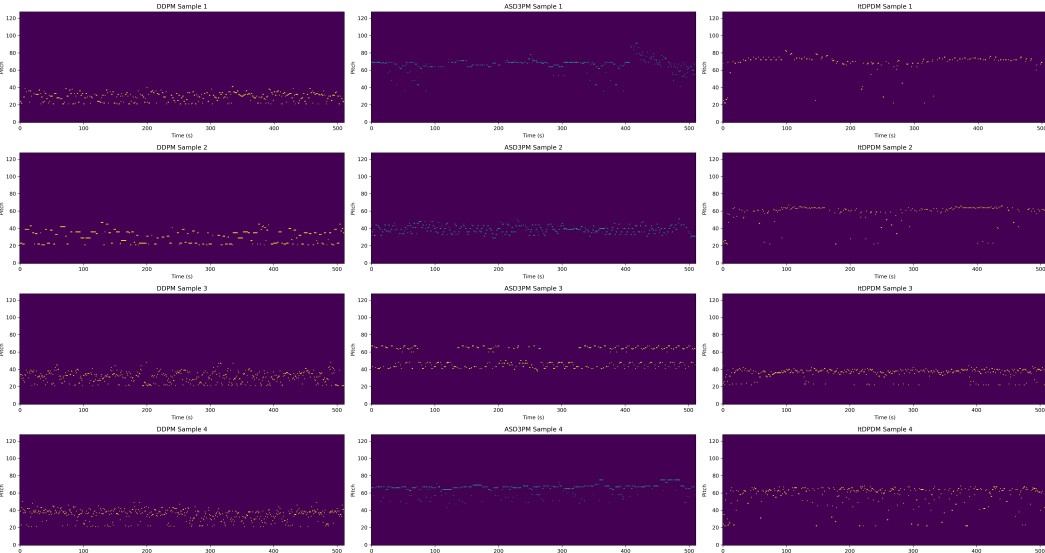

Figure 9: Piano roll visualizations of generated samples from DDPM (left), ASD3PM (middle), and ItDPDM (right). Each row corresponds to a particular random sample. Higher vertical positions represent higher pitches.

To further assess how the generated music matches the statistical properties of the training data, we also compare the generated pitch distributions with the ground truth. Figure 10 shows the histogram of MIDI pitch values for ItDPDM generated sequences alongside the empirical distribution from the training data with a close alignment indicating that the model captures global pitch statistics, such as register, range, and note density. Another observation is that in the generated samples, the note velocity is *slightly amplified* in comparison to the ground truth distribution.

## C   Synthetic Benchmark Details

### C.1   Discrete benchmark details

We evaluate model performance on a suite of synthetic univariate discrete distributions designed to challenge generative models with features such as overdispersion, multimodality, sparsity, and skewness. All distributions take values in $\mathbb{N}_0$ and are non-negative.

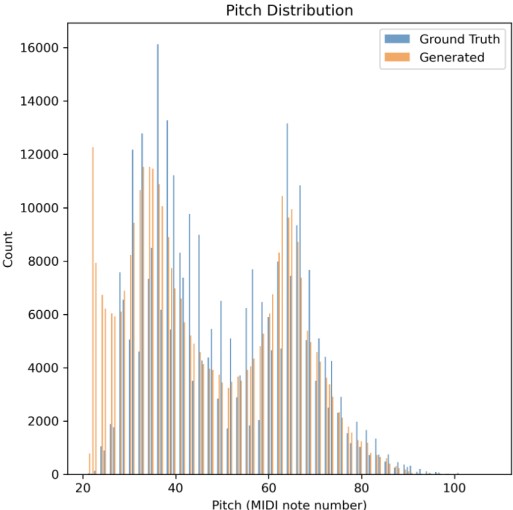

Figure 10: Comparing pitch distributions for ground truth and ItDPDM generated samples

**Poisson Mixture (PoissMix):** This is a bimodal mixture of Poisson distributions:

$$0.1 \cdot \text{Poisson}(\lambda = 1) + 0.9 \cdot \text{Poisson}(\lambda = 100),$$

producing a highly skewed and dispersed distribution with modes at both low and high counts, simulating tasks where most values are large but a minority remain near zero.

**Zero-Inflated Poisson (ZIP):** To simulate data with an excess of zeros, we use a zero-inflated Poisson distribution: which samples zero with probability $\pi_0$, and otherwise follows a Poisson distribution:

$$P(k) = \begin{cases} \pi_0 + (1 - \pi_0) \cdot e^{-\lambda}, & k = 0 \\ (1 - \pi_0) \cdot \dfrac{e^{-\lambda}\lambda^k}{k!}, & k > 0 \end{cases} \text{ with } \pi_0 = 0.7, \, \lambda = 5.$$

This models structured sparsity common in count data with dropout.

**Negative Binomial Mixture (NBinomMix):** This is a mixture of two negative binomial distributions: $0.8 \cdot \text{NB}(1, 0.9) + 0.2 \cdot \text{NB}(10, 0.1)$, where the first mode has high probability near zero, while the second exhibits broader dispersion. It introduces skew and multimodality in count data.

**Beta-Negative-Binomial (BNB):** The BNB distribution integrates a Beta prior over the success probability $p$ of the negative binomial:

$$P(k) = \int_0^1 \text{NB}(k; 1, p) \cdot \text{Beta}(p; a = 1.5, b = 1.5) \, dp, \, k \in \mathbb{N}_0.$$

We use parameters $a = 0.5$, $b = 1.5$, and $r = 5$, inducing a heavy-tailed count distribution with long-range dependencies.

**Zipf Distribution:** This power-law distribution is defined as:

$$P(x) = \frac{x^{-\alpha}}{\zeta(\alpha)}, \, \alpha = 1.7$$

, where $\zeta(\alpha)$ is the Riemann zeta function. Zipf distributions model naturally occurring frequencies, such as word counts or node degrees.

**Yule–Simon Distribution:** The Yule–Simon distribution is defined as:

$$P(k) = \rho \cdot B(k, \rho + 1) = \rho \cdot \frac{\Gamma(k)\Gamma(\rho + 1)}{\Gamma(k + \rho + 1)}, \quad \rho = 2.0, \, k \in \mathbb{N}_{\geq 1},$$

where $B$ is the Beta function and $\Gamma$ is the gamma function. It is used to model data with power-law decay, often arising in preferential attachment or self-reinforcing (e.g. rich-get-richer) processes. These distributions form a challenging testbed for evaluating generative performance on discrete, non-negative data.

Table 5 summarizes the discrete synthetic benchmarks used in our study. Each distribution is selected to represent a different pathological regime—bi-modality, zero-inflation, overdispersion, or power-law behavior—intended to stress PMF concentration and test model robustness. For completeness, we specify parameter values used in generation and annotate tail behaviors to clarify their impact on sample complexity and generalization.

| Distribution | Parameters | Tail behaviour |
|---|---|---|
| PoissMix | $\lambda = \{1, 100\}$ | bi-modal |
| Zero-Inflated Poisson | $\pi_0 = 0.7$, $\lambda = 5$ | spike at 0 |
| NBinomMix | $(r, p) = \{(1, 0.9), (10, 0.1)\}$ | Var > E |
| BNB | $a = 0.5$, $b = 1.5$, $r = 5$ | power-law |
| Zipf | $\alpha = 1.7$ | $\sim x^{-\alpha}$ |
| Yule-Simon | $\rho = 2.0$ | heavier than Zipf |

Table 5: Specification of discrete synthetic benchmarks. All distributions are heavy-tailed, zero-inflated, or multi-modal, stressing PMF concentration.

## C.2 Training Details & Metrics

In addition to the ConditionalMLP, a timestep embedding network additionally projects diffusion steps into a 64-dimensional space using SiLU activations. Models are trained for 200 epochs using the Adam optimizer ($\eta = 10^{-3}$, $\beta_1 = 0.9$, $\beta_2 = 0.999$) with a batch size of 128. The Gaussian DDPM employs a linear noise schedule $\beta_t \in [10^{-4}, 2 \cdot 10^{-2}]$ over $T = 100$ diffusion steps. Our ItDPDM framework adopts a linear gamma schedule $\gamma_t \in [1.0, 0.0]$ over the same number of steps. For Poisson diffusion, the initial sample mean is set to 10.0.

**Wasserstein-1 distance** Wasserstein-1 distance [48] between two univariate distributions $p$ and $q$ is defined as: $W_1(p, q) := \int_{\mathbb{R} \times \mathbb{R}} |x - y| \, d\pi(x, y) = \int_{\mathbb{R}} |P(x) - Q(x)| \, dx$, where $\pi(x, y)$ is a joint coupling of $p$ and $q$, and $P, Q$ are their respective cumulative distribution functions (CDFs). When $p$ and $q$ are empirical distributions of the same size $n$, this reduces to: $W_1(p, q) = \frac{1}{n} \|\text{sort}(X) - \text{sort}(Y)\|_1$, where $X, Y \in \mathbb{R}^n$ are the sorted samples from $p$ and $q$.

For each empirical distribution of 50,000 generated samples over 5 runs, say $\hat{p}_{gen}$ (with $\hat{p}_{test}$ denoting the empirical distribution of 50,000 test samples), we compute the Wasserstein-1 distance (WD) [48] and negative log-likelihood (NLL) as:

$$\text{WD} = W_1(\hat{p}_{\text{test}}, \hat{p}_{\text{gen}}), \ \text{NLL} = -\frac{1}{n_{\text{test}}} \sum_i \log \hat{p}_{\text{gen}}(x_i) \tag{24}$$

where $x_i$ denote the held–out samples.

## C.3 Probability Mass Function Estimation:

For discrete distributions, we estimate the empirical probability mass function (PMF) $\hat{p}(x)$ from generated samples $\{x_i\}_{i=1}^N$ using a histogram-based approach with binning over a finite support $\mathcal{X} = \{0, 1, \ldots, K\}$:

$$\hat{p}(x) = \frac{1}{N} \sum_{i=1}^N \mathbb{I}(x_i = x), \tag{25}$$

where $\mathbb{I}(\cdot)$ is the indicator function and $K$ is the truncation value. We set $K = 50$ across all experiments to standardize the support. To reduce sampling noise and better visualize differences across models, we additionally compute a *smoothed PMF estimate* using a discrete Gaussian kernel:

$$\hat{p}_{\text{smooth}}(x) = \frac{1}{N} \sum_{i=1}^N K_h(x - x_i), \tag{26}$$

where $K_h(\cdot)$ is a Gaussian kernel defined on the integer lattice:

$$K_h(x) = \frac{1}{Z} \exp\left(-\frac{x^2}{2h^2}\right), \tag{27}$$

with normalization constant $Z = \sum_{x' \in \mathcal{X}} \exp\left(-\frac{x'^2}{2h^2}\right)$ ensuring that $K_h$ sums to 1 over the support. The bandwidth $h$ is selected empirically per distribution to balance smoothness and fidelity to the empirical histogram. To assess variability in PMF estimation, we also compute error bands via non-parametric bootstrapping. Specifically, we generate 10 bootstrap resamples of the model outputs, re-estimate the (smoothed) PMF for each, and plot the mean $\pm$ standard deviation across these resampled estimates. Each plot includes in Fig. 6 includes: a) ground-truth PMF (when known), and b) the empirical unsmoothed and smoothed PMFs for each model (e.g., ItDPDM, DDPM, LTJ), with any shaded error bands reflecting bootstrap variability.

**Implementation Details:**

| Aspect | Details |
|---|---|
| Sample size | $N = 10{,}000$ samples per model and distribution |
| Support | $\mathcal{X} = \{0, 1, \ldots, 50\}$ for discrete; bounded $x$ for continuous |
| Smoothing bandwidth | $h$ tuned per distribution (discrete); KDE bandwidth default |
| Bootstrap | 10 resamples per model for uncertainty estimation |
| Visualization | True distribution, model estimates, and error bands plotted |

Table 6: Summary of implementation settings for PMF and PDF estimation.

**Zoomed-in look at PMF plots:** Building on the analysis in Section 5, Figure 11 and Figure 12 provides a magnified view of the Yule–Simon and Zipf fits produced by each model. ItDPDM exhibits the closest alignment to the target distribution, particularly in the critical low-support region.

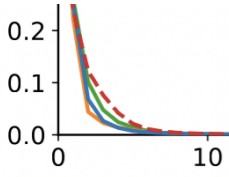

Figure 11: Zoomed-in Yule–Simon fits                Figure 12: Zoomed-in Zipf law fits

### C.4  Non-negative Continuous Scenarios

As stated earlier, we extend our analysis to six skewed continuous densities: Gamma, Log-Normal, Lomax, Half-Cauchy, Half-t, Weibull, (along with Beta and Uniform distributions) as outlined in this section. Our goal here is to assess how well generative models capture asymmetry, concentration, and long-range dependencies in continuous data.

**Descriptions and parameters:**

**Gamma Distribution:** The Gamma distribution is defined by a shape parameter '$a$' and a scale parameter '$\theta$':

$$p(x) = \frac{1}{\Gamma(a)\theta^a} x^{a-1} e^{-x/\theta}, \ x \geq 0.$$

We use $a = 0.5$, $\theta = 2$, which produces a sharp mode near zero and a long right tail. Gamma distributions are commonly used to model wait times, energy release, and insurance claims—making them valuable for stress-testing the model's handling of high variance and positive skew.

**Log-Normal Distribution:** A log-normal distribution arises when the logarithm of a variable is normally distributed:

$$p(x) = \frac{1}{xs\sqrt{2\pi}} \exp\left(-\frac{(\log x - \mu)^2}{2s^2}\right), \ x > 0.$$

We use $\mu = 0$, $s = 1.5$, producing a distribution with significant positive skew and heavy tails. Log-normal models appear in financial returns, biological measurements, and natural language modeling, where multiplicative effects dominate.

**Lomax Distribution:** Also known as the Pareto Type II distribution, the Lomax is defined as:

$$p(x) = \frac{c}{s} \left(1 + \frac{x}{s}\right)^{-(c+1)}, \ x \geq 0.$$

We use $c = 2.0$, $s = 1.0$, resulting in a fat-tailed distribution often used in reliability engineering and modeling rare, catastrophic events. It challenges models to capture high-probability mass near zero with occasional large outliers.

**Half-Cauchy Distribution:** The Half-Cauchy is the positive part of a Cauchy distribution:

$$p(x) = \frac{2}{\pi s \left[1 + \left(\frac{x}{s}\right)^2\right]}, \ x \geq 0.$$

With $s = 1$, this distribution has undefined mean and variance, and extremely heavy tails. It is commonly used as a prior in hierarchical Bayesian models due to its robustness to outliers.

**Half-t Distribution:** The Half-*t* distribution is the absolute value of a Student's *t*-distributed variable:

$$p(x) = 2 \cdot t(x; \nu, 0, s), \ x \geq 0.$$

We use $\nu = 3$, $s = 1$, yielding a distribution with heavy but finite tails. This is another robust prior used in Bayesian inference, particularly for variances in hierarchical models, where it prevents over-shrinkage.

**Weibull Distribution:** The Weibull distribution, defined by shape $k$ and scale $\lambda$, is given by:

$$p(x) = \frac{k}{\lambda} \left(\frac{x}{\lambda}\right)^{k-1} e^{-(x/\lambda)^k}, \ x \geq 0.$$

We use $k = 1.5$, $\lambda = 1$, producing a distribution with increasing hazard rate and moderate skew. This is widely used in survival analysis, material failure modeling, and wind speed distributions.

**Beta Distribution (bounded support):** Though often used on $[0, 1]$, the Beta distribution provides diverse shapes depending on the parameters:

$$p(x) = \frac{\Gamma(a + b)}{\Gamma(a)\Gamma(b)} x^{a-1}(1 - x)^{b-1}, \ 0 \leq x \leq 1.$$

We use $a = 2$, $b = 2$, leading to a density concentrated near zero. The Beta distribution tests the model's ability to learn bounded distributions with asymmetric mass concentration, relevant in probabilistic modeling and reinforcement learning. A key limitation to note here is that in case of asymmetric/skewed beta distributions, all the models notably fail to learn the distribution.

**Uniform Distribution (flat support):** The uniform distribution provides a baseline for bounded, structureless densities:

$$p(x) = \frac{1}{b - a}, \ a \leq x \leq b.$$

We set $a = 0$, $b = 1$, resulting in a constant density over the unit interval. Although simple, it serves as a sanity check for model calibration and ability to avoid mode collapse under flat distributions. Together, these distributions offer a comprehensive testbed for evaluating generative modeling under varied support, skewness, and tail behavior. They also represent common scenarios encountered in practice, ensuring relevance to real-world generative tasks.

**Results:**

Table 7 compares the Wasserstein distance for all the continuous cases, and in the continuous case, we omit NLL values as they can be overly sensitive to skewness and outliers, making them unreliable for fair comparison. More critically, whereas the true NLL in continuous distributions can often be negative while our discrete estimator cannot possibly yield a negative NLL.

For each distribution, we visualize the estimated PDFs from all models alongside the true density. Figure 13 summarizes the results across all eight distributions, providing a qualitative comparison of how closely each model recovers the underlying data-generating process.

Table 7: WD for continuous cases (↓ lower is better). Bold indicates best.

| | WD | | |
|---|---|---|---|
| **Distribution** | **DDPM** | **ItDPDM** | **LTJ** |
| Gamma | $0.27 \pm 0.09$ | $\mathbf{0.12 \pm 0.05}$ | $0.14 \pm 0.05$ |
| Log-Normal | $2.39 \pm 0.53$ | $\mathbf{1.94 \pm 0.71}$ | $1.99 \pm 0.66$ |
| Lomax | $0.39 \pm 0.20$ | $\mathbf{0.31 \pm 0.17}$ | $1.15 \pm 0.41$ |
| Half-Cauchy | $6.67 \pm 2.45$ | $6.35 \pm 2.56$ | $\mathbf{5.45 \pm 2.23}$ |
| Half-t | $\mathbf{0.20 \pm 0.07}$ | $0.21 \pm 0.02$ | $0.22 \pm 0.04$ |
| Weibull | $0.29 \pm 0.05$ | $\mathbf{0.23 \pm 0.02}$ | $0.23 \pm 0.06$ |
| Beta | $0.28 \pm 0.07$ | $\mathbf{0.18 \pm 0.03}$ | $0.19 \pm 0.06$ |
| Uniform | $0.12 \pm 0.05$ | $0.12 \pm 0.03$ | $0.12 \pm 0.02$ |

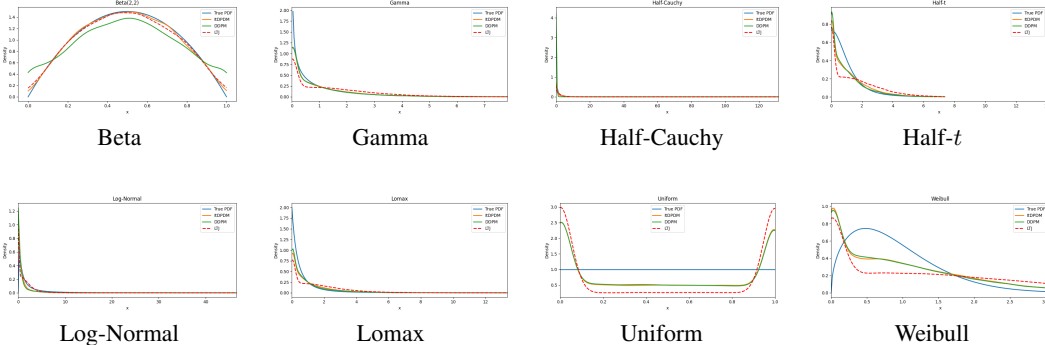

| Beta | Gamma | Half-Cauchy | Half-$t$ |
|---|---|---|---|

| Log-Normal | Lomax | Uniform | Weibull |
|---|---|---|---|

Figure 13: Comparison of estimated PDFs for various continuous distributions in the synthetic dataset. Each plot shows the true distribution and model-generated estimates.

**PDF estimation:**

For continuous non-negative distributions, we estimate the probability density function (PDF) $\hat{f}(x)$ using kernel density estimation (KDE) with a Gaussian kernel:

$$\hat{f}(x) = \frac{1}{N} \sum_{i=1}^{N} \frac{1}{\sqrt{2\pi}h} \exp\left(-\frac{(x - x_i)^2}{2h^2}\right), \text{ where by default } h = \sigma N^{-1/(d+4)}, \ d = 1 \Rightarrow h = \sigma N^{-1/5},$$

(28)

with $\sigma$ denoting the sample standard deviation of $\{x_i\}$ and $N$ the number of samples.

We compute error bands by bootstrapping: for each model, we resample its generated samples 10 times, compute the KDE for each resample, and display the mean $\pm$ standard deviation across estimates. For bounded distributions (e.g., Beta, Uniform), we clip model-generated samples to the distribution's support before applying KDE. Each PDF plot includes: a) ground-truth PDF, and b) the average KDE for each model, with any shaded error bands indicating bootstrap uncertainty.

### C.5 Experiments for Multivariate Setting

While most of the empirical validation focuses on the univariate formulation for clarity, we additionally evaluate `ItDPDM` and other baselines on multivariate (2D) discrete distributions to demonstrate its ability to capture joint count dependencies.

**Independent multivariate setting:** We first construct bivariate distributions as products of two independent univariate marginals using the same parameters as in the 1D synthetic experiments (see App.C.2 above). The Wasserstein Distance (WD) metrics averaged over 5 runs are summarized in Table 8, showing that `ItDPDM` maintains consistent improvements over both DDPM and LTJ baselines.

**Dependent multivariate setting:** We further study two correlated 2D count models to validate `ItDPDM`'s flexibility in learning structured dependencies:

- **Bivariate Poisson:** Let $U \sim \text{Pois}(\lambda_0)$, $V_1 \sim \text{Pois}(\lambda_1)$, and $V_2 \sim \text{Pois}(\lambda_2)$ be independent; define $X_1 = U + V_1$, $X_2 = U + V_2$, giving marginals $X_1 \sim \text{Pois}(\lambda_0 + \lambda_1)$ and $X_2 \sim \text{Pois}(\lambda_0 + \lambda_2)$. For $\lambda_0 = 3$, $\lambda_1 = 2$, $\lambda_2 = 2$, we obtain correlated Poisson pairs $(X_1, X_2)$.

Table 8: Independent bivariate (2D) extensions of univariate synthetic datasets.

| Distribution | DDPM | ItDPDM | LTJ |
|---|---|---|---|
| PoissMix | $7.90 \pm 0.64$ | $\mathbf{2.65 \pm 0.23}$ | $2.96 \pm 0.29$ |
| ZIP | $4.89 \pm 0.72$ | $\mathbf{1.78 \pm 0.27}$ | $1.92 \pm 0.38$ |
| NBinomMix | $10.83 \pm 0.88$ | $3.53 \pm 0.57$ | $\mathbf{3.26 \pm 0.48}$ |
| BNB | $5.07 \pm 0.72$ | $\mathbf{1.82 \pm 0.36}$ | $1.67 \pm 0.41$ |
| Zipf | $3.51 \pm 0.58$ | $\mathbf{0.97 \pm 0.21}$ | $1.49 \pm 0.32$ |
| YS | $0.87 \pm 0.23$ | $\mathbf{0.32 \pm 0.05}$ | $0.36 \pm 0.08$ |

- **Gamma–Poisson (compound Poisson):** Let $\Theta \sim \text{Gamma}(k = 2, \beta = 1)$, and draw $X_i | \Theta \sim \text{Pois}(\Theta)$ independently for $i = 1, 2$.

The WD scores for these dependent models are reported in Table 9, showing strong performance gains for `ItDPDM`.

Table 9: Dependent bivariate count distributions.

| Distribution | DDPM | ItDPDM | LTJ |
|---|---|---|---|
| Bivariate Poisson | 3.502 | **0.826** | 1.323 |
| Gamma–Poisson | 0.855 | **0.337** | 0.624 |

Overall, these results demonstrate that the proposed information-theoretic Poisson diffusion framework (ItDPDM) scales gracefully to multivariate discrete settings without additional architectural modifications.

# D Section 3 Proofs

## D.1 On the Poisson Loss Function:

Here, as outlined in 3.2, we establish that the function $l_0(x) = x \log x - x + 1$ serves as the convex conjugate of the Poisson distribution's log moment generating function (log MGF). We begin by deriving the log MGF of the Poisson distribution, and finally computing its convex conjugate through the Legendre-Fenchel transform. Let $X$ be a random variable following a Poisson distribution with parameter $\lambda > 0$. The probability mass function (PMF) of $X$ is given by:

$$P(X = k) = \frac{\lambda^k e^{-\lambda}}{k!}, \quad \text{for } k = 0, 1, 2, \ldots$$

The moment generating function (MGF) can be evaluated as:

$$M_X(t) = E[e^{tX}] = \sum_{k=0}^{\infty} e^{tk} P(X = k) = \sum_{k=0}^{\infty} e^{tk} \frac{\lambda^k e^{-\lambda}}{k!} = e^{-\lambda} \sum_{k=0}^{\infty} \frac{(\lambda e^t)^k}{k!} = e^{-\lambda} e^{\lambda e^t} = e^{\lambda(e^t - 1)}$$

Let $\phi(t)$ be the log moment generating function as shown:

$$\phi(t) = \log M_X(t) = \lambda(e^t - 1)$$

Without any loss of generality, let $\lambda = 1$ (since scaling does not affect the form of the conjugate), implying $\phi(t) = e^t - 1$. The **convex conjugate** of a convex function $\phi : \mathbb{R} \to \mathbb{R} \cup \{+\infty\}$, denoted by $\phi^*(x)$, is defined as:

$$\phi^*(x) = \sup_{t \in \mathbb{R}} \{xt - \phi(t)\}$$

This transformation maps the original function $\phi(t)$ to its dual function $\phi^*(x)$, and then finds the supremum of linear functions subtracted by $\phi(t)$.

Let $\phi(t) = e^t - 1$ be the log moment generating function (log MGF) of a Poisson distribution with parameter $\lambda = 1$. Then, the convex conjugate of $\phi$, denoted by $\phi^*(x)$, is given by:

$$\phi^*(x) = \begin{cases} x \log x - x + 1 & \text{if } x > 0, \\ +\infty & \text{otherwise.} \end{cases}$$

*Proof.* By definition: $\phi^*(x) = \sup_{t \in \mathbb{R}}\{xt - \phi(t)\} = \sup_{t \in \mathbb{R}} \{xt - e^t + 1\}$

To find the supremum, we find the value of $t$ that maximizes this expression. First-order conditions imply: $\frac{d}{dt}\left(xt - e^t\right) = x - e^t = 0$ so we have $t = \log x$. This critical point exists only if $x > 0$, as $e^t > 0$ for all $t \in \mathbb{R}$. From the second-order condition, we get:

$$\frac{d^2}{dt^2}\left(xt - e^t\right) = -e^t < 0 \quad \forall t \in \mathbb{R}$$

The negative second derivative confirms that the function is concave at $t = \log x$, ensuring a global maximum at this point. So for $t = \log x$,

$$\phi^*(x) = x(\log x) - e^{\log x} + 1 = x \log x - x + 1$$

Therefore, for $x > 0$:

$$\phi^*(x) = x \log x - x + 1$$

For $x \leq 0$, the supremum is unbounded above, leading to: $\phi^*(x) = +\infty$ Combining these cases gives:

$$\phi^*(x) = \begin{cases} x \log x - x + 1 & \text{if } x > 0, \\ +\infty & \text{otherwise.} \end{cases}$$

This establishes that $l_0(x) = x \log x - x + 1$ is the convex conjugate of the Poisson distribution's log moment generating function $\phi(t) = e^t - 1$ and therefore, a natural loss function.

**Connection to Bregman Divergence**

The Poisson loss function we defined $l(x, \hat{x})$ is a member of the broader family of Bregman divergences, which are pivotal in various domains such as machine learning, information theory, and optimization. A Bregman divergence is defined for a strictly convex and differentiable function $\psi : \mathbb{R}^d \to \mathbb{R}$ as follows:

$$\mathcal{L}_\psi(x, \hat{x}) = \psi(x) - \psi(\hat{x}) - \langle \nabla\psi(\hat{x}), x - \hat{x} \rangle,$$

where $\langle \cdot, \cdot \rangle$ denotes the inner product in $\mathbb{R}^d$, and $\nabla\psi(\hat{x})$ represents the gradient of $\psi$ evaluated at $\hat{x}$.

For the Poisson loss function, the generating function $\psi$ is chosen as:

$$\psi(x) = x \log x - x.$$

Substituting this into the Bregman divergence definition yields:

$$\mathcal{L}_\psi(x, \hat{x}) = x \log x - x - (\hat{x} \log \hat{x} - \hat{x}) - (\log \hat{x} \cdot (x - \hat{x})).$$

Simplifying the expression, we obtain:

$$\mathcal{L}_\psi(x, \hat{x}) = x \log \left(\frac{x}{\hat{x}}\right) - x + \hat{x},$$

which is precisely the Poisson loss function $l(x, \hat{x})$.

This framework not only encapsulates the Poisson loss but also generalizes it to encompass other widely-used loss functions by merely altering the generating function $\psi$. Well-known examples include squared error loss (choosing $\psi(x) = \frac{1}{2}x^2$ and Itakura-Saito divergence (choosing $\psi(x) = -\log x$). Bregman divergences exhibit key properties that make them valuable in optimization and learning. They are **non-negative**, vanishing only when $x = \hat{x}$, due to the strict convexity of $\psi$. They are also **asymmetric**, meaning $\mathcal{L}_\psi(x, \hat{x}) \neq \mathcal{L}_\psi(\hat{x}, x)$ in general and their **projection property** enables efficient optimization over convex sets.

By leveraging the Bregman divergence framework, Poisson and Gaussian diffusion schemes can be unified under a single theoretical umbrella, where squared error loss ($\psi(x) = \frac{1}{2}x^2$) corresponds to Gaussian noise, and Poisson loss aligns with count-based data modeling. This unification enables extending optimization techniques across different noise models by adjusting the generating function $\psi$. Viewing Poisson loss function as a Bregman divergence thus broadens its theoretical and practical utility discrete data modelling.

**Optimality of Conditional Expectation**

Let $\phi : \mathbb{R}^d \to \mathbb{R}$ be a strictly convex and differentiable function. The Bregman divergence $D_\phi$ induced by $\phi$ is defined by

$$D_\phi(X, Y) = \phi(X) - \phi(Y) - \nabla\phi(Y)^\top (X - Y).$$

Consider a random variable $X \in \mathbb{R}^d$ and a sigma-algebra $\sigma(Z)$ with $Y = Y(Z)$ being any measurable function of $Z$. Let $Y^* = E[X|Z]$ denote the conditional expectation of $X$ given $Z$. The objective is to show that $Y^*$ uniquely minimizes the expected Bregman loss $E[D_\phi(X, Y)]$ among all measurable functions $Y(Z)$. For any such function $Y$, consider the difference in expected Bregman losses:

$$E[D_\phi(X, Y)] - E[D_\phi(X, Y^*)] = E\left[\phi(X) - \phi(Y) - \nabla\phi(Y)^\top (X - Y)\right] - E\left[\phi(X) - \phi(Y^*) - \nabla\phi(Y^*)^\top (X - Y^*)\right]$$

Simplifying, the terms involving $\phi(X)$ cancel out, yielding

$$E[D_\phi(X, Y)] - E[D_\phi(X, Y^*)] = E\left[\phi(Y^*) - \phi(Y) - \nabla\phi(Y)^\top (Y^* - Y)\right].$$

Recognizing that $Y^*$ is the conditional expectation $E[X|Z]$, we utilize the law of total expectation to express the above as

$$E\left[\phi(Y^*) - \phi(Y) - \nabla\phi(Y)^\top (Y^* - Y)\right] = E\left[D_\phi(Y^*, Y)\right].$$

Due to the strict convexity of $\phi$, the Bregman divergence satisfies $D_\phi(u, v) \geq 0$ for all $u, v \in \mathbb{R}^d$, with equality if and only if $u = v$. Therefore,

$$E[D_\phi(X, Y)] - E[D_\phi(X, Y^*)] = E[D_\phi(Y^*, Y)] \geq 0,$$

with equality holding if and only if $Y = Y^*$ almost surely. This establishes that

$$E[D_\phi(X, Y)] \geq E[D_\phi(X, Y^*)],$$

for all measurable functions $Y(Z)$, and thus $Y^* = E[X|Z]$ is the unique minimizer of the expected Bregman loss $E[D_\phi(X, Y)]$. .

## D.2   Section 3 Lemma Proofs

**Proof of Lemma 1: Properties of Poisson Loss**   Consider the loss function defined as $l(x, \hat{x}) = \hat{x} \cdot l_0\left(\frac{x}{\hat{x}}\right)$, where $l_0(z) = z \log z - z + 1$.

**1. Non-negativity:** Since $l_0(z)$ achieves its minimum value of 0 at $z = 1$ and is non-negative for all $z > 0$, it follows that $l(x, \hat{x}) \geq 0$ for all $x, \hat{x} > 0$. Equality holds if and only if $\frac{x}{\hat{x}} = 1$, i.e., $x = \hat{x}$.

**2. Convexity:** The function $l_0(z)$ is convex in $z$ because its second derivative $l_0''(z) = \frac{1}{z}$ is positive for all $z > 0$. Therefore, $l(x, \hat{x}) = \hat{x} \cdot l_0\left(\frac{x}{\hat{x}}\right)$ is convex in $\hat{x}$ for each fixed $x$, and similarly, it is convex in $x$ for each fixed $\hat{x}$, as the composition of a convex function with an affine transformation preserves convexity. (We can also directly use the Bregman divergence framework to argue its convexity)

**3. Scaling:** For any $\alpha > 0$, consider scaling both arguments of the loss function:

$$l(\alpha x, \alpha \hat{x}) = \alpha \hat{x} \cdot l_0\left(\frac{\alpha x}{\alpha \hat{x}}\right) = \alpha \hat{x} \cdot l_0\left(\frac{x}{\hat{x}}\right) = \alpha \cdot l(x, \hat{x}).$$

This demonstrates that the loss function scales linearly with $\alpha$.

**4. Unboundedness for Underestimation:** For any fixed $x > 0$, as $\hat{x} \to 0^+$, the ratio $\frac{x}{\hat{x}} \to \infty$. Evaluating the loss function in this limit:

$$l(x, \hat{x}) = \hat{x} \cdot \left(\frac{x}{\hat{x}} \log\left(\frac{x}{\hat{x}}\right) - \frac{x}{\hat{x}} + 1\right) = x \log\left(\frac{x}{\hat{x}}\right) - x + \hat{x}.$$

As $\hat{x} \to 0^+$, $\log\left(\frac{x}{\hat{x}}\right)$ grows without bound, causing $l(x, \hat{x}) \to \infty$. This shows that the loss becomes unbounded as $\hat{x}$ underestimates $x$.

**Proof of Lemma 2.** Let $Z_\gamma$ be a Poisson random variable with parameter $\gamma X$, meaning $Z_\gamma | X = x \sim \text{Pois}(\gamma x)$. Suppose the conditional expectation $\langle X \rangle_z = E[X|Z_\gamma = z]$ is affine in $z$,

$$\langle X \rangle_z = az + b,$$

for some $a$ and $b$, with $0 < a < 1/\gamma$ and $b > 0$. We aim to show that $X$ follows a Gamma distribution with shape $\alpha = \frac{1-\gamma a}{a}$ and rate $\beta = \frac{a}{b}$, i.e.,

$$X \sim \text{Gamma}\left(\frac{1-\gamma a}{a}, \frac{a}{b}\right).$$

Define $U = X$ and $Y = Z_\gamma \sim \mathcal{P}(\gamma U)$. Assume $E[U|Y = z] = az + b$. By the law of total expectation,

$$0 = E[(U - (aY + b))g(Y)]$$

for any function $g$ satisfying integrability. Choosing $g(Y) = e^{-tY}$ for $t > 0$,

$$E[(U - (aY + b))e^{-tY}] = 0.$$

Rewriting $Y \sim \mathcal{P}(\gamma U)$, we use the known conditional Laplace transform relation for a $\mathcal{P}(\lambda)$ random variable $Y$,

$$E[e^{-tY}|U = u] = \exp(u(\gamma(e^{-t} - 1))).$$

Hence,

$$E[e^{-tY}] = E[\exp(U\gamma(e^{-t} - 1))],$$

which is the Laplace transform of $U$ evaluated at $s = \gamma(1 - e^{-t})$. Denote

$$L_U(s) = E[e^{-sU}], \quad \text{so that} \quad E[e^{-tY}] = L_U(\gamma(1 - e^{-t})).$$

Similarly,

$$E[Ue^{-tY}] = -\frac{d}{ds}L_U(s)\Big|_{s=\gamma(1-e^{-t})}, \quad E[Ye^{-tY}] = -\frac{d}{dt}E[e^{-tY}].$$

From the orthogonality condition,

$$E[(U - (aY + b))e^{-tY}] = 0.$$

Using the above expressions,

$$0 = E[Ue^{-tY}] - aE[Ye^{-tY}] - bE[e^{-tY}].$$

Substituting $s = \gamma(1 - e^{-t})$ and differentiating as needed, we obtain a first-order linear differential equation for $L_U(s)$,

$$-((1 - a\gamma) + a\gamma s)L_U'(s) = bL_U(s).$$

The unique solution with $L_U(0) = 1$ is

$$L_U(s) = \left(1 + \frac{b}{1-\gamma a}s\right)^{-\frac{1-\gamma a}{a}}.$$

This is the Laplace transform of a $\text{Gamma}(\frac{1-\gamma a}{a}, \frac{a}{b})$ random variable. Hence, $U = X$ follows this Gamma distribution. For the Gamma distribution to be well-defined with a positive shape parameter, we require $\alpha = \frac{1-\gamma a}{a} > 0$, which holds for $0 < a < \frac{1}{\gamma}$. The rate parameter $\beta = \frac{a}{b} > 0$ requires $b > 0$. Under these conditions, $X \sim \text{Gam}(\frac{1-\gamma a}{a}, \frac{a}{b})$, completing one side of proof.

**Converse ("if" part):** The converse follows directly from Poisson–Gamma conjugacy. Suppose $X \sim \text{Gamma}(\alpha, \beta)$ and conditional on $X$, we draw $Z_\gamma \mid X \sim \text{Pois}(\gamma X)$. Then the posterior is

$$X|Z_\gamma = z \sim \text{Gamma}(\alpha + z, \beta + \gamma),$$

with mean given by

$$\mathbb{E}[X|Z_\gamma = z] = \frac{\alpha + z}{\beta + \gamma} = az + b, \quad \text{where } a = \frac{1}{\beta + \gamma}, \quad b = \frac{\alpha}{\beta + \gamma}.$$

Hence the conditional expectation is affine in $z$, establishing the "if" direction and completing the proof of Lemma 2.

**Proof of Lemma 3.** When $X = 0$ almost surely, $E[X] = 0$, and the identity holds by convention. Else, $E[X] > 0$, and we have:

$$E\left[l(X, \hat{x})\right] = E\left[X \log\left(\frac{X}{\hat{x}}\right) - X + \hat{x}\right] = E[X \log X] - E[X \log \hat{x}] - E[X] + \hat{x}$$
$$= E\left[X \log X - X \log E[X] - X + E[X]\right] + l\left(E[X], \hat{x}\right)$$
$$= E\left[l\left(X, E[X]\right)\right] + l\left(E[X], \hat{x}\right).$$

**Proof of Lemma 4.** Consider $Z_\gamma = \mathcal{P}(\gamma X)$, where $Z_\gamma$ is a Poisson random variable with parameter $\gamma X$. To determine $\langle X \rangle_z = E[X|Z_\gamma = z]$ for each $z \geq 0$, we start by applying the definition of conditional expectation:

$$\langle X \rangle_z = \frac{E\left[X \cdot P_{Z_\gamma}(z|X)\right]}{P_{Z_\gamma}(z)}.$$

Given that $Z_\gamma | X = x \sim \text{Pois}(\gamma x)$, the conditional probability mass function is

$$P_{Z_\gamma}(z|X = x) = \frac{(\gamma x)^z e^{-\gamma x}}{z!}.$$

Substituting this into the expression for $\langle X \rangle_z$ yields

$$\langle X \rangle_z = \frac{E\left[X \cdot \frac{(\gamma X)^z e^{-\gamma X}}{z!}\right]}{P_{Z_\gamma}(z)}.$$

To relate $\langle X \rangle_z$ to $P_{Z_\gamma}(z+1)$, observe that

$$P_{Z_\gamma}(z+1) = E\left[\frac{(\gamma X)^{z+1} e^{-\gamma X}}{(z+1)!}\right] = \frac{\gamma}{z+1} E\left[X \cdot \frac{(\gamma X)^z e^{-\gamma X}}{z!}\right].$$

Rearranging the above equation, we obtain

$$E\left[X \cdot \frac{(\gamma X)^z e^{-\gamma X}}{z!}\right] = \frac{(z+1)}{\gamma} P_{Z_\gamma}(z+1).$$

Substituting this back into the expression for $\langle X \rangle_z$, we have

$$\langle X \rangle_z = \frac{\frac{(z+1)}{\gamma} P_{Z_\gamma}(z+1)}{P_{Z_\gamma}(z)} = \frac{1}{\gamma} \frac{(z+1) P_{Z_\gamma}(z+1)}{P_{Z_\gamma}(z)}.$$

This completes the proof of Lemma 4.

The conditional expectation over a Poisson noise channel also has other unique properties, some of which are stated below. The next property is useful in showing that the conditional expectation in this case is unique for every input distribution.

**Lemma 5.** *Let $Z_\gamma = \mathcal{P}(\gamma X)$. Then, for every positive integer $k$ and every non-negative integer $z$,*

$$E\left[(\gamma X)^k | Z_\gamma = z\right] = \prod_{i=0}^{k-1} E\left[\gamma X | Z_\gamma = z + i\right].$$

**Proof of Lemma 5.** Let $Z_\gamma = \mathcal{P}(\gamma X)$. We claim that for every positive integer $k$ and nonnegative integer $z$,

$$E[(\gamma X)^k | Z_\gamma = z] = \prod_{i=0}^{k-1} E[\gamma X | Z_\gamma = z + i].$$

From the affine formula in Lemma 4, the conditional expectation of $\gamma X$ given $Z_\gamma = z$ is related to the ratio of marginal probabilities. More generally, for higher-order moments,

$$E[(\gamma X)^k | Z_\gamma = z] = \frac{(z+k)!}{z!} \frac{P_{Z_\gamma}(z+k)}{P_{Z_\gamma}(z)}. \tag{29}$$

We can also express $(\gamma X)^k$ as a product of $\gamma X$ terms and use the Poisson shifting property of $\mathcal{P}(\gamma X)$. Applying Lemma 4 and Eq. 29 for each shift $z \to z + i$ gives

$$E[(\gamma X)^k | Z_\gamma = z] = \prod_{i=0}^{k-1} E[\gamma X | Z_\gamma = z + i].$$

Each factor on the right captures the conditional expectation of $\gamma X$ at consecutive levels $z, z + 1, \ldots, z + k - 1$, so all higher-order moments of $\gamma X$ follow from the first conditional moment $E[\gamma X | Z_\gamma = z]$. This completes the proof.

*Proof Sketch of Eq. 29:* The key observation behind the formula is that, for the Poisson distribution, shifting from $y$ to $y + k$ multiplies the corresponding probability mass by $\frac{(aX+\lambda)^k}{k!}$. Evaluating the expectation leverages the ratio of adjacent Poisson probabilities $P_Y(y + k)/P_Y(y)$ and tracks how $(aX + \lambda)^k$ factors. In essence, a product expansion shows how each additional factor $aX + \lambda$ increases the count from $y$ to $y + 1$, and iterating this argument recovers the moment expression. As shown in [61], for Poisson observations $Z_\gamma \sim \mathcal{P}(aX + \lambda)$, the sequence of conditional expectations $\{\mathbb{E}[X | Z_\gamma = z]\}_{z \geq 0}$ uniquely determines the input distribution $P_X$. This supports our information-theoretic derivation and strengthens the foundation for learning in discrete-state noise models. For our Poisson setting, we also have:

**Lemma 6.** *Let $Z_\gamma = \mathcal{P}(\gamma X)$. Then, for every $\gamma > 0$ and $y = 0, 1, \ldots$,*

$$\frac{d}{d\gamma} P_{Z_\gamma | X}(y|x) = x \left( P_{Z_\gamma | X}(y - 1|x) - P_{Z_\gamma | X}(y|x) \right), \quad \gamma \frac{d}{d\gamma} P_{Z_\gamma}(y) = y P_{Z_\gamma}(y) - (y + 1) P_{Z_\gamma}(y + 1)$$

*where $P_{Z_\gamma | X}(-1|x) = P_{Z_\gamma}(-1) = 0$.*

**Proof of Lemma 6.** Let $Z_\gamma = \mathcal{P}(\gamma X)$, where $Z_\gamma$ is a Poisson random variable with parameter $\gamma X$. We first compute the derivative of the conditional probability mass function $P_{Z_\gamma}(z|X = x)$ with respect to $\gamma$.

Since $Z_\gamma$ given $X = x$ follows a Poisson distribution with mean $\gamma x$, we have

$$P_{Z_\gamma}(z|X = x) = \frac{(\gamma x)^y e^{-\gamma x}}{y!}.$$

Taking the derivative with respect to $\gamma$ and using product rule, we obtain:

$$\frac{d}{d\gamma} P_{Z_\gamma}(z|X = x) = \frac{d}{d\gamma} \left( \frac{(\gamma x)^z e^{-\gamma x}}{z!} \right) = \frac{z(\gamma x)^{z-1} x e^{-\gamma x}}{z!} - \frac{x(\gamma x)^z e^{-\gamma x}}{z!}.$$

Simplifying the terms, we obtain

$$\frac{d}{d\gamma} P_{Z_\gamma}(z|X = x) = x \left( \frac{(\gamma x)^{z-1} e^{-\gamma x}}{(z - 1)!} - \frac{(\gamma x)^z e^{-\gamma x}}{z!} \right).$$

Notice that

$$\frac{(\gamma x)^{z-1} e^{-\gamma x}}{(z - 1)!} = P_{Z_\gamma}(z - 1|X = x),$$

we can rewrite the derivative as

$$\frac{d}{d\gamma} P_{Z_\gamma}(z|X = x) = x \left( P_{Z_\gamma}(z - 1|X = x) - P_{Z_\gamma}(z|X = x) \right).$$

This establishes the first part of the lemma.

Next, we compute the derivative of the marginal probability $P_{Z_\gamma}(z)$ with respect to $\gamma$. By the law of total probability, we have

$$P_{Z_\gamma}(y) = E \left[ P_{Z_\gamma}(z|X) \right].$$

Differentiating both sides with respect to $\gamma$, we obtain

$$\frac{d}{d\gamma} P_{Z_\gamma}(z) = E \left[ \frac{d}{d\gamma} P_{Z_\gamma}(z|X) \right].$$

Substituting the result from above, we get

$$\frac{d}{d\gamma} P_{Z_\gamma}(z) = E\left[x\left(P_{Z_\gamma}(z-1|X) - P_{Z_\gamma}(z|X)\right)\right].$$

This can be expressed as

$$\gamma \frac{d}{d\gamma} P_{Z_\gamma}(z) = \gamma E\left[x P_{Z_\gamma}(z-1|X)\right] - \gamma E\left[x P_{Z_\gamma}(z|X)\right].$$

Noting that for a Poisson distribution, $E\left[x P_{Z_\gamma}(z|X)\right] = \frac{z}{\gamma} P_{Z_\gamma}(z)$ and $E\left[x P_{Z_\gamma}(z-1|X)\right] = \frac{z}{\gamma} P_{Z_\gamma}(z)$, we substitute to obtain

$$\gamma \frac{d}{d\gamma} P_{Z_\gamma}(z) = z P_{Z_\gamma}(z) - (z+1) P_{Z_\gamma}(z+1).$$

Thus, the second part of the lemma is established.

**Other properties of the Conditional Expectation**

**Lemma 7.** *Let $Z_\gamma = \mathcal{P}(\gamma X)$ where $X$ is a nonnegative random variable, and $\gamma > 0$. Then, for every $\gamma > 0$ and integer $z \geq 0$,*

$$\frac{d}{d\gamma} E\left[X|Z_\gamma = z\right] = -z\gamma \operatorname{Var}\left(X|Z_\gamma = z-1\right),$$

*where* $\operatorname{Var}\left(X|Z_\gamma = -1\right) = 0$.

*Proof.* Fix an integer $z \geq 0$. Consider the conditional expectation

$$E\left[X|Z_\gamma = z\right] = \frac{1}{\gamma}\left((z+1)\frac{P(Z_\gamma = z+1)}{P(Z_\gamma = z)}\right).$$

Differentiating both sides with respect to $\gamma$, we obtain

$$\frac{d}{d\gamma} E\left[X|Z_\gamma = z\right] = \frac{1}{\gamma}\frac{d}{d\gamma}\left((z+1)\frac{P(Z_\gamma = z+1)}{P(Z_\gamma = z)}\right) - \frac{1}{\gamma^2}\left((z+1)\frac{P(Z_\gamma = z+1)}{P(Z_\gamma = z)}\right).$$

Applying the quotient rule to the derivative inside the parentheses, we get

$$\frac{d}{d\gamma}\left(\frac{P(Z_\gamma = z+1)}{P(Z_\gamma = z)}\right) = \frac{P(Z_\gamma = z)\frac{d}{d\gamma}P(Z_\gamma = z+1) - P(Z_\gamma = z+1)\frac{d}{d\gamma}P(Z_\gamma = z)}{P(Z_\gamma = z)^2}.$$

Using the properties of the Poisson distribution, specifically the identity

$$\frac{P(Z_\gamma = z+1)}{P(Z_\gamma = z)} = \frac{\gamma X}{z+1},$$

we can simplify the derivative expression. Substituting back, we obtain

$$\frac{d}{d\gamma} E\left[X|Z_\gamma = z\right] = -z\gamma \operatorname{Var}\left(X|Z_\gamma = z-1\right).$$

For the case $z = 0$, the derivative simplifies to $\frac{d}{d\gamma} E\left[X|Z_\gamma = 0\right] = 0$, since $\operatorname{Var}\left(X|Z_\gamma = -1\right) = 0$ by definition.

The result for higher moments follows similarly. For any positive integer $k$, differentiating $E\left[(\gamma X)^k|Z_\gamma = z\right]$ with respect to $\gamma$ and applying the quotient rule leads to the stated piecewise expression. This completes the proof.

Moreover, for any positive integer $k$,

$$\frac{d}{d\gamma} E\left[(\gamma X)^k \| Z_\gamma = z\right] = \begin{cases} k E\left[(\gamma X)^{k-1}|Z_\gamma = 0\right], & z = 0, \\ \dfrac{(z+k) E\left[(\gamma X)^{k-1}|Z_\gamma = z\right] E\left[\gamma X|Z_\gamma = z-1\right] - z E\left[(\gamma X)^k|Z_\gamma = z\right]}{E\left[\gamma X|Z_\gamma = z-1\right]}, & z \geq 1. \end{cases}$$

**Lemma 8.** *Let $Z_\gamma \sim \mathcal{P}(\gamma X)$. Then, for every fixed $\gamma > 0$ and any non-degenerate $X$, the mapping $z \mapsto E[X|Z_\gamma = z]$ is strictly increasing.*

*Proof.* To show that $E[X|Z_\gamma = z]$ is strictly increasing, we define $U = \gamma X$ and consider the Poisson marginal probability:

$$P_{Z_\gamma}(k) = \frac{1}{k!} E\left[U^k e^{-U}\right] \tag{30}$$

Applying the Cauchy-Schwarz inequality, we obtain

$$P_{Z_\gamma}(k) \leq \frac{1}{k!} \sqrt{E\left[U^{k+1} e^{-U}\right] E\left[U^{k-1} e^{-U}\right]}. \tag{31}$$

Rewriting in terms of factorial expressions, we get

$$P_{Z_\gamma}(k) \leq \sqrt{\frac{k+1}{k} P_{Z_\gamma}(k+1) P_{Z_\gamma}(k-1)}. \tag{32}$$

Now, substituting this bound into the Turing-Good-Robbins (TGR) formula from Lemma 4:

$$E[U|Z_\gamma = z] = \frac{(z+1) P_{Z_\gamma}(z+1)}{P_{Z_\gamma}(z)}, \tag{33}$$

we obtain the lower bound

$$E[U|Z_\gamma = z] \geq \frac{(z+1) \frac{z}{z+1} P_{Z_\gamma}^2(z)}{P_{Z_\gamma}(z) P_{Z_\gamma}(z-1)}. \tag{34}$$

Simplifying, this reduces to

$$E[U|Z_\gamma = z] \geq \frac{z P_{Z_\gamma}(z)}{P_{Z_\gamma}(z-1)}. \tag{35}$$

Using the same formulation for $z - 1$, we conclude

$$E[U|Z_\gamma = z] \geq E[U|Z_\gamma = z-1]. \tag{36}$$

Since $X = U/\gamma$, it follows that $E[X|Z_\gamma = z]$ is strictly increasing in $z$, completing the proof.

### D.3 Incremental Channel Approach to I-MPRL and related proofs:

Here, we derive interesting relations between the mutual information in a Poisson noise channel and various parameters of the channel. The general distribution we consider here is $Y \sim \text{Poisson}(\alpha X + \lambda)$.

**Theorem 9.** *Let $\lambda > 0$ and let $X$ be a positive random variable satisfying $E\{X \log X\} < \infty$. Consider the Poisson random transformation $X \mapsto Z_\lambda = \mathcal{P}(X + \lambda)$. Then, the derivative of the mutual information between $X$ and $Z_\lambda$ with respect to the dark current $\lambda$ is given by*

$$\frac{d}{d\lambda} I(X; Z_\lambda) = E\left[\log(X + \lambda) - \log\langle X + \lambda \rangle\right],$$

*where $\langle X + \lambda \rangle = E[X + \lambda | Z_\lambda = z]$.*

*Proof:* Let $Y_0 = \mathcal{P}(X)$ and $N_\lambda = \mathcal{P}(\lambda)$ be independent Poisson random variables with means $X$ and $\lambda$, respectively. Define $Y_\lambda = Y_0 + N_\lambda$, which has the same distribution as $\mathcal{P}(X + \lambda)$. By the definition of mutual information,

$$I(X; Y_0) - I(X; Y_\lambda) = E\{L(X, Y_0, Y_\lambda)\},$$

where the expectation is over the joint distribution of $(X, Y_0, Y_\lambda)$, and the log-likelihood ratio is

$$L(x, k, \ell) = \log \frac{P_{Y_0|X}(k|x)}{P_{Y_0}(k)} - \log \frac{P_{Y_\lambda|X}(\ell|x)}{P_{Y_\lambda}(\ell)}.$$

Given that $Y_0|X = x \sim \mathcal{P}(x)$ and $Y_\lambda|X = x \sim \mathcal{P}(x + \lambda)$, the conditional probabilities are

$$P_{Y_0|X}(k|x) = \frac{x^k e^{-x}}{k!}, \quad P_{Y_\lambda|X}(\ell|x) = \frac{(x+\lambda)^\ell e^{-(x+\lambda)}}{\ell!}.$$

Substituting these into the log-likelihood ratio, we obtain

$$L(X, Y_0, Y_\lambda) = Y_0 \log X - Y_\lambda \log(X + \lambda) + U,$$

where $U$ encompasses terms involving the logarithms of the marginal probabilities. Taking the expectation, we have

$$E[L] = E\{X \log X - (X + \lambda) \log(X + \lambda)\} + E[U].$$

Expanding $Y_\lambda = Y_0 + N_\lambda$ and leveraging the independence of $N_\lambda$ from $Y_0$, we analyze the behavior of $E[U]$ as $\lambda$ becomes small. Through a series of manipulations and applying the dominated convergence theorem, we find that

$$I(X; Y_\lambda) - I(X; Y_0) = \lambda\, E\left[\log \frac{X}{\langle X \rangle}\right] + o(\lambda).$$

Dividing both sides by $\lambda$ and taking the limit as $\lambda \to 0$, we obtain

$$\frac{d}{d\lambda} I(X; Y_\lambda) = E\left[\log(X + \lambda) - \log\langle X + \lambda \rangle\right],$$

where $\langle X + \lambda \rangle = E[X + \lambda | Y_\lambda = z]$. This completes the proof of Lemma 9.

**Theorem 10.** *For every Poisson transformation $\mathcal{P}_X$ with $E\{X \log X\} < \infty$, and as $\delta \to 0$,*

$$I\big(X; \mathcal{P}((1 + \delta)X)\big) - I\big(X; \mathcal{P}(X)\big) = \delta\, E\{X \log X - \langle X \rangle \log\langle X \rangle\} + o(\delta).$$

*Proof:* Consider first the case $\delta \to 0^+$. Let $Y = \mathcal{P}(X)$ and $Z = \mathcal{P}(\delta X)$ be independent conditioned on $X$. Define $Y_\delta = Y + Z$. Then, the left-hand side of the lemma can be expressed as

$$I(X; Y_\delta) - I(X; Y) = E\left\{\log \frac{P_{Y_\delta|X}(Y_\delta|X)}{P_{Y_\delta}(Y_\delta)} - \log \frac{P_{Y|X}(Y|X)}{P_Y(Y)}\right\}.$$

Expanding the log-likelihood ratio, we have

$$= E\left\{Z \log X - \delta X - \log \frac{E\{(X')^{Y_\delta} e^{-(1+\delta)X'} | Y_\delta\}}{E\{(X')^Y e^{-X'} | Y\}}\right\}.$$

Here, $X'$ is identically distributed as $X$ but independent of $Y$ and $Z$.

To analyze the expression as $\delta \to 0$, we approximate $\Delta = \mathcal{P}(\delta X)$ by a Bernoulli random variable that takes the value 1 with probability $\delta X$ (conditioned on $X$) and 0 otherwise. This approximation is valid because for small $\delta$, the Poisson distribution $\mathcal{P}(\delta X)$ closely resembles a Bernoulli distribution.

Substituting this approximation into the previous step, we obtain

$$I(X; Y_\delta) - I(X; Y) = E\left\{Z \log X - \delta X - \log\left[(1 - \delta X)\, E\{(X')^Y e^{-X'} | Y\} + \delta X\, E\{(X')^{Y+1} e^{-X'} e^{-\delta X'} | Y\}\right]\right\} + o(\delta) \tag{37}$$

Expanding $e^{-\delta X'}$ to first order in $\delta$, we have $e^{-\delta X'} \approx 1 - \delta X'$. Therefore,

$$E\{(X')^{Y+1} e^{-X'} e^{-\delta X'} | Y\} \approx E\{(X')^{Y+1} e^{-X'} | Y\} - \delta E\{(X')^{Y+2} e^{-X'} | Y\} + o(\delta) \tag{38}$$

Substituting this back into the logarithm and applying the first-order Taylor expansion $\log(1 + \epsilon) \approx \epsilon$ for small $\epsilon$, we obtain

$$\log\left[(1 - \delta X)\, E\{(X')^Y e^{-X'} | Y\} + \delta X\, E\{(X')^{Y+1} e^{-X'} | Y\}\right]$$

$$\approx \log\left[E\{(X')^Y e^{-X'} | Y\}\right] + \frac{\delta X\, E\{(X')^{Y+1} e^{-X'} | Y\} - \delta X\, E\{(X')^Y e^{-X'} | Y\}}{E\{(X')^Y e^{-X'} | Y\}} + o(\delta)$$

$$= \log\langle X \rangle - \delta X \frac{E\{(X')^Y e^{-X'} | Y\} - E\{(X')^{Y+1} e^{-X'} | Y\}}{E\{(X')^Y e^{-X'} | Y\}} + o(\delta),$$

where $\langle X \rangle = E\{X | Y\}$.

Substituting this approximation back into equation 37, we get

$$I(X;Y_\delta) - I(X;Y) = E\left\{Z \log X - \delta X - \log\langle X\rangle + \delta X \frac{E\{(X')^{Y+1}e^{-X'}|Y\}}{E\{(X')^Y e^{-X'}|Y\}}\right\} + o(\delta) \quad (39)$$

Noting that $Z$ is Poisson with parameter $X$, we have $E\{Z|X\} = X$, and thus $E\{Z \log X\} = E\{X \log X\}$.

Furthermore, we know that $\langle X\rangle = E\{X|Y\}$, and from Lemma 4, we have

$$E\left\{(X')^{Y+1}e^{-X'}|Y\right\} = \langle X\rangle e^{-\langle X\rangle}(Y+1).$$

Substituting these into equation 39, we simplify to

$$I(X;Y_\delta) - I(X;Y) = \delta E\{X \log X - \langle X\rangle \log\langle X\rangle\} + o(\delta),$$

Dividing both sides by $\delta$ and taking the limit as $\delta \to 0$, we obtain

$$\left.\frac{d}{d\delta}I(X;Y_\delta)\right|_{\delta=0} = E\left[X \log X - \langle X\rangle \log\langle X\rangle\right],$$

where $\langle X\rangle = E[X|Y]$. This completes the proof of the lemma.

## E  Tail Bounds

As we know the output $z_\gamma$ given the input $x$ is modeled as $z_\gamma \sim \mathcal{P}(\gamma x)$, where $x \geq 0$ is the non-negative input random variable, and $\gamma$ represents the signal-to-noise ratio (SNR). The negative log-likelihood when estimating $z_\gamma$ using $x$, is given by:

$$l(x, z_\gamma) = -\log p(z_\gamma|x) = -\log\left(\frac{e^{-\gamma x}(\gamma x)^{z_\gamma}}{z_\gamma!}\right) = \gamma x - z_\gamma \log(\gamma x) + \log z_\gamma!$$

We define the expected negative log-likelihood as $M(\gamma) = E_{(x,z_\gamma)}\left[l(x, z_\gamma)\right] = E_x\left[E_{(z_\gamma|x)}\left[l(x, z_\gamma)\right]\right]$. We now consider a mean constraint $\mu = E[x]$ in this case and our objective then is to determine the input distribution $p_X(x)$ over $x \geq 0$ that maximizes the above function. To compute the expected loss, let us first evaluate $E_{z_\gamma|x}[l(x, z_\gamma)]$ and using $E_{z_\gamma|x}[z_\gamma] = \gamma x$ gives:

$$E_{z_\gamma|x}[l(x, z_\gamma)] = E_{z_\gamma|x}\left[\gamma x - z_\gamma \log(\gamma x) + \log z_\gamma!\right] = \quad (40)$$

$$\gamma x - \log(\gamma x) \cdot E_{z_\gamma|x}[z_\gamma] + E_{z_\gamma|x}[\log z_\gamma!] = \gamma x - \gamma x \log(\gamma x) + E_{z_\gamma|x}[\log z_\gamma!] \quad (41)$$

We can write $M(\gamma)$ in terms of the the conditional entropy of $z_\gamma$ given $x$ as:

$$M(\gamma) = E_x[H(z_\gamma|x)], \text{ since } H(z_\gamma|x) = E_{z_\gamma|x}\left[-\log p(z_\gamma|x)\right] = E_{z_\gamma|x}[l(x, z_\gamma)].$$

The entropy $H(z_\gamma|x)$ of a Poisson distribution with parameter $\gamma x$ is given by:

$$HS(\gamma x) = -\sum_{k=0}^{\infty} P(z_\gamma = k) \log P(z_\gamma = k)$$

where $P(z_\gamma = k) = \frac{(\gamma x)^k e^{-\gamma x}}{k!}$. So substituting this into the entropy expression, we obtain:

$$HS(\gamma x) = -\sum_{k=0}^{\infty} \frac{(\gamma x)^k e^{-\gamma x}}{k!} \log\left(\frac{(\gamma x)^k e^{-\gamma x}}{k!}\right) = \gamma x - \gamma x \log(\gamma x) + \sum_{k=0}^{\infty} \frac{(\gamma x)^k e^{-\gamma x}}{k!} \log k!$$

It is natural to assume that the Shannon entropy $HS(\lambda)$ of a Poisson distribution strictly increases with $\lambda \in (0, +\infty)$. We will prove this result, as well as the concavity property of $HS(\lambda)$, in the following lemma.

**Lemma 11.** *The Shannon entropy $HS(\lambda)$, $\lambda \in (0, +\infty)$, is strictly increasing and concave in $\lambda$.*

*Proof.* The Shannon entropy $HS(\lambda)$ of a Poisson distribution is as outlined above. To analyze the monotonicity and concavity of $HS(\lambda)$, we compute its first and second derivatives with respect to $\lambda$.

First, the first derivative $HS'(\lambda)$ is:

$$H'_S(\lambda) = -\log\left(\frac{\lambda}{e}\right) - 1 - e^{-\lambda} \sum_{k=2}^{\infty} \frac{\lambda^k \log k!}{k!} + e^{-\lambda} \sum_{k=2}^{\infty} \frac{\lambda^{k-1} \log k!}{(k-1)!} \tag{42}$$

$$= -\log\lambda + e^{-\lambda} \sum_{k=1}^{\infty} \frac{\lambda^k \log(k+1)!}{k!} - e^{-\lambda} \sum_{k=2}^{\infty} \frac{\lambda^k \log k!}{k!} \tag{43}$$

Simplifying, we get:

$$HS'(\lambda) = -\log\lambda + e^{-\lambda} \sum_{k=1}^{\infty} \frac{\lambda^k}{k!} \log(k+1)$$

It is clear that both terms on the right-hand side of (2) are non-negative for $\lambda \in (0,1]$, and the second term is strictly positive. Therefore, $H'_S(\lambda) > 0$ for $\lambda \in (0,1]$. Now, it remains to prove that $H'_S(\lambda) > 0$ for $\lambda > 1$. Let's calculate:

$$H''_S(\lambda) = -\frac{1}{\lambda} - e^{-\lambda} \sum_{k=1}^{\infty} \frac{\lambda^k \log(k+1)}{k!} + e^{-\lambda} \sum_{k=1}^{\infty} \frac{\lambda^{k-1} \log(k+1)}{(k-1)!}$$

$$= -\frac{1}{\lambda} + e^{-\lambda} \sum_{k=0}^{\infty} \frac{\lambda^k \log(k+2)}{k!} - e^{-\lambda} \sum_{k=1}^{\infty} \frac{\lambda^k \log(k+1)}{k!}$$

$$= -\frac{1}{\lambda} + e^{-\lambda} \log 2 + e^{-\lambda} \sum_{k=1}^{\infty} \frac{\lambda^k \log\left(1 + \frac{1}{k+1}\right)}{k!}$$

$$= -\frac{1}{\lambda} + e^{-\lambda} \sum_{k=0}^{\infty} \frac{\lambda^k \log\left(1 + \frac{1}{k+1}\right)}{k!}$$

$$< -\frac{1}{\lambda} + e^{-\lambda} \sum_{k=0}^{\infty} \frac{\lambda^k}{(k+1)!} < -\frac{1}{\lambda} + e^{-\lambda} \frac{1}{\lambda} \sum_{k=0}^{\infty} \frac{\lambda^{k+1}}{(k+1)!}$$

$$< -\frac{1}{\lambda} + e^{-\lambda} \frac{1}{\lambda} e^{\lambda} = 0.$$

So, $H''_S(\lambda) < 0$ for all $\lambda > 0$. Therefore, $H'_S(\lambda)$ strictly decreases in $\lambda$, proving **concavity** and it is sufficient to prove that $\lim_{\lambda \to \infty} H'_S(\lambda) \geq 0$ After further simplification,

$$\lim_{\lambda \to \infty} H'_S(\lambda) = \lim_{\lambda \to \infty} \log\lambda \left(e^{-\lambda}(\log\lambda)^{-1} \sum_{k=1}^{\infty} \frac{\lambda^k \log(k+1)}{k!} - 1\right),$$

and it is sufficient to establish that

$$\liminf_{\lambda \to \infty} e^{-\lambda}(\log\lambda)^{-1} \sum_{k=1}^{\infty} \frac{\lambda^k \log(k+1)}{k!} \geq 1.$$

This inequality is outlined in [62]. Using this, we get that $H'_S(\lambda) > 0$ for all $\lambda \geq 0$ and $H''_S(\lambda) < 0$ for all $\lambda \geq 0$, hence the proof follows.

Given that $H(z_\gamma|x)$ is an increasing and concave function of $x$ for $x > 0$, we aim to maximize $E_x[H(z_\gamma|x)]$ under the mean constraint $E[x] = \mu$. The functional to maximize is $J[p_X(x)] = \int_0^\infty H(z_\gamma|x)p_X(x)\,dx$, subject to the normalization and mean constraints: $\int_0^\infty p_X(x)\,dx = 1$ and $\int_0^\infty x p_X(x)\,dx = \mu$

Introducing Lagrange multipliers $\lambda$ and $\nu$ for these constraints, the Lagrangian becomes:

$$\mathcal{L}[p_X(x)] = \int_0^\infty H(z_\gamma|x)p_X(x)\,dx - \lambda\left(\int_0^\infty p_X(x)\,dx - 1\right) - \nu\left(\int_0^\infty xp_X(x)\,dx - \mu\right)$$

Taking the functional derivative of $\mathcal{L}$ with respect to $p_X(x)$ and setting it to zero for optimality yields: $\frac{\delta\mathcal{L}}{\delta p_X(x)} = H(z_\gamma|x) - \lambda - \nu x = 0$

Given the properties of $H(z_\gamma|x)$, the solution corresponds to an exponential distribution. The exponential distribution with mean $\mu$ is given by:

$$p_X(x) = \frac{1}{\mu}e^{-x/\mu}, \quad x \geq 0$$

Maximizing the entropy of $x$ leads to a distribution that spreads the probability mass, thereby increasing uncertainty and consequently maximizing the mprl. Now, using this exponential prior, we will derive an expression for $\mathrm{mprl}(\gamma)$ which we use for deriving the left and right tail bounds.

Now, the prior distribution for $X$ is assumed to be an exponential distribution:

$$f_X(x) = \lambda e^{-\lambda x}$$

We introduce the latent variable $Z_\gamma$ such that:

$$P(Z_\gamma = z|X = x) = \frac{e^{-\gamma x}(\gamma x)^z}{z!}$$

which follows a Poisson distribution. The conditional density of $X$ given $Z_\gamma = z$ is derived as:

$$f_{X|Z}(x|z) = \frac{P(Z_\gamma = z|X = x)f_X(x)}{P(Z_\gamma = z)}$$

$$f_{X|Z}(x|z) = \frac{(\beta x)^z}{z!}\lambda e^{-\lambda x}e^{-\beta x}$$

$$= \frac{(\beta x)^z \lambda e^{-(\lambda+\beta)x}}{z!P(Z_\gamma = z)}$$

and we can notice that this is a Gamma distribution: $X|Z_\gamma = z \sim \mathrm{Gamma}(z + 1, \lambda + \beta)$ The posterior mean of $X$ given $Z_\gamma$ is:

$$E[X|Z_\gamma = z] = \frac{z + 1}{\lambda + \beta} \tag{44}$$

and this serves as the optimal estimate $\hat{X}^*$. Now, let us consider the following expectation: (where $l$ is the previously defined Poisson loss function)

$$E_{X|Z_\gamma}[l(X, X^*)] = E[X\log\left(\frac{X}{X^*}\right) - X + X^*] = E\left[X\log\left(\frac{X}{X^*}\right)\Big|Z_\gamma\right] - E[X|Z_\gamma] + X^* \tag{45}$$

Using integration by parts and properties of the Gamma function, if $W \sim \mathrm{Gamma}(\alpha, \beta)$, then: [63]

$$E[W\log W] = \frac{\alpha}{\beta}\left[\psi(\alpha + 1) - \log\beta\right]$$

where we defined the **digamma function** $\psi(\alpha)$ as: $\psi(\alpha) = \frac{d}{d\alpha}\log\Gamma(\alpha)$. The above results would also follow from differentiating the moment formula:

$$E[X^n] = \frac{\Gamma(\alpha + n)}{\Gamma(\alpha)\beta^n}$$

Applying this this result in our case gives us:

$$E[X \log X | Z_\gamma] = \frac{z+1}{\lambda + \beta} \left[ \psi(z+2) - \log(\lambda + \alpha) \right]$$

We also have from Equation. 44:

$$\log(X^*) = \log(z+1) - \log(\lambda + \alpha)$$

Taking expectation, the first term in Eq. 45 can be written as:

$$E\left[ X \log\left( \frac{X}{X^*} \right) \Big| Z \right] = \frac{z+1}{\lambda + \beta} \left[ \psi(z+2) - \log(\lambda + \alpha) \right] - \frac{z+1}{\lambda + \alpha} \left[ \log(z+1) - \log(\lambda + \alpha) \right]$$

(46)

$$= \frac{z+1}{\lambda + \beta} \left[ \psi(z+2) - \log(z+1) \right]$$

(47)

Now, we compute the marginal distribution as follows:

$$P(Z_\gamma = z) = \int_0^\infty P(Z_\gamma = z | X = x) f_X(x) \, dx = \frac{\lambda \beta^z}{z!} \int_0^\infty x^z e^{-(\lambda+\beta)x} dx.$$

Using the Gamma integral property stated as follows:

$$\int_0^\infty x^z e^{-(\lambda+\beta)x} dx = \frac{\Gamma(z+1)}{(\lambda+\beta)^{z+1}},$$

we obtain (since $\Gamma(z+1) = z!$):

$$P(Z_\gamma = z) = \frac{\lambda \beta^z}{z!} \cdot \frac{\Gamma(z+1)}{(\lambda+\beta)^{z+1}} = \frac{\lambda \beta^z}{(\lambda+\beta)^{z+1}} = (1-p)p^z, \text{ where } p = \frac{\beta}{\lambda+\beta}$$

Now, the $\mathrm{mprl}(\gamma)$ expression obtained is as follows:

$$\mathrm{mprl}(\gamma) = \sum_{z=0}^\infty (1-p)p^z \left[ \frac{z+1}{\lambda+\beta} \left[ \psi(z+2) - \log(z+1) \right] \right] = \frac{\lambda}{(\lambda+\beta)^2} \sum_{z=0}^\infty (z+1)p^z \left[ \psi(z+2) - \log(z+1) \right].$$

### E.1 Left Tail Bound

In case of $(\gamma_0, \gamma_1)$ being the relevant range of integration, the left tail integral is defined as: $\int_0^{\gamma_0} \mathrm{mprl}(\gamma) \, d\gamma$

First, we interchange the sum and the integral:

$$\int_0^{\gamma_0} \mathrm{mprl}(\gamma) \, d\gamma = \sum_{z=0}^\infty (z+1) \left[ \psi(z+2) - \log(z+1) \right] \int_0^{\gamma_0} \frac{\lambda}{(\lambda+\gamma)^2} \left( \frac{\gamma}{\lambda+\gamma} \right)^z d\gamma.$$

We define the inner integral as

$$I_z = \int_0^{\gamma_0} \frac{\lambda}{(\lambda+\gamma)^2} \left( \frac{\gamma}{\lambda+\gamma} \right)^z d\gamma.$$

Substitute $u = \lambda + \gamma$, which implies $\gamma = u - \lambda$ and $d\gamma = du$. The bounds change accordingly: $u = \lambda$ when $\gamma = 0$ and $u = \lambda + \gamma_0$ when $\gamma = \gamma_0$. The integral becomes

$$I_z = \lambda \int_\lambda^{\lambda+\gamma_0} \frac{(u-\lambda)^z}{u^{z+2}} \, du.$$

Next, using the substitution $v = \frac{u-\lambda}{u}$, leading to $u = \frac{\lambda}{1-v}$ and $du = \frac{\lambda}{(1-v)^2} \, dv$. The bounds transform to $v = 0$ when $u = \lambda$ and $v = \frac{\gamma_0}{\lambda+\gamma_0}$ when $u = \lambda + \gamma_0$. Substituting these into the integral yields

$$I_z = \int_0^{\frac{\gamma_0}{\lambda+\gamma_0}} v^z \, dv.$$

The integral $I_z$ can be evaluated as

$$I_z = \left[ \frac{v^{z+1}}{z+1} \right]_0^{\frac{\gamma_0}{\lambda+\gamma_0}} = \frac{\left( \frac{\gamma_0}{\lambda+\gamma_0} \right)^{z+1}}{z+1}.$$

Substituting $I_z$ back into the expression for the expectation, gives:

$$\int_0^{\gamma_0} \mathrm{mprl}(\gamma)\, d\gamma = \sum_{z=0}^{\infty} \left[ \psi(z+2) - \log(z+1) \right] \left( \frac{\gamma_0}{\lambda+\gamma_0} \right)^{z+1}$$

Let the above sum be $S$ which we use in the sections below. By re-indexing the sum with $k = z+1$, the final result can more elegantly be expressed as:

$$\int_0^{\gamma_0} \mathrm{mprl}(\gamma)\, d\gamma = \sum_{k=1}^{\infty} \left[ \psi(k+1) - \log(k) \right] \left( \frac{\gamma_0}{\lambda+\gamma_0} \right)^k.$$

We aim to establish an upper bound for the sum

$$S = \sum_{z=0}^{\infty} (z+1) \left[ \psi(z+2) - \log(z+1) \right] \left( \frac{\gamma_0}{\lambda+\gamma_0} \right)^{z+1},$$

where $\psi$ denotes the digamma function, $\gamma_0 > 0$, and $\lambda > 0$.

Let us define $x = \frac{\gamma_0}{\lambda+\gamma_0}$. Given that $\gamma_0 > 0$ and $\lambda > 0$, it follows that $0 < x < 1$. From, [64], we recall the expansion of the digamma function:

$$\psi(z+2) = H_{z+1} - \gamma_E,$$

where $H_n$ is the $n$-th harmonic number and $\gamma_E$ is the Euler-Mascheroni constant. For large $z$,

$$H_{z+1} = \log(z+1) + \gamma_E + \frac{1}{2(z+1)} - \frac{1}{12(z+1)^2} + \cdots.$$

Substituting this into the expression for $\psi(z+2)$ yields:

$$\psi(z+2) - \log(z+1) = \frac{1}{2(z+1)} - \frac{1}{12(z+1)^2} + \cdots.$$

From this expansion, it is evident that

$$\psi(z+2) - \log(z+1) < \frac{1}{2(z+1)}$$

for all $z \geq 0$, since the higher-order terms $-\frac{1}{12(z+1)^2} + \cdots$ contribute negatively, thereby decreasing the overall value.

Consequently, each term in the sum satisfies

$$(z+1) \left[ \psi(z+2) - \log(z+1) \right] x^{z+1} < \frac{1}{2} x^{z+1}.$$

Summing over $z$ from 0 to $\infty$, we obtain

$$S < \frac{1}{2} \sum_{z=0}^{\infty} x^{z+1}.$$

Using the simplification of the geometric series $\sum_{z=0}^{\infty} x^{z+1}$

$$\sum_{z=0}^{\infty} x^{z+1} = \frac{x}{1-x} \implies S < \frac{1}{2} \frac{x}{1-x}.$$

Substituting back $x = \frac{\gamma_0}{\lambda+\gamma_0}$, we have

$$1 - x = 1 - \frac{\gamma_0}{\lambda+\gamma_0} = \frac{\lambda}{\lambda+\gamma_0} \implies \frac{x}{1-x} = \frac{\frac{\gamma_0}{\lambda+\gamma_0}}{\frac{\lambda}{\lambda+\gamma_0}} = \frac{\gamma_0}{\lambda}.$$

Putting this into the inequality for $S$, we obtain

$$S < \frac{1}{2}\frac{\gamma_0}{\lambda}.$$

Hence, the upper bound for the sum in the scalar case (for a single input-output realization) is

$$\boxed{\sum_{z=0}^{\infty}(z+1)\left[\psi(z+2)-\log(z+1)\right]\left(\frac{\gamma_0}{\lambda+\gamma_0}\right)^{z+1}\leq\frac{\gamma_0}{2\lambda}.}$$

(**Note**: This $z$ is different from the $z_\gamma$ notation used throughout the paper.)

Extending this result to the vector case, consider a $d$-dimensional random vector $x \in X \subset \mathbb{Z}^d$ with covariance matrix $\Sigma$, whose eigenvalues are $\{\lambda_i\}_{i=1}^{d}$, all positive. Assuming the problem is separable across the eigenbasis of $\Sigma$, each dimension can be treated independently.

For the vector case, the sum becomes

$$S_{\text{vector}}=\sum_{i=1}^{d}\sum_{z=0}^{\infty}(z+1)\left[\psi(z+2)-\log(z+1)\right]\left(\frac{\gamma_0}{\lambda_i+\gamma_0}\right)^{z+1}.$$

Applying the scalar bound to each eigenvalue $\lambda_i$, we have

$$\sum_{z=0}^{\infty}(z+1)\left[\psi(z+2)-\log(z+1)\right]\left(\frac{\gamma_0}{\lambda_i+\gamma_0}\right)^{z+1}\leq\frac{\gamma_0}{2\lambda_i}.$$

Summing over all $i$ from 1 to $d$, the vector sum satisfies

$$S_{\text{vector}}\leq\sum_{i=1}^{d}\frac{\gamma_0}{2\lambda_i}=\frac{\gamma_0}{2}\sum_{i=1}^{d}\frac{1}{\lambda_i}.$$

In the special case where the covariance matrix $\Sigma$ is isotropic, meaning all eigenvalues $\lambda_i = \lambda$ for $i = 1, \ldots, d$, the bound simplifies to

$$S_{\text{vector}}\leq\frac{d\gamma_0}{2\lambda}.$$

This concludes the derivation of the left tail bounds for both the scalar and vector cases.

## E.2 Right Tail Bound

In case of $(\gamma_0, \gamma_1)$ being the relevant range of integration, the right tail integral is defined as: $\int_{\gamma_1}^{\infty}\mathrm{mprl}(\gamma)\,d\gamma$

Consider a discrete variable $x = (x_1, x_2, \ldots, x_d) \in X \subset \mathbb{Z}^d$, where each component $x_i$ belongs to a discrete set $\{i\,\Delta\,|\,i \in \mathbb{Z}\}$. Observations are modeled as $z_{\gamma,i} \sim \mathcal{P}(\gamma x_i)$ for a large signal-to-noise ratio (SNR) parameter $\gamma$. The estimator $\hat{x}_i(z_{\gamma,i})$ is typically the maximum likelihood estimator (MLE), implemented by rounding $z_{\gamma,i}$ to the nearest bin $\{k\,\Delta\}$.

The loss function per component is defined as

$$L(x_i, \hat{x}_i)=x_i\log\left(\frac{x_i}{\hat{x}_i}\right)-x_i+\hat{x}_i,$$

and the $\mathrm{mprl}(\gamma)$ is given by $\mathbb{E}[L(x_i, \hat{x}_i)]$ over the randomness of $z_{\gamma,i}$. The right-tail integral of interest is

$$I_R=\int_{\gamma_1}^{\infty}E\left[\sum_{i=1}^{d}L(x_i, \hat{x}_i(z_{\gamma,i}))\right]d\gamma,$$

which we aim to upper bound.

At high SNR ($\gamma \to \infty$), the noise is relatively small compared to $x_i$, but rare rounding errors of size $j\Delta$ can still occur. Focusing on a single component $x_i$, an error of size $j\Delta$ happens if

$$\hat{x}_i=x_i-j\Delta \quad\Longleftrightarrow\quad z_{\gamma,i}\in[\gamma(x_i-j\Delta-0.5\Delta),\gamma(x_i-j\Delta+0.5\Delta)).$$

For $z_{\gamma,i} \sim \text{Poisson}(\mu)$ with $\mu = \gamma x_i$, the Poisson Chernoff bound [65] provides that the probability of such a deviation is at most $\exp(-c_{i,j}\gamma)$, where $c_{i,j} > 0$ is a constant dependent on $\Delta$, $x_i$, and the shift $j\Delta$. Hence,

$$P(\text{error of size } j\Delta) \le e^{-c_{i,j}\gamma}.$$

The per-component contribution to the mean MLE loss is

$$\text{mprl}_i(\gamma) = E_{z_{\gamma,i}}\left[L(x_i, \hat{x}_i(z_{\gamma,i}))\right].$$

When the estimation error is $j\Delta$, the loss becomes

$$L(x_i, x_i - j\Delta) = x_i \log\left(\frac{x_i}{x_i - j\Delta}\right) - x_i + (x_i - j\Delta).$$

Therefore, the mean loss satisfies

$$\text{mprl}_i(\gamma) \le \sum_{j=1}^{j_{\max}} \left[x_i \log\left(\frac{x_i}{x_i - j\Delta}\right) - x_i + (x_i - j\Delta)\right] e^{-c_{i,j}\gamma}.$$

Summing over all components $i = 1, \ldots, d$, we obtain

$$\text{mprl}(\gamma) = \sum_{i=1}^{d} \text{mprl}_i(\gamma) \le \sum_{i=1}^{d}\sum_{j=1}^{j_{\max}} \left[x_i \log\left(\frac{x_i}{x_i - j\Delta}\right) - x_i + (x_i - j\Delta)\right] e^{-c_{i,j}\gamma}.$$

The right-tail integral $I_R$ can thus be bounded as

$$I_R = \int_{\gamma_1}^{\infty} \text{mprl}(\gamma)\, d\gamma \le \sum_{i=1}^{d}\sum_{j=1}^{j_{\max}} \left[x_i \log\left(\frac{x_i}{x_i - j\Delta}\right) - x_i + (x_i - j\Delta)\right] \int_{\gamma_1}^{\infty} e^{-c_{i,j}\gamma} d\gamma.$$

Evaluating the integral, we find

$$\int_{\gamma_1}^{\infty} e^{-c_{i,j}\gamma} d\gamma = \frac{e^{-c_{i,j}\gamma_1}}{c_{i,j}},$$

Leading to the final right-tail bound

$$\boxed{I_R = \int_{\gamma_1}^{\infty} E\left[\sum_{i=1}^{d} L(x_i, \hat{x}_i)\right] d\gamma \le \sum_{i=1}^{d}\sum_{j=1}^{j_{\max}} \left[x_i \log\left(\frac{x_i}{x_i - j\Delta}\right) - j\Delta\right] \frac{e^{-c_{i,j}\gamma_1}}{c_{i,j}}.}$$

In the above expression, $c_{i,j} > 0$ represents the Chernoff-type exponent from the Poisson large-deviation bound for the event causing an error of size $j\Delta$ in component $i$. We determine these parameters empirically, and the parameter $j_{\max}$ indicates the largest error shift considered, which is typically small in practice and can be tuned empirically. For empirical purposes, it might also be worthwhile to note that the bracketed term in Eq. 47 can be approximated as the sum over a few starting $z$ beyond which it effectively dies out as illustrated in Figure 14.

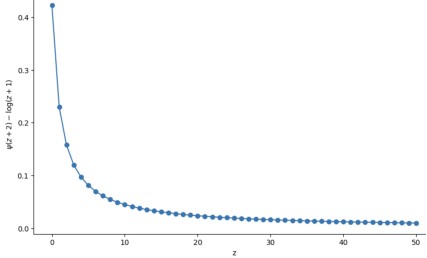

Figure 14: Approximating the Digamma term

# F    Proof Sketch of Pointwise Poisson Denoising Relation

For Poisson channel defined earlier, we derive the pointwise denoising relation:

**Theorem 12.** *The KL divergence derivative satisfies:*

$$\frac{d}{d\gamma} D_{KL}[P(z_\gamma|x)\|P(z_\gamma)] = \mathrm{mprl}(x, \gamma)$$

*where the pointwise MPRL is:*

$$\mathrm{mprl}(x, \gamma) \equiv E_{P(z_\gamma|x)}\left[l\left(x, \hat{x}^*(z_\gamma)\right)\right]$$

*with $l(x, x^*) = x \log \frac{x}{x^*} - x + x^*$ and $\hat{x}^*(z_\gamma) = E[X|z_\gamma]$.*

*Proof.* For the Poisson channel $Z_\gamma|X = x \sim \mathrm{Pois}(\gamma x)$ define

$$R_x(\gamma) := \sum_{z \geq 0} p_\gamma(z|x) \log \frac{p_\gamma(z|x)}{p_\gamma(z)}, \ \ \hat{x}^*(z, \gamma) := E[X|Z_\gamma = z], \ \ell(x, \hat{x}) := x \log \frac{x}{\hat{x}} - x + \hat{x}.$$

Differentiate the series using the product rule to get

$$\frac{d}{d\gamma} R_x = \sum_z \partial_\gamma p_\gamma(z|x) \log \frac{p_\gamma(z|x)}{p_\gamma(z)} + \sum_z p_\gamma(z|x)\left(\partial_\gamma \log p_\gamma(z|x) - \partial_\gamma \log p_\gamma(z)\right) = T_1 + T_2.$$

**Term $T_2$:**    For the Poisson distribution with mean $\gamma x$,

$$p_\gamma(z|x) = e^{-\gamma x} \frac{(\gamma x)^z}{z!}.$$

Taking the derivative,

$$\frac{d}{d\gamma} \log p_\gamma(z|x) = -x + \frac{z}{\gamma}.$$

Hence, for the conditional Poisson law, we have

$$\partial_\gamma \log p_\gamma(z|x) = \frac{z}{\gamma} - x.$$

Similarly for the marginal,

$$p_\gamma(z) = \int p_\gamma(z|x)P(x)dx.$$

Taking the log-derivative,

$$\frac{d}{d\gamma} \log p_\gamma(z) = \frac{1}{p_\gamma(z)} \int (-x + \frac{z}{\gamma})p_\gamma(z|x)p(x)dx.$$

Identifying this as a conditional expectation gives:

$$\partial_\gamma \log p_\gamma(z) = E\left[-X + \frac{Z}{\gamma} \ \middle| \ Z = z\right].$$

Hence

$$T_2 = E_{p_\gamma(z|x)}\left[E[X|Z] - \frac{Z}{\gamma}\right].$$

**Term $T_1$:**    Let $r(z) := \log \frac{p_\gamma(z|x)}{p_\gamma(z)}$ and $\lambda := \gamma x$. Since $\partial_\gamma p_\gamma(z|x) = p_\gamma(z|x)\left(\frac{z}{\gamma} - x\right)$,

$$T_1 = E_{p_\gamma(z|x)}\left[\left(\frac{Z_\gamma}{\gamma} - x\right)r(Z_\gamma)\right] = \frac{1}{\gamma}E\left[(Z_\gamma - \lambda)\, r(Z_\gamma)\right].$$

Now, we state and prove what we call the "Poisson-Stein" identity.

**Lemma 13** (Poisson–Stein identity). *For $Z_\gamma \sim \text{Pois}(\lambda)$, we have:*

$$E\big[(Z_\gamma - \lambda)h(Z_\gamma)\big] = \lambda\, E\big[h(Z_\gamma + 1) - h(Z_\gamma)\big].$$

*Proof sketch.* Let $p_\lambda(k) = \mathbb{P}\{Z_\gamma = k\} = e^{-\lambda}\lambda^k/k!$ for $k \geq 0$. Then, for any $h$ such that $\mathbb{E}|(Z_\gamma - \lambda)h(Z_\gamma)| < \infty$,

$$\mathbb{E}[(Z_\gamma - \lambda)h(Z_\gamma)] = \sum_{k=0}^{\infty}(k - \lambda)h(k)\, p_\lambda(k)$$

$$= \sum_{k=1}^{\infty} k\, h(k)\, p_\lambda(k) - \lambda \sum_{k=0}^{\infty} h(k)\, p_\lambda(k).$$

Using $k\, p_\lambda(k) = \lambda\, p_\lambda(k-1)$ for $k \geq 1$ and reindexing,

$$\sum_{k=1}^{\infty} k\, h(k)\, p_\lambda(k) = \lambda \sum_{k=1}^{\infty} h(k)\, p_\lambda(k-1) = \lambda \sum_{j=0}^{\infty} h(j+1)\, p_\lambda(j),$$

so that

$$\mathbb{E}[(Z_\gamma - \lambda)h(Z_\gamma)] = \lambda \sum_{j=0}^{\infty} \big(h(j+1) - h(j)\big)p_\lambda(j) = \lambda\, \mathbb{E}[h(Z_\gamma + 1) - h(Z_\gamma)].$$

This discrete integration-by-parts argument establishes the Poisson–Stein identity.

Now with $h = r$ in Lemma 13, we have

$$T_1 = \frac{\lambda}{\gamma} E\big[r(Z_\gamma + 1) - r(Z_\gamma)\big] = x\, E\big[r(Z_\gamma + 1) - r(Z_\gamma)\big].$$

Using the ratio formulas (from Lemma 4) gives:

$$\frac{p_\gamma(z+1|x)}{p_\gamma(z|x)} = \frac{\gamma x}{z+1}, \quad \frac{p_\gamma(z+1)}{p_\gamma(z)} = \frac{\gamma\, \langle X\rangle_z}{z+1}, \quad \langle X\rangle_z := E[X|Z_\gamma = z],$$

we get

$$r(z+1) - r(z) = \log \frac{x}{\langle X\rangle_z}, \quad T_1 = x\, E\left[\log \frac{x}{\langle X\rangle_{Z_\gamma}}\right].$$

Now, combining both the terms gives:

$$\frac{d}{d\gamma}R_x = x\, E\left[\log \frac{x}{\langle X\rangle_{Z_\gamma}}\right] + E\left[\langle X\rangle_{Z_\gamma} - \frac{Z_\gamma}{\gamma}\right].$$

Since $E[Z_\gamma|X = x] = \gamma x$, the second expectation equals $E[\langle X\rangle_{Z_\gamma}] - x$, and hence

$$\frac{d}{d\gamma}R_x = E_{p_\gamma(z|x)}\left[x \log \frac{x}{\langle X\rangle_{Z_\gamma}} - x + \langle X\rangle_{Z_\gamma}\right] = E_{p_\gamma(z|x)}\big[\ell(x, \hat{x}^*)\big].$$

$\square$

This equation can also be derived as a special case of Lemma 4.2 from [29].

**Link to the MPRL Loss:** We already defined the loss function:

$$\ell\big(x, \hat{x}^*\big) = x \log \frac{x}{\hat{x}^*} - x + \hat{x}^*.$$

If $\hat{x}^* \equiv E[X|z_\gamma]$ is the estimator of $x$ given $z_\gamma$, then by standard properties of conditional expectation,

$$E_{P(z_\gamma|x)}[\hat{x}^*] = E[E[X|z_\gamma]] = E[X] = x \quad \text{(if $x$ is deterministic, replace $E[X]$ by $x$).}$$

Hence,

$$E_{P(z_\gamma|x)}[\ell(x, \hat{x}^*)] = E[x \log x - x \log \hat{x}^* - x + \hat{x}^*] = x \log x - x - xE[\log \hat{x}^*] + E[\hat{x}^*].$$

Since $E[\hat{x}^*] = x$,

$$E_{P(z_\gamma|x)}[\ell(x, \hat{x}^*)] = x(\log x - E[\log \hat{x}^*]).$$

One can show (by comparing with the final expression in the KL derivative) that this expectation aligns with $E_{P(z_\gamma|x)}[E[X|z_\gamma] - \frac{z_\gamma}{\gamma}]$, thus establishing the link between the MPRL and the derivative of the KL divergence. We can generalize this relation to any loss function that belongs to the class of Bregman divergences in a Poisson channel using the framework described in [37].

### F.1 Tweedie's for Poisson Denoising

A well-known result in Gaussian denoising is *Tweedie's Formula*, which expresses the conditional expectation of the latent variable in terms of the derivative of the log-pdf of noisy observation. [32]. Specifically, for $Z_\gamma = \sqrt{\gamma}X + \varepsilon$ with $\varepsilon \sim \mathcal{N}(0, I)$, we have:

$$E[X|Z_\gamma = z] = \frac{z}{\sqrt{\gamma}} + \frac{1}{\gamma}\nabla \log f_{Z_\gamma}(z), \tag{48}$$

In the Poisson setting, we cannot directly take derivatives of $\log P_{Z_\gamma}(z)$ with respect to discrete $z$ since they are undefined. Instead, the *forward difference* of the log of the marginal PMF serves as a discrete analog. This culminates in the Turing-Good-Robbins (TGR) formula, already presented in Lemma 4.

Hence, just like Tweedie's Formula in the continuous Gaussian case, TGR expresses the conditional mean $\langle X \rangle_z$ purely in terms of the marginal distribution $P_{Z_\gamma}(z)$, bypassing any need to compute the conditional distribution $P_{X|Z_\gamma}$. In effect, the ratio $\gamma \cdot \langle X \rangle_z$ plays the role of a *score function* for the Poisson channel, analogous to the logarithmic derivative in the Gaussian case. This discrete variant underpins our Poisson diffusion framework, allowing us to efficiently compute the optimal denoiser $E[X|Z_\gamma]$ directly from the marginal PMF.

## G Continuous Extension of ItDPDM

We extend the continuous-time channel with discrete states (CTDS) to continuous states through the following construction:

**Definition 14** (Continuous-Time Channel with States (CTCS))**.** Let $\{X_t\}_{t \geq 0}$ be a right-continuous state process with left limits (càdlàg) taking values in $\mathbb{R}_+$. The output process $\{Y_t\}_{t \geq 0}$ is a counting process satisfying:

$$Y_t = \mathcal{P}\left(\int_0^t X_s ds\right) \tag{49}$$

where $\mathcal{P}(\cdot)$ denotes a Poisson counting measure.

For measurable intensity $X_t$, the output increments also satisfy:

$$Y_{t+\delta} - Y_t \sim \mathcal{P}\left(\int_t^{t+\delta} X_s ds\right), \quad \forall t, \delta \geq 0 \tag{50}$$

with $\{Y_{t_k} - Y_{t_{k-1}}\}_{k=1}^n$ independent given $X_{[0,T]}$ for any finite partition $\{t_k\}$.

The mutual information between state and observation processes over $[0, T]$ is given by:

$$I(X^T; Y^T) = E\left[\log \frac{dP_{Y^T|X^T}}{dP_{Y^T}}\right] \tag{51}$$

The key connection to discrete-time systems emerges through infinitesimal discretization:

**Lemma 15** (Mutual Information Rate)**.** *For the CTCS in Definition 14, the mutual information rate satisfies:*

$$\lim_{T \to \infty} \frac{1}{T} I(X^T; Y^T) = \lim_{\delta \to 0} \frac{1}{\delta} I(X_\delta; Y_\delta) \tag{52}$$

*where $X_\delta := X_{[0,\delta)}$ and $Y_\delta := Y_\delta - Y_0$ corresponds to the discrete-time channel $\mathcal{P}(\delta X)$.*

*Proof Sketch.* Consider time partitions $0 = t_0 < t_1 < \cdots < t_n = T$ with $\max |t_{k+1} - t_k| \leq \delta$. By the chain rule of mutual information:

$$I(X^T; Y^T) = \sum_{k=0}^{n-1} I(X^{t_{k+1}}; Y_{t_{k+1}} | Y^{t_k})$$

$$= \sum_{k=0}^{n-1} \left[ I(X_{[t_k, t_{k+1})}; Y_{[t_k, t_{k+1})}) + \epsilon_k \right]$$

where $\epsilon_k$ captures residual dependence between time intervals. Using the Markov property of Poisson counters [66] and taking $\delta \to 0$, the residual terms vanish by the Asymptotic Equipartition Property (AEP) for Poisson processes [67]. The result follows from Lemma 9 applied to each infinitesimal interval.

The continuous-time counterpart of the derivative relationship becomes:

**Lemma 16** (Information Rate Derivative). *For the CTCS system, the time derivative of mutual information satisfies:*

$$\frac{d}{dt}I(X^t; Y^t) = E\left[X_t \log X_t - \langle X_t \rangle \log\langle X_t \rangle\right] \tag{53}$$

*where $\langle X_t \rangle := E[X_t | Y^t]$ is the causal MPRL estimator.*

*Proof.* From Lemma 15 and the DTCS derivative, we have:

$$\frac{d}{dt}I(X^t; Y^t) = \lim_{\delta \to 0} \frac{1}{\delta}\left[I(X_{t+\delta}; Y_{t+\delta}|Y^t) - I(X_t; Y_t)\right]$$
$$= \lim_{\delta \to 0} \frac{1}{\delta}E\left[\delta X_t \log X_t - \delta\langle X_t \rangle \log\langle X_t \rangle\right] + o(1)$$

The result follows by dominated convergence and the tower property of conditional expectation. This continuous-time formulation preserves the essential duality between information and estimation seen in discrete time, with the Poisson channel's inherent noise characteristics governing both regimes. The CTCS framework enables analysis of real-time filtering and prediction [68] through differential versions of the key discrete-time identities.

## H  Detailed comparison of ItDPDM vs. Learning to Jump (LTJ, [20])

Table 10 shows a detailed comparison below:

In the Learning-to-Jump (LTJ) framework [20], the per-step training loss is written as $D_\phi(x, f_\theta(z_t, t))$, where

- $x \in \mathbb{N}$ is the true discrete count.
- $z_t$ is the noisy observation at step $t$, obtained by binomial thinning of $z_{t-1}$.
- $f_\theta(z_t, t)$ is the denoising network (parameterized by $\theta$), which takes $(z_t, t)$ and outputs an estimate $\hat{x}_t$ of $x$.
- $D_\phi(u, v)$ is the Bregman divergence induced by a convex generator $\phi$: $D_\phi(u, v) = \phi(u) - \phi(v) - \langle\nabla\phi(v), u - v\rangle$. For the Poisson channel one uses $\phi(u) = u \log u$, yielding $D_\phi(x, \hat{x}) = \hat{x}\log\frac{\hat{x}}{x} - \hat{x} + x$, i.e. the Poisson–Bregman (relative-entropy) loss.

## I  Noised and Denoised Image Comparison

Figure 19 presents a comparison of noisy and denoised images under Gaussian and Poisson noise conditions at a logSNR of 4.01. The left column displays the input images corrupted by Gaussian (Figure 17) and Poisson noise (Figure 15), while the right column shows the corresponding denoised outputs (Figures 16 and Figures 16). Notably, the Poisson noise case exhibits a higher level of degradation than the Gaussian noise case, making recovery more challenging. However, the denoising process effectively reconstructs meaningful image structures in both cases, demonstrating the model's robustness to varying noise distributions.

## J  Theoretical Runtime Analysis of ItDPDM Architecture

We present a theoretical runtime analysis of the proposed *Information-Theoretic Discrete Poisson Diffusion Model* (ItDPDM), focusing on the core components contributing to its computational cost during training and inference.

**Poisson Noise Sampling**

Table 10: Side-by-side comparison of our **ItDPDM** vs the **Learning-to-Jump** (LTJ) framework. We note that both methods employ a Poisson-Bregman (relative-entropy) loss—denoted PRL for ItDPDM and $D_\phi$ for LTJ but they diverge sharply in how that loss is used and how it connects to likelihood, as summarised below.

| Aspect | ItDPDM (ours) | Learning-to-Jump (LTJ) [20] |
|---|---|---|
| **Forward "noising"** | Single-shot **Poisson channel** $Z_\gamma \sim$ Pois$(\gamma X)$ with *continuous* SNR $\gamma \in (0, \infty)$ | **Binomial thinning chain** $z_t \sim$ Binomial$(z_{t-1}, \alpha_t/\alpha_{t-1})$ for $t = 1, \ldots, T$ |
| **Reverse / generation** | sampling operates in log-SNR space via a continuous-time reverse SDE or ODE; sampling can flexibly subsample the SNR continuum (e.g. 20–50 steps) without quality loss, in contrast to fixed-step chains | 'Count-thickening' Markov chain with shifted-Poisson jumps; sampling requires executing all $T$ discrete steps with no flexibility to skip or subsample, so the full $T$-step chain is incurred for every generated sample |
| **Bounds on NLL** | *Information-theoretic*, extends the classic I-MMSE identity to the Poisson channel, giving the **exact** relation: $-\log p(x) = \int_0^\infty$MPRL$(x, \gamma)\, d\gamma$ | *Variational ELBO*, multi-term KL-divergence sum with binomial/Poisson factors; yields only an **approximate** bound on $-\log p(x)$ |
| **Training Loss** | **PRL**: $\ell(x, \hat{x}) = \hat{x} \cdot \log(\hat{x}/x) - \hat{x} + x$, integrated over *continuous* $\gamma$, producing an **exact** NLL upper bound and provides analytic tail bounds & an importance-sampling estimator; empirically yields lower NLL than all baselines. | Per-step relative-entropy $D_\phi(x, f_\theta(z_t, t))$ inside an ELBO with an identical Bregman form, *but* summed over discrete $T$ only with **no closed-form** link between the total loss and the true likelihood. |
| **Scheduling** | Choose only a continuous SNR grid (e.g., 1000-point logistic); no $\alpha_t$ or $T$ hyper-parameters. | Must hand-design thinning schedule $\{\alpha_t\}_{t=1}^T$ and pick $T$ (typically $T = 1000$). |
| **Likelihood evaluation** | Exact tail bounds + importance sampling; likelihood (**NLL**) (in bits-per-dim) on real-world data, both WD & NLL on synthetic data evaluated | Likelihood *not* estimated; evaluation solely via Wasserstein distance (WD) of histograms. |
| **Sampling speed** | Compatible with fast ODE solvers (20–50 steps) due to continuous $\gamma$. | Must run all $T$ thickening steps. |
| **Theoretical extensions** | Poisson-Tweedie identity; mutual-information derivative; CTCS extension. | — |

The forward diffusion process in ItDPDM is governed by a Poisson noise channel $z_\gamma \sim \text{Poisson}(\gamma x)$, where $x \in \mathbb{R}_+^D$ denotes the input data vector and $\gamma$ is the signal-to-noise ratio (SNR). Sampling from a Poisson distribution can be performed in $\mathcal{O}(1)$ per element using rejection sampling or table-based methods, resulting in a total cost of $\mathcal{O}(D)$ per data point.

**Neural Denoising**

The denoiser is instantiated as a neural network, such as a U-Net (for images) or a Transformer encoder (for symbolic music). The input to the denoiser is the reparameterized form

$$\tilde{z}_\gamma = \frac{z_\gamma}{1 + \gamma},$$

which improves numerical stability. The forward pass of the denoiser has cost $\mathcal{O}(D)$ per data point, assuming conventional convolutional or attention-based layers.

**Poisson Loss Function Evaluation**

The proposed loss function is based on a Bregman divergence tailored to Poisson noise:

$$\ell(x, \hat{x}) = x \log \left( \frac{x}{\hat{x}} \right) - x + \hat{x},$$

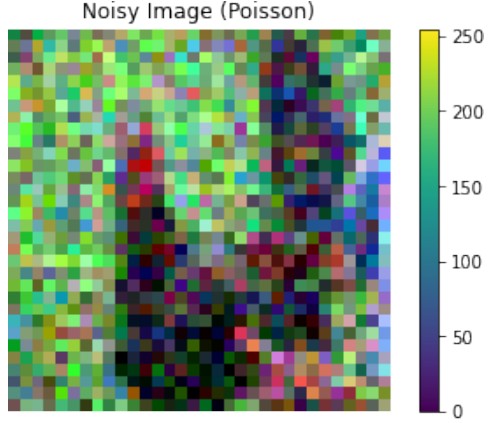

Figure 15: Noisy Image (Poisson Noise)

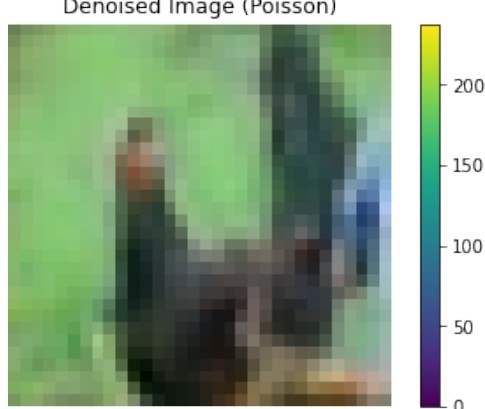

Figure 16: Denoised Image (Poisson Noise)

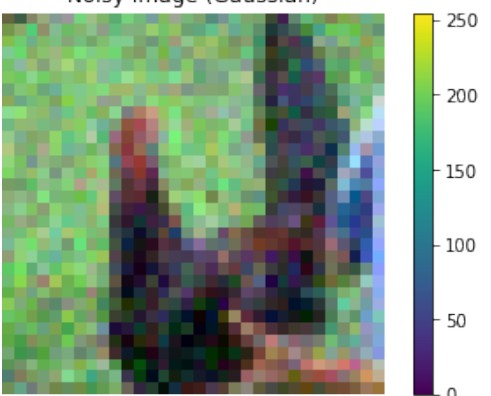

Figure 17: Noisy Image (Gaussian Noise)

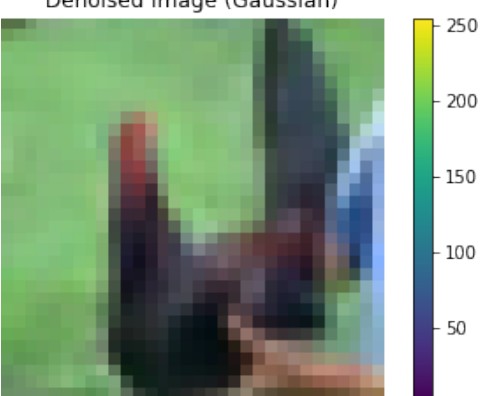

Figure 18: Denoised Image (Gaussian Noise)

Figure 19: Comparison of noisy and denoised images for poisoned and Gaussian noise birds with logsnr=4.01.

which is convex, differentiable, and evaluated pointwise. The cost of loss evaluation and gradient computation is $\mathcal{O}(D)$ per sample.

**Integral Estimation over SNR**

A defining component of the ItDPDM framework is the estimation of the negative log-likelihood using thermodynamic integration:

$$-\log P(x) = \int_0^\infty \mathrm{mprl}(x, \gamma)\, d\gamma,$$

where MPRL denotes the minimum mean likelihood error. In practice, this integral is approximated numerically using $n$ log-SNR values (e.g., $n = 1000$), obtained via uniform or importance sampling over $\alpha = \log \gamma$.

Each SNR point requires a forward pass through the denoiser and loss computation, yielding a total per-sample complexity of $\mathcal{O}(n \cdot D)$. To reduce overhead, the model uses importance sampling from a truncated logistic distribution over $\alpha$ and closed-form tail integral bounds to truncate the SNR domain (see Eqs. (28)–(29) in the main text).

Given a batch size $B$ and number of training epochs $E$, the overall training complexity becomes: $\mathcal{O}(B \cdot E \cdot n \cdot D)$. This is comparable to standard continuous-state diffusion models using discretized time steps, but the Poisson-specific formulation and MPRL integral introduce unique architectural and optimization challenges that are efficiently addressed via reparameterization and sampling strategies. Additionally, in terms of wall-clock times for training/sampling, we observe that ItDPDM is comparable to standard DDPM-style models.

| Component | Complexity | Description |
|---|---|---|
| Poisson noise sampling | $\mathcal{O}(D)$ | Efficient per-sample noise generation |
| Neural denoising | $\mathcal{O}(D)$ | Forward pass through CNN or Transformer |
| Poisson loss function | $\mathcal{O}(D)$ | Evaluated pointwise for each data coordinate |
| Integral over SNR | $\mathcal{O}(n \cdot D)$ | Dominant cost due to repeated inference and loss evaluations |
| **Total per-sample cost** | $\mathcal{O}(n \cdot D)$ | For fixed number of SNR grid points |

Table 11: Asymptotic complexity of key components in the ItDPDM training pipeline.

# K   Extended Related Work

Diffusion models have evolved along two orthogonal dimensions—*noise type* and *state space*. Classical DDPMs corrupt continuous data with additive Gaussian noise and learn the reverse process with score matching or variational bounds [2, 38, 49–51]. An information-theoretic viewpoint links these objectives to mutual–information integrals [12, 22], and has recently motivated non-Gaussian extensions based on annealed score matching [52, 53] and SDE formalisms [54]. Parallel work seeks native *discrete-state* alternatives: masking schemes such as Blackout Diffusion employ an irreversible "black" token that blocks exact likelihood computation [55]; Learning-to-Jump (LTJ) replaces Gaussian noise by binomial thinning/thickening yet remains limited to discrete time and a variational ELBO [20]. Very recent approaches move to continuous-time jump processes, but still approximate the likelihood: [69] devise a categorical SDE whose reverse dynamics are learned by discrete score matching, while [57] estimate probability *ratios* rather than scores to reduce perplexity on text.

**Score Entropy Discrete Diffusion (SEDD)** [57] represents a significant advancement in discrete diffusion modeling. It introduces the *Score Entropy* loss, a novel objective that extends score matching to discrete spaces by directly modeling the ratios of data probabilities. This approach addresses the challenges of applying traditional score matching to discrete data and enables the construction of discrete diffusion models that are both theoretically sound and empirically effective. SEDD demonstrates competitive performance with autoregressive models like GPT-2 on standard language modeling benchmarks. Notably, it achieves comparable zero-shot perplexities and offers advantages in generation quality and efficiency. For instance, SEDD can generate high-quality text samples with 4× lower generative perplexity when matching function evaluations and requires 16× fewer function evaluations to match the generative perplexity of standard autoregressive sampling methods. Moreover, SEDD enables arbitrary infilling beyond standard left-to-right prompting, matching the quality of nucleus sampling without the need for specialized training or sampling techniques.

Concurrently, several non-Gaussian *continuous* diffusion models have been proposed to address the limitations of traditional Gaussian-based approaches, particularly in handling data with bounded support or preserving structural details in images.

**Beta Diffusion** [70] introduces a novel generative modeling method that integrates demasking and denoising to generate data within bounded ranges. Utilizing scaled and shifted beta distributions, it employs multiplicative transitions over time to create both forward and reverse diffusion processes. This approach maintains beta distributions in both the forward marginals and the reverse conditionals, given the data at any point in time. Unlike traditional diffusion models relying on additive Gaussian noise and reweighted evidence lower bounds (ELBOs), Beta Diffusion is multiplicative and optimized with KL-divergence upper bounds (KLUBs) derived from the convexity of the KL divergence. Experimental results demonstrate its unique capabilities in generative modeling of range-bounded data and validate the effectiveness of KLUBs in optimizing diffusion models.

**Blurring Diffusion Models** [71] propose a generalized class of diffusion models that offer the best of both standard Gaussian denoising diffusion and inverse heat dissipation. By defining blurring through a Gaussian diffusion process with non-isotropic noise, this approach bridges the gap between inverse heat dissipation and denoising diffusion. It sheds light on the inductive bias resulting from this modeling choice and demonstrates the capability to better learn the low-to-mid frequencies within datasets, which plays a crucial role in representing shapes and structural information.

**Edge-Preserving Noise** [72] for diffusion introduces a content-aware diffusion model explicitly trained to learn the non-isotropic edge information in a dataset. Inspired by anisotropic diffusion in image processing, this model incorporates an edge-aware noise scheduler that varies between edge-preserving and isotropic Gaussian noise. The generative process converges faster to results that more closely match the target distribution and better learns the low-to-mid frequencies within the dataset, crucial for representing shapes and structural information. This edge-preserving diffusion process consistently outperforms state-of-the-art baselines in unconditional image generation and is particularly robust for generative tasks guided by a shape-based prior, such as stroke-to-image generation

While these models offer significant advancements in handling specific data characteristics, they still require dequantization and rely on surrogate objectives. In contrast, **ItDPDM** models corruption with a *reversible Poisson channel*, maintaining a discrete latent space, supporting bidirectional perturbations, and—via the I-MPRL identity—transforming the Minimum Poisson Reconstruction Loss into an *exact* likelihood integral instead of a bound. This unifies the tractability of information-theoretic Gaussian diffusion with the fidelity of discrete-state models, yielding closed-form NLLs, scalable continuous-time sampling, and strong empirical performance on sparse, skewed, and over-dispersed count data

**ItDPDM** differs fundamentally from the above lines. By modelling corruption with a *reversible Poisson channel*, ItDPDM keeps the latent space discrete, supports bidirectional perturbations, and—via the I-MPRL identity—turns the Minimum Poisson Reconstruction Loss into an *exact* likelihood integral instead of a bound. This unifies the tractability of information-theoretic Gaussian diffusion with the fidelity of discrete-state models, yielding closed-form NLLs, scalable continuous-time sampling, and strong empirical performance on sparse, skewed, and over-dispersed count data.

## NeurIPS Paper Checklist

The checklist is designed to encourage best practices for responsible machine learning research, addressing issues of reproducibility, transparency, research ethics, and societal impact. Do not remove the checklist: **The papers not including the checklist will be desk rejected.** The checklist should follow the references and follow the (optional) supplemental material. The checklist does NOT count towards the page limit.

Please read the checklist guidelines carefully for information on how to answer these questions. For each question in the checklist:

- You should answer [Yes] , [No] , or [NA] .
- [NA] means either that the question is Not Applicable for that particular paper or the relevant information is Not Available.
- Please provide a short (1–2 sentence) justification right after your answer (even for NA).

**The checklist answers are an integral part of your paper submission.** They are visible to the reviewers, area chairs, senior area chairs, and ethics reviewers. You will be asked to also include it (after eventual revisions) with the final version of your paper, and its final version will be published with the paper.

The reviewers of your paper will be asked to use the checklist as one of the factors in their evaluation. While "[Yes] " is generally preferable to "[No] ", it is perfectly acceptable to answer "[No] " provided a proper justification is given (e.g., "error bars are not reported because it would be too computationally expensive" or "we were unable to find the license for the dataset we used"). In general, answering "[No] " or "[NA] " is not grounds for rejection. While the questions are phrased in a binary way, we acknowledge that the true answer is often more nuanced, so please just use your best judgment and write a justification to elaborate. All supporting evidence can appear either in the main paper or the supplemental material, provided in appendix. If you answer [Yes] to a question, in the justification please point to the section(s) where related material for the question can be found.

IMPORTANT, please:

- **Keep the checklist subsection headings, questions/answers and guidelines below.**

- **Do not modify the questions and only use the provided macros for your answers**.

