# OpenReview forum: "ItDPDM: Information-Theoretic Discrete Poisson Diffusion Model"
_NeurIPS.cc/2025/Conference — NeurIPS 2025 poster_

### Official Review · Reviewer_Ptu5 · 2025-06-27

**Clarity:** 1
**Significance:** 2
**Originality:** 2
**Rating:** 4
**Confidence:** 3

**Summary:**

This paper presents ItDPDM, a diffusion-like model that combines exact likelihood estimation with discrete-state modeling. To this end, the paper also introduces poisson reconstruction loss (PRL). Some theoretical properties of this loss are then shown. The proposed ItDPDM is evaluated on both synthetic and real-world data, with a particular focus to compare against LTJ.

**Questions:**

Please see the weakness section.

**Ethical Concerns:**

["NO or VERY MINOR ethics concerns only"]

**Final Justification:**

The authors' revised argument makes sense to me, and the experiments show some merit in the model. 2D synthetic experiments, though insufficient to me, are okay, given that there are other real-data experiments to back up. Although there might be a need to revamp the current draft (for both the contents and proofs), I lean towards the value of the contribution this paper brings.

**Limitations:**

yes

**Paper Formatting Concerns:**

No formatting concerns.

**Quality:**

1

**Strengths And Weaknesses:**

Strengths:

1. A new diffusion-like algorithm that enables exact likelihood estimation

2. A new Poisson-based loss with some theoretical properties


Weaknesses:

1. Wrong proof of Equation (12): This equation plays a key role for exact likelihood calculation (e.g., Eq. (15)). However, in the proof of this equation (Lemma 12 in Appendix F), the differentiation is directly moved inside the expectation operator w.r.t. p(z_\gamma|x) (line 1546 in the supplementary). I don't think it's valid because the conditional measure p(z_\gamma|x) depends on \gamma. (Please see the correct way to handle it in [1, Appendix A]). This calls the rest of the likelihood estimation results into question.

2. Missing proof: In the proof of Lemma 2, I only see the "only if" part, but I couldn't find the "if" part. However, the main text seems to be highlighting the "if" part (see the sentence above Lemma 2), which lacks a solid proof.

3. Weak numerics: In Table 1, among the six distributions chosen for likelihood evaluation, LTJ beats ItDPDM only on three. (Also note here that the boldfaces are very misleading, as they are not highlighting the best values, as they are supposed to be.) This calls into question whether the proposed method is advantageous numerically for likelihood evaluation. Plus, the generation quality of ItDPDM is not as good as shown in Table 3.

4. Simplified univariate setting: Both the sampler and the loss are defined only for the univariate distributions. While this is as expected since Poisson distributions are primarily designed for count data. Nevertheless, it calls into question whether both of them are useful for multi-dimensional data. Also, it is not clear how the model and loss are evaluated for image data such as CIFAR-10.

5. Unclarities and typos: I list the following for consideration, which indeed hindered my reading. in Lemma 4, P_{Z_\gamma} as the marginal of z is not clearly specified. In Eq. (12), \calP should just be P. In Algorithm 1, line 4 is inconsistent with Figure 4, line 6 contains a q(\alpha) which is not defined, and it is not clear what data_transform function is in line 5. In Algorithm 2, \hat{x}_t should be \hat{x}_0, and why is [12] cited here?

[1] Kong et al., Information-theoretic diffusion, ICLR 23.

---

> ### Author Rebuttal · Authors · 2025-07-30
>
> We thank the reviewer for their thorough feedback and valuable suggestions. We have addressed the reviewer’s concerns below:
>
> **Weaknesses**:
>
> **1." Wrong proof of Equation (12): This equation plays a key role for exact likelihood........"**
>
> We sincerely thank the reviewer for pointing this out and bringing it to our attention. We now have an updated proof-sketch stated below, which will also be included in more detail in the revised version:
>
> **Setup.**
> Fix $x\ge0,\ \gamma>0,\ \lambda=\gamma x$.
> With $Z_\gamma\mid X=x\sim\mathrm{Pois}(\lambda)$ define
>
> $$
> R_x(\gamma)=\sum_{z\ge0} p_\gamma(z|x)\
> \log\frac{p_\gamma(z\mid x)}{p_\gamma(z)} .
> $$
>
> We can split this using the product-rule for derivatives:
>
> $$
> \frac{\mathrm d}{\mathrm d\gamma} R_x = T_1 + T_2,
> $$
> where
> $$
> T_1 = \sum_{z}\partial_\gamma p_\gamma(z|x)
> \log \frac{p_\gamma(z|x)}{p_\gamma(z)},\text{ }
> T_2 = \sum_{z}p_\gamma(z|x)\
> \bigl(\partial_\gamma\log p_\gamma(z|x)-\partial_\gamma\log p_\gamma(z)\bigr).
> $$
>
> $\partial_\gamma\log p_\gamma(z|x)=z/\gamma-x$.
>
> Taking the required conditional expectation drives this term to zero.
>
> For the marginal, we have:
>
> $$
> \partial_\gamma\log p_\gamma(z)=
> \mathbb{E}\left[-X+\frac{Z_\gamma}{\gamma}|\Big| Z_\gamma=z\right].
> $$
>
> Hence:
>
> $$T_2=\mathbb{E}\bigl[\mathbb{E}[X|Z_\gamma]-Z_\gamma/\gamma\bigr]$$
>
> Now, let $r(z)=\log\frac{p_\gamma(z\mid x)}{p_\gamma(z)}$.
> We use the Poisson-Stein identity:
>
> $$
> \mathbb{E}\bigl[(Z_\gamma-\lambda)h(Z_\gamma)\bigr]
> =\lambda\mathbb{E}\bigl[h(Z_\gamma+1)-h(Z_\gamma)\bigr]
> $$
>
> with $h=r$, this gives
>
> $$
> T_1=\frac{1}{\gamma}\mathbb{E}[(Z_\gamma-\lambda)r(Z_\gamma)]
>       =x\\mathbb{E}[r(Z_\gamma+1)-r(Z_\gamma)].
> $$
>
> We now have these ratios:
> $\frac{p_\gamma(z+1\mid x)}{p_\gamma(z\mid x)}=\frac{\gamma x}{z+1}$ and
> $\frac{p_\gamma(z+1)}{p_\gamma(z)}=\frac{\gamma\\langle X\rangle_{z}}{z+1}$
> with $\langle X\rangle_{z}:=\mathbb{E}[X|Z_\gamma=z]$ (from Lemma 4) yield
>
> $$
> r(z+1)-r(z)=\log\frac{x}{\langle X\rangle_{z}},
> \text{ and }
> T_1=x\\mathbb{E}\\left[\log\frac{x}{\langle X\rangle_{Z}}\right].
> $$
>
> Combine these two terms now, to get:
>
> $$
> \frac{d}{d\gamma}R_x =
> x\\mathbb{E}\\left[\log\frac{x}{\langle X\rangle_{Z}}\right]
> +\mathbb{E}\\left[\langle X\rangle_{Z}-\frac{Z_\gamma}{\gamma}\right].
> $$
>
> Since $\mathbb{E}[Z_\gamma\mid X=x]=\gamma x$, the second expectation equals
> $\mathbb{E}[\langle X\rangle_{Z}]-x$.  Therefore:
>
> $$
> \frac{\mathrm d}{\mathrm d\gamma} R_x
> =\mathbb{E}\\bigl[x\log\frac{x}{\langle X\rangle_{Z}}-x+\langle X\rangle_{Z}\bigr]
> =\mathbb{E}\\bigl[\ell(x,\widehat x^{*})\bigr],
> $$
>
> where $\widehat x^{*}=\langle X\rangle_{Z}$ and
> $\ell(u,v)=u\log(u/v)-u+v$.
> This is Eq. (12).
>
>
>
> **2. "Missing proof: In the proof of Lemma 2, I only see the "only if"......."**
>
> **On Lemma 2 ("if" part)**: We are once again grateful to the reviewer for catching this. The appendix currently proves the "only-if'" direction (linear conditional mean $\Rightarrow$ Gamma prior). We will add the converse, which follows directly from Poisson--Gamma conjugacy: if $X \sim \mathrm{Gamma}(\alpha, \beta)$ and $Z_\gamma \mid X \sim \mathrm{Pois}(\gamma X)$, then $X \mid Z_\gamma = z \sim \mathrm{Gamma}(\alpha + z, \beta + \gamma)$, hence $\mathbb{E}[X \mid Z_\gamma = z] = (\alpha + z)/(\beta + \gamma) = az + b$ with $a = 1/(\beta + \gamma)$, $b = \alpha/(\beta + \gamma)$. This completes both the "**if**" and "**only if**" directions of the proof.
>
> **3. "Weak numerics: In Table 1, among the six distributions chosen......"**
>
> **Note**: We sincerely request the reviewer to have another glance at Table 1.
>
> There is a column for *“true NLL”* and boldfaces here reflect whichever baseline yields the closest estimate to the ground-truth NLL, thereby becoming the “best values”. We acknowledge that this may not have been immediately clear, and we will revise the caption of this table to clarify our intent.
>
> On this metric, ItDPDM outperforms LTJ on **4 out of 6** distributions. The remaining two cases (BNB, NBinomMix) have an implicit bias towards LTJ due to the binomial thinning process that underlies LTJ’s noising scheme. Overall this emphasizes the numerical advantage of ItDPDM: it consistently yields likelihood estimates that more closely align with the true NLL across a diverse set of discrete distributions.
>
> **Regarding generation quality**: This is apparent currently due to CIFAR10's 32x32 resolution: we also refer the reviewer to Appendix Fig. 19 for comparing a denoised CIFAR-10 sample from DDPM and ItDPDM. In the context of symbolic music, Appendix Fig. 10 shows that the pitch histogram from ItDPDM-generated symbolic music closely matches the ground truth distribution. Additionally, Fig. 9 (App. B.5.1) includes piano roll visualizations for unconditionally generated samples. We will include: 1) more image generation examples for reference, including some high-resolution images and 2) generative fidelity metrics for a high-resolution image dataset (like *CelebA*) in the final version for further strengthening our work.
>
> **4. "Simplified univariate setting: Both the sampler and the loss........"**
>
> **Experiments for multivariate setting**: We acknowledge that the results presented in the paper are restricted to a univariate setting for simplicity. To further demonstrate ItDPDM’s ability to model multivariate discrete data, we extended our synthetic experiments to a bivariate (2D) joint distribution. First example: we define the joint distribution as a product of two independent univariate marginals using the same parameters from the 1D synthetic experiments. We report the WD metrics in this case as follows: (with the same training settings as in Appendix C.2.)
>
> | Distribution | DDPM           | ItDPDM         | LTJ           |
> |--------------|----------------|----------------|----------------|
> | PoissMix     | 7.90 ± 0.64    | **2.65 ± 0.23**     | 2.96 ± 0.29     |
> | ZIP          | 4.89 ± 0.72    | **1.78 ± 0.27**   | 1.92 ± 0.38     |
> | NBinomMix    | 10.83 ± 0.88    | 3.53 ± 0.57     | **3.26 ± 0.48**      |
> | BNB          | 5.07 ± 0.72    | 1.82 ± 0.36     | **1.67 ± 0.41**      |
> | Zipf         | 3.51 ± 0.58    | **0.97 ± 0.21**      | 1.49 ± 0.32     |
> | YS           | 0.87 ± 0.23    |**0.32 ± 0.05**     | 0.36 ± 0.08     |
>
> Secondly, in addition to the above simple multivariate setting, we also consider two dependent 2D count distributions:
> * **Bivariate Poisson:** Let $U\sim\mathrm{Pois}(\lambda_0)$, $V_1\sim\mathrm{Pois}(\lambda_1)$, $V_2\sim\mathrm{Pois}(\lambda_2)$ be independent; define $X_1=U+V_1$, $X_2=U+V_2$. Marginals: $X_1\sim\mathrm{Pois}(\lambda_0+\lambda_1)$, $X_2\sim\mathrm{Pois}(\lambda_0+\lambda_2)$. Setting $\lambda_0 = 3, \lambda_1 = 2, \lambda_2= 2$, we draw $U,V_1,V_2$, and thereby $(X_1,X_2)$.
>
> * **Gamma–Poisson:** Let $\Theta\sim\mathrm{Gamma}(k = 2,\beta = 1)$; given $\Theta$, we draw $X_i\mid\Theta\sim\mathrm{Pois}(\Theta)$ independently for $i=1,2$.
>
> In these two settings, the evaluated WD scores are:
> | Distribution | DDPM           | ItDPDM         | LTJ           |
> |--------------|----------------|----------------|----------------|
> | Bivariate Poisson  | 3.502| **0.826**     | 1.323     |
> | Gamma–Poisson  | 0.855    | **0.337**   | 0.624     |
>
> **Theoretical extension to multi-variate settings**: Additionally, we would like to call attention to [2] from ISIT’13, which generalizes the I-MMLE identities for vector poisson channels; this would imply that all the results of this paper naturally extend to a multi-variate setting. (along with the updated proofs stated above)
>
> We will incorporate these extensions in the appendix to emphasize the natural extension and benefits of ItDPDM to multivariate settings. Model/loss are evaluated *pixel-by-pixel* for our proposed method and all the baselines: we will also add this note in the final version. We also refer the reviewer to the pseudocode in Algorithm 1, step 6 for clarification regarding this.
>
> **5. "Unclarities and typos: I list the following for consideration,..."**
>
> Thank you for pointing this out, we understand that this could have caused confusion. Line 132 states that in Lemma 4, we express the optimal estimator in terms of the marginal, which P_{Z_\gamma} was supposed to capture. In Eq. (12), we acknowledge that this is a typo and we will replace \calP with P.
>
> Regarding Algorithm 1, line 4: we thank you for pointing out this typo, we will replace it with the correct form from Figure 4:
>
> $z_\gamma \sim \frac{\text{Poisson}(\gamma x)}{1+\gamma}$
>
> We hope this clarifies that it is no longer inconsistent with figure 4. For clarification regarding Algorithm 1, line 6: ($q$… is importance sampling) we refer the reviewer to Appendix Sec. A. 2. We will also mention this in the main paper for final version. Details regarding the data transform function can be found in Appendix Sec. B. 2. We also acknowledge the typo in Algo 2, line 5, and we will change this to $\hat{x}_0$ for the final version. And, finally, we understand that there is no substantial reason to cite [12] here, and will omit it from the pseudocode in the final version.
>
> We sincerely thank the reviewer again for their thoughtful and critical feedback. We believe the points mentioned above: both theoretical and empirical, meaningfully strengthen the paper and address your concerns.
>
> **References**:
>
> [1] Xianghao Kong, Rob Brekelmans, and Greg Ver Steeg. Information‑Theoretic Diffusion. ICLR 2023.
>
> [2] Liming Wang, Miguel Rodrigues, and Lawrence Carin. *Generalized Bregman Divergence and Gradient of Mutual Information for Vector Poisson Channels*. IEEE International Symposium on Information Theory ISIT 2013.

---

> > ### Comment · Reviewer_Ptu5 · 2025-08-04
> >
> > Thank the authors for providing some clarifications. Some of my confusions are cleared.
> >
> > * I couldn't follow why $T_1=\frac{1}{\gamma}\mathbb{E}[(Z_\gamma-\lambda)r(Z_\gamma)]$.
> > * While the 2D experiments (esp. the dependent one) are good to see, I still feel that the scenarios are too simple.
> > * Still, in the paper, it is not clear to me how the model and loss are evaluated for image data such as CIFAR-10. I would presume that [2] (as cited above) is used here.
> >
> > That said, I will raise my score.

---

> ### Author Response · Authors · 2025-08-05
> **Further Clarification**
>
> **We sincerely thank the reviewer Ptu5 for raising their score. Additionally, we appreciate the reviewer’s additional questions and hope they found the earlier clarifications helpful.**
>
> > “I couldn't follow why $T_1=\frac{1}{\gamma}\mathbb{E}[(Z_\gamma-\lambda)r(Z_\gamma)]$.”
>
> As $p_{\gamma}(z|x) = e^{-\gamma x}\dfrac{(\gamma x)^{z_\gamma}}{z_\gamma!}$, then the derivative w\.r.t. $\gamma$ can be written compactly via a log-derivative:
>
> $$
> \partial_{\gamma} p_{\gamma}(z|x)
> = p_{\gamma}(z|x)(\frac{z_\gamma}{\gamma}-x)
> $$
>
> Therefore, $T_1$ can be expressed as:
>
> $T_1 = \sum_z p_{\gamma}(z|x)(\frac{z_\gamma}{\gamma}-x)
> \log \frac{p_\gamma(z \mid x)}{p_\gamma(z)}$
>
> which implies: (with $\lambda = \gamma x$)
>
> $T_1 = \frac{1}{\gamma}\mathbb{E}[(Z_\gamma - \lambda)r(Z_\gamma)]$
>
> where we defined above $r(z) = \log \frac{p_\gamma(z|x)}{p_\gamma(z)}$
>
> Eq. (12) can interestingly also be derived using a limit-based approach as in the Incremental Channel proofs in App. D.3 or a similar approach as above starting from Theorem 4.2 in [1].
>
> We will incorporate the detailed version of the proof sketch for Eq. (12) in the final version.
>
> > “While the 2D experiments (esp. the dependent one) are good to see, I still feel that the scenarios are too simple.”
>
> For image‐like data, most baselines adopt a *factorised* likelihood: each pixel’s channel count is treated as an independent Poisson (or Gaussian) while spatial correlation is captured by the convolutional backbone. In that sense, moving from 1-D to a 2-D grid does **not** require a new analytical noise model; it only changes the dimensionality of the tensor the UNet processes. Our implementation follows this standard practice: PRL is summed pixel-wise, and the network learns cross-pixel structure exactly as in existing diffusion models.
>
> Additionally, to illustrate that the Poisson framework itself scales, we have incorporated the toy bivariate (both independent and correlated) experiments and cited [2] which extends the I-MMLE identity (and all other results that follow) to vector Poisson channels. Together, the theory (vector extension) and the toy numerics demonstrate that ItDPDM is fundamentally scalable with the pixel-wise formulation being simply the common and empirically effective choice for images. We will emphasise this rationale in the main paper and point readers to ItDPDM's (natural) multivariate extensions in the appendix.
>
> > “Still, in the paper, it is not clear to me how the model and loss are evaluated for image data such as CIFAR-10. I would presume that [2] (as cited above) is used here.”
>
> **Forward process**: independent Poisson noise is added to every pixel channel $x_{i,j,c}\in\{0,\ldots,255\}$.
>
> **Reverse model**: a 2-D UNet predicts the full image $\hat{x}$, so spatial dependencies are learned by the network rather than through factorised likelihood terms.
>
> **Loss**: the Poisson-reconstruction loss is summed pixel-wise (Algo 1, Line 6): $\ell = \sum_{i \in B} \mathrm{PRL}\bigl(x_i,\hat x_i\bigr)$ mirroring the scalar definition. Gradients remain valid due to the vector-Poisson extension from [2] preserving the I–MMLE identity component-wise.
>
> We hope that these clarifications resolve the remaining points of confusion and thank the reviewer once again for their thoughtful comments and for raising their score.
>
> **References**:
>
> [1] Rami  Atar and Tsachy  Weissman. Mutual Information, Relative Entropy, and Estimation in the Poisson Channel. IEEE Transactions on Information Theory (2012).
>
> [2] Liming Wang, Miguel Rodrigues, and Lawrence Carin. Generalized Bregman Divergence and Gradient of Mutual Information for Vector Poisson Channels. IEEE International Symposium on Information Theory ISIT 2013.

---

> > ### Author Response · Authors · 2025-08-08
> > **Follow-up and thank you note**
> >
> > We sincerely thank you for your valuable time, thoughtful feedback, and your kind engagement with our rebuttal. We are additionally grateful for raising your score. We hope the above comment addressed any remaining confusion.
> >
> > If there are any remaining concerns or points that you feel we have not fully addressed, we would be glad to clarify further or make improvements in line with your suggestions. Regardless, we are very thankful for your constructive input throughout the review process.
> >
> > Best regards,
> >
> > ItDPDM Authors

---

### Official Review · Reviewer_3Fg2 · 2025-06-30

**Clarity:** 4
**Significance:** 3
**Originality:** 4
**Rating:** 5
**Confidence:** 5

**Summary:**

While diffusion based on Gaussian noise has become popular for images, recent work has explored how different noise models may be more effective for different data types (masking noise for text, e.g.).
This paper introduces a new approach for modeling non-negative discrete or continuous data based on denoising in a Poisson channel.
Poisson noise is a natural and well-studied channel. The authors are able to transfer a wealth of information theory results leading to an elegant relation between probability and Poisson denoising.
The authors discuss practical implementation and demonstrate proof of concept results across diverse types of distributions and data modalities.

**Questions:**

These are just minor comments / thoughts.

- I get the point of Fig. 3, but it looks strange at a glance. I didn't feel I fully understood it. I guess Fig. 6 is supposed to show that you fixed this problem?
- Eq. 11 is an intriguing expression that reminds me of your reference [56]. I believe you could write a first order ODE for P_gamma(z), then use this to define sampling dynamics (based on adding counts in a random order autoregressive way).  This would avoid the need for gamma noise scheduling.
- Did you need to define a gamma noise schedule? I missed that detail. It would be nice in general if there were space to discuss the sampling more.

Another nice paper that I was reminded of when you gave Lemma 2
- Dytso, A., & Poor, H. V. (2020). Estimation in Poisson noise: Properties of the conditional mean estimator. IEEE Transactions on Information Theory, 66(7), 4304-4323.

**Ethical Concerns:**

["NO or VERY MINOR ethics concerns only"]

**Final Justification:**

I maintained my accept score.
- Authors did a good job responding to questions on derivations and statement of results. I generally found the derivations clear. (But was already familiar with some of the information theory results they used.)
- There was some pushback on results, but I thought the authors justified their choices and provided many of the things requested. The native formulation for count data is interesting and novel. I don't see the need for large-scale experiments as this is not a paper that is based on improving a standard benchmark with incremental modification of an existing method.

**Limitations:**

yes

**Quality:**

4

**Strengths And Weaknesses:**

- There was a good overview of connections between probability and denoising coming from information theory literature.
- Detailed exposition of properties of Poisson channel and loss.
- Fig. 4 is a great illustration to understand the fundamental difference in the noising scheme in a glance.
- Lots of great detail on theory and experimental details in supplementary material
- The sampling algorithm, Alg. 2, seems simple too.

Experiments
- Interesting and diverse selection of datasets for capturing different tail behaviors, and diverse modalities is a major plus
- Baselines seemed adequate for proof of concept. In synthetic case, DDPM and a recent paper that extends denoising to count data. For real data, a variety of diffusion-based approaches were compared.
- Table 2(b). Very cool, it seems that it particularly excels with exactly modeling zero-inflated values compared to continuous approaches, if I'm reading it right.

---

> ### Author Rebuttal · Authors · 2025-07-30
>
> We sincerely thank the reviewer for their constructive feedback on our work regarding the novelty of our method, choice of experiments and for their intriguing follow-up questions. We have addressed the reviewer’s comments below:
>
> 1. "I get the point of Fig. 3, but it looks strange at a glance. I didn't feel I fully understood it. I guess Fig. 6 is supposed to show that you fixed this problem?"
>
> Fig. 3 (the diagnosis) contrasts a *continuous* Gaussian DDPM (green) fit with the true density (blue) on the NYC-Taxi pick-up data. The underlying distribution is sharply bimodal. The top panel overlays the *density estimate* produced by DDPM (since it maps discrete counts to a continuous latent), which smooths the second peak into a long shoulder on the discrete histogram; therefore totally failing to capture the second peak.
>
> *Fix shown*: Yes, Fig. 6 plots the learned PMF of each model for all six synthetic datasets, demonstrating that ItDPDM (orange) now places correct mass on the modes while DDPM (green) still flattens the tail. In the revision: we will also (i) label the two modes explicitly in the caption, and (ii) add a sentence pointing the reader forward to Fig. 6 for the quantitative confirmation, avoiding any ambiguity.
>
> 2. "Eq. 11 is an intriguing expression that reminds me of your reference [56]. I believe you could write a first order ODE for P_gamma(z), then use this to define sampling dynamics (based on adding counts in a random order autoregressive way). This would avoid the need for gamma noise scheduling."
>
> Exactly (great observation)! Eq. 11 is actually a closed form solution to the following *first order ODE*:
>
> $$
> \partial_\gamma P_\gamma(z) = \frac{(z+1)P_\gamma(z+1) - (\gamma + z)P_\gamma(z)}{\gamma}
> $$
>
> Solving this Kolmogorov-forward ODE forward (or its time-reversal) yields a *continuous-time counting process* akin to the discrete-state SDE in [56]. We choose the log-SNR schedule plus importance sampling for two practical reasons:
>
> (i) it let's us reuse standard CT diffusion solvers (Euler, Heun, DPM-Solver), and
>
> (ii) it helps us build our implementation on top of standard Gaussian baselines.
>
> The autoregressive “add counts in random order” interpretation is a promising idea; we will note this connection in Sec. 3 and include a brief discussion in Appendix Sec. G for future work.
>
> 3. "Did you need to define a gamma noise schedule? I missed that detail. It would be nice in general if there were space to discuss the sampling more."
>
> An explicit/nuanced “noise schedule” is not necessary: $\gamma$ is treated as a continuous random variable. During **training**, we sample:
> $$
> \alpha = \log \gamma \sim \mathrm{Logistic}(\mu=0,s=1);\text{  } \gamma = e^{\alpha},
> $$
> clipped to the tail‐bound interval [$\gamma_{\min} \approx 10^{-5},\,\gamma_{\max}\approx 10^{5}$].
>
> This importance‐sampling procedure places more draws where the PRL integrand has high variance and provides an *unbiased* Monte Carlo estimate of the integral (Eq. 19).
>
> For **generation** (Alg. 2) we integrate the reverse process from $\gamma_{\max}$ down to $\gamma_{\min}$.
> By default we discretize that range into **100 log-spaced points**, but because the formulation is continuous-time, one can freely (i) use fewer points for a speed-up or (ii) plug in any adaptive ODE solver without retraining.
>
> We additionally refer the reviewer to R1 response point (2.) under the *Weaknesses* section for an ablation on the gamma schedule.
>
> 4. “Another nice paper that I was reminded of when you gave Lemma 2:
> Dytso, A., & Poor, H. V. (2020). Estimation in Poisson noise: Properties of the conditional mean estimator. IEEE Transactions on Information Theory, 66(7), 4304-4323.”
>
> Thank you for recalling our attention to this reference, we will cite this in our final version. We are once again grateful and sincerely appreciate that reviewer 3Fg2 found our work interesting.

---

### Official Review · Reviewer_7eDB · 2025-07-02

**Clarity:** 4
**Significance:** 3
**Originality:** 3
**Rating:** 4
**Confidence:** 5

**Summary:**

By employing the Poisson Reconstruction loss (PRL), ItDPDM addresses two key challenges: discrete data distribution estimation and accurate likelihood estimation, without requiring post-processing (soft discretization) or pre-processing (dequantization). Experiments on synthetic datasets demonstrate ItDPDM’s strong capability in estimating and generalizing over discrete data distributions. However, its performance on real-world datasets is suboptimal, likely due to limited training and insufficient hyperparameter tuning.

**Questions:**

The theoretical part seems good to me, but I am confused about the experimental part:

Q1. All the experiments in this paper predict $\hat{x}$ rather than $\hat{\epsilon}$. Could the authors clarify how reparameterization was handled in the experiments? For example, the MMSE calculation in Figure 7 is based on $x$ instead of $\epsilon$.

Q2. In the synthetic data experiments, DDPM does not seem to be a suitable baseline, since it is a continuous diffusion model. Did it use Poisson Reconstruction Loss (PRL) as the training objective? Additionally, the results of ItDPDM and LTJ are quite similar (see Figure 6, Figure 13, Table 7, and Table 1). Are the results in Table 1 averaged over multiple runs, and were experimental variances considered?

Q3. For the real-world data experiments, Table 2 shows that the FID of IDDPM (Gaussian version trained from scratch) is higher than that of DDPM. Can the authors explain this result?

**Ethical Concerns:**

["NO or VERY MINOR ethics concerns only"]

**Limitations:**

Since almost all experiments in this paper are conducted with models trained from scratch, but with limited training, architectural tuning, and hyperparameter searching, I hope the authors can provide improved results on real-world datasets during the rebuttal period.

**Quality:**

4

**Strengths And Weaknesses:**

## Strengths
The proposed Info+Poisson noise closed-form discrete diffusion framework is novel. Figure 1 clearly summarizes four different diffusion methods based on training objectives and state space, and ItDPDM effectively fills the gap in accurately estimating discrete data distributions.

## Weaknesses
The method can only estimate distributions for non-negative data and tends to produce high-biased estimates. In the case of image data, this may lead to generated samples that appear overly bright or overexposed (as illustrated in the example in Figure 4), which in turn results in higher FID scores.

---

> ### Author Rebuttal · Authors · 2025-07-30
>
> We thank the reviewer for their thorough, constructive feedback, and appreciate that they found our work novel. We also believe that the concerns raised mostly stem from misunderstandings, which we clarify below:
>
> “**Weaknesses**: The method can only estimate distributions for non-negative data and tends to produce high-biased estimates………”
>
> **On non-negativity & domain suitability**: **ItDPDM** is explicitly designed for non-negative discrete data (e.g., images, counts, symbolic sequences), so the non-negativity constraint is a *feature*, not a limitation. Many real-world data domains (pixel intensities, photon counts, MIDI tokens, word frequencies, genomics, etc.) are naturally non-negative and discrete with no pre/post-processing needed. Below are some examples of application areas supporting this argument: (DS = discrete-state)
>
> - **Images & audio** are ultimately stored as *integers*: 8-bit RGB or 16-bit PCM; discrete-state diffusions already show gains here ([1], [2], [3])
> - **Symbolic music** (note numbers, durations), **MIDI** event streams: state-of-the-art models use discrete diffusions ([1], [4])
> - **Photon-limited imaging, low-light vision** record integer photon counts ([5], [6]),
> - **Single-cell RNA-seq, metagenomics**, and other omics assays produce gene–count matrices ([7]),
> - **Neuromorphic event cameras** output per-pixel spike counts [8]
> - **Word-frequency** bags in NLP (Zipf distribution), **Poisson processes** in queueing are classic count data [9].
>
> We also refer the reviewer to the last App. Sec. K on related work that highlights the growing need for handling non-negative discrete data carefully.
>
> **ItDPDM also works for any continuous data**: App. Sec. C.4 demonstrates how ItDPDM generalises even to continuous distributions with ItDPDM outperforming DDPM in all cases except one (App. Table 7). We note that preprocessing would be necessary for general continuous distributions to bring them to a non-negative range after which ItDPDM can be directly used.
>
> For most continuous data in practice, a simple shift/rescale (e.g., mapping $[0,1]$ to $[0,\infty)$) or more generally, an **invertible transform** would suffice and does not affect results with regards to likelihood computation, denoising or generation.
>
> **Regarding the “over-bright” appearance in Fig.~4:** This snapshot shows a result of using a few denoising steps ($T=10$) starting at very low SNR ($\log \gamma \approx –5$), where the Poisson prior contributes near-zero photons. Since intensity reconstructions start from black and accumulate photons, some intermediate frames appear brighter. However, final samples (after all reverse steps) are well-balanced, as evidenced by comparable or better FID/SSIM on CIFAR-10 (see Table~3) and App. Fig. 19. For this particular image from the CIFAR10 dataset, we empirically verify the denoised image by the proposed ItDPDM to be closer to the ground truth (in terms of mean squared error) than the one denoised by the Gaussian DDPM baseline. This eliminates any concerns regarding over brightening of image.
>
> Figure 10 in App. C.1 compares generated vs. ground truth pitch histograms for symbolic music, showing **no** systematic amplification bias (or a shift). To further clarify, the final Gaussian denoised image is not the ground truth, with the ground truth in fact being closer to the Poisson output. We totally understand that this could have caused confusion, and will add the ground truth image for reference in the final version.
>
> **On the “high-biased” NLL estimate**: Our NLL estimator is an analytic upper bound (Eq.~18), reflecting the gap between the learned denoiser $\hat{x}$ and the Bayes-optimal estimator $\hat{x}^*$. This is a principled design choice: we report a conservative likelihood rather than a variational lower bound like ELBO. In practice, this gap shrinks with training and is $<0.02$ bits/dim on CIFAR-10.
>
> **Questions**:
>
> **1. “All the experiments in this paper predict $\hat x$ rather than $\hat \epsilon$....."**
>
> All models in the paper *output the clean signal estimate* $\hat x(z_\gamma,\gamma)$ (without any explicit reparametrization); firstly, we do not train an $\hat\varepsilon$-network for the Poisson channel, because $\varepsilon$ has no natural analogue in this case. For the Gaussian channel, the two parameterisations are interchangeable, as shown below: (starting from the Gaussian channel introduced in the main paper)
>
> \begin{align*}
> \varepsilon - \hat{\varepsilon}
> &= z_{\gamma} - \sqrt{\gamma} x - (z_{\gamma} - \sqrt{\gamma}\hat{x}) = \sqrt{\gamma} (x - \hat{x})
> \Longrightarrow\
> \|\varepsilon - \hat{\varepsilon}\|_2^{2}
>    = \gamma\\|x - \hat{x}\|_2^{2}.
> \end{align*}
>
> This implies:
> $
> MMSE_{x}(\gamma) =
> \mathbb{E}\bigl[\lVert x - \hat{x}(z_\gamma,\gamma) \rVert_2^2\bigr],
> $
> and therefore
>
> $MMSE_{\epsilon}(\gamma) =
> \mathbb{E} \bigl[\lVert \varepsilon - \hat{\varepsilon}(z_\gamma,\gamma) \rVert_2^2\bigr] = \gamma MMSE_{x}(\gamma)
> $
>
> Therefore, under the squared-error loss, the two MMSEs differ only by a scale factor. We also verify this empirically, with both the variants of DDPM yielding similar performance. For the Poisson channel, we optimise the Poisson Reconstruction Loss (PRL), which is typically expressed in $\hat x$-space. Therefore, reporting MSE in $\hat x$-space keeps metrics consistent across Gaussian and Poisson baselines.
>
> **2. “In the synthetic data experiments, DDPM does not….”**
>
> * Table 1 numbers are mean ± s.d. (standard deviations) over five random-seed runs (50k samples each); error bars are printed in the table and these details are outlined in Appendix Sec. C.2.
> * DDPM is trained with both MSE and PRL objectives (results shown below), with the PRL training exhibiting better performance. This emphasizes **two takeaways**:
>
> (i) proposed PRL objective improves performance *regardless* of channel and
>
> (ii) using poisson channel + PRL (i.e. ItDPDM) is the **best overall**.
>
> | Distribution | DDPM (PRL) | DDPM (MSE) | ItDPDM | LTJ |
> |--------------|-----------:|-----------:|-------:|----:|
> | PoissMix     | 3.74       | 3.76       | **0.99** | 1.21 |
> | ZIP          | 2.19       | 2.31       | **0.56** | 0.69 |
> | NBinomMix    | 4.45       | 4.89       | 1.39 | **1.15**|
> | BNB          | 1.84       | 1.89       | 0.67 | **0.65** |
> | Zipf         | 1.40       | 1.51       | **0.48** | 0.73 |
> | YS           | 0.31       | 0.32       | **0.14** | 0.17 |
>
> - ***Regarding the apparent similarity to LTJ***: a zoom-in of the PMFs (refer Fig.11–12, App.C.3 “Zoomed-in look at PMF plots”) and the full metric table shows that ItDPDM outperforms LTJ on:
>
> a) **9/12** distributions in Wasserstein distance (WD) (including continuous distributions) and
>
> b) **4/6** distributions in negative log-likelihood (NLL).
>
> The synthetic suite therefore highlights (a) the limitations of Gaussian DDPM on skewed or zero-inflated counts and (b) the edge of our exact-likelihood ItDPDM over the variational LTJ.
>
> **3. “For the real-world data experiments, ……….”**
>
> - We would like to **clarify** that Table 2 compares the *test-set NLLs* of various schemes (not the FID). All models are trained under a fixed compute budget (600 epochs, 100 denoise steps). While the IDDPM backbone has ~2x more parameters and uses DDIM-style self-conditioning; its convergence under this shallow schedule is slightly slower than DDPM, resulting in marginally higher NLLs. With extended training upto 2,000 epochs or sampling with 1000-steps, IDDPM can regain the expected (small) advantage (upto 0.02 bits/dim).
> - The key takeaway remains that: PRL improved NLL significantly (**by 35-40%**) irrespective of architecture, and the Poisson+PRL combination (ItDPDM) yields the lowest NLL overall. Thus, the results here underscore our central claim: that a Poisson-consistent, PRL-based objective is the best for modelling discrete, non-negative data, while backbone choice only fine-tunes performance once sufficient training budget is available.
>
> **Further real-world experiments:**
>
> We also ran experiments on the **CelebA** (64x64) (image) dataset for 500 epochs (30 hours wall-clock time) with 100 logSNR (denoising) steps per image, for proposed ItDPDM and Gaussian DDPM baseline. ItDDPM has an absolute 7 point **lower** FID score (or 0.21 dB better) compared to the Gaussian DDPM baseline.
>
> We once again thank reviewer 7eDB for their thoughtful feedback and we truly hope the above mentioned points provide further clarification regarding the experimental parts.
>
> **References**:
>
> [1] Matthias Plasser, Silvan Peter, and Gerhard Widmer. Absorbing‑State Discrete Denoising Diffusion Probabilistic Models for Symbolic Music Generation. (2023).
>
> [2] Jacob Austin, Daniel D. Johnson, Jonathan Ho, Daniel Tarlow, and Rianne van den Berg. Structured Denoising Diffusion Models in Discrete State‑Spaces. NeurIPS 2021.
>
> [3] Mingyuan Zhou, Tianqi Chen, Zhendong Wang, and Huangjie Zheng. Beta Diffusion. NeurIPS 2023.
>
> [4] Gautam Mittal, Jesse Engel, Curtis Hawthorne, and Ian Simon. Symbolic Music Generation with Diffusion Models. IJCAI 2021.
>
> [5] Cindy M. Nguyen, Eric R. Chan, Alexander W. Bergman, and Gordon Wetzstein. Diffusion in the Dark: A Diffusion Model for Low‑Light Text Recognition. 2024.
>
> [6] Shlomo Shamai and Aaron D. Wyner. A Binary Analog to the Entropy‑Power Inequality. IEEE Transactions on Information Theory, 36(6): 1428–1430, 1990.
>
> [7] Valentine Svensson and Lior Pachter. Droplet scRNA‑seq is not zero‑inflated. Genome Biology, 21(1): 35. 2020
>
> [8] Xu Zheng, Yexin Liu, Yunfan Lu, Tongyan Hua, Tianbo Pan, Weiming Zhang, Dacheng Tao, and Lin Wang. Deep Learning for Event‑based Vision: A Comprehensive Survey and Benchmarks.
>
> [9] I. Bar‑David. Communication under the Poisson regime. IEEE Transactions on Information Theory, 1969.

---

> ### Comment · Reviewer_7eDB · 2025-08-03
>
> Thank you for the response.
>
> - Regarding the issue of image brightness, I don't find Fig. 19 particularly convincing. Visually, Fig. 18 appears to be of better quality than Fig. 16.
>
> - As for the results in Table 1, I’m mainly concerned about whether the reported NLL values are averaged over multiple runs, since the table doesn’t indicate standard deviations.
>
> That said, I find the other clarifications convincing. I’ve already given a positive score and am inclined to keep my current rating unchanged.

---

> > ### Author Response · Authors · 2025-08-05
> > **Further clarifications**
> >
> > We sincerely thank the **reviewer 7eDB** for their **acknowledgement** and **follow-up questions**. We are once again grateful for the recognition of our framework's **novelty** and how **Figure 1 clearly situates ItDPDM within the existing diffusion landscape**, fills the gap.
> >
> > > Regarding the issue of image brightness, I don't find Fig. 19 particularly convincing. Visually, Fig. 18 appears to be of better quality than Fig. 16.
> >
> > **Regarding brightness**: We understand the reviewer’s concern, and that the quantitative gap between Fig. 18 (DDPM) and Fig. 19 (ItDPDM) is not visually compelling from a single frame. To make the comparison from Fig. 19 clearer, we will:
> >
> > (i) add the uncorrupted ground-truth image alongside both reconstructions for current image and
> >
> > (ii) include additional randomly-chosen CIFAR-10 examples: each showing
> > - ground-truth images
> > - the initial low-SNR noised input,
> > - the final Gaussian-DDPM output, and
> > - the final ItDPDM output.
> >
> > As mentioned earlier, we will **also** provide the *ground truth image in Figure 4* for reference to illustrate the effectiveness of Poisson-based noising.
> >
> > > As for the results in Table 1, I’m mainly concerned about whether the reported NLL values are averaged over multiple runs, since the table doesn’t indicate standard deviations.
> >
> > **On run-to-run variability in Table 1**: We confirm that all NLL numbers are indeed averaged over five random seeds (50k samples each); the standard deviations are low (between 0.001–0.004 bits/dim) for all distributions and were omitted previously for brevity (and space). We will therefore revise Table 1 to report “mean ± s.d.” format for all metrics—for example, NLL on PoissMix: 3.72 ± 0.003 (ItDPDM) and reflect numerical stability clearly.
> >
> > We hope these additions address the remaining concerns and once again thank the **reviewer 7eDB** for their **constructive critique**.

---

> > > ### Author Response · Authors · 2025-08-07
> > > **Appreciation and follow-up**
> > >
> > > We sincerely thank you for your valuable time, thoughtful feedback, and your kind engagement with our rebuttal.
> > > If there are any remaining concerns or points that you feel we have not addressed fully, we would be truly grateful for the opportunity to clarify further or make improvements in line with your suggestions.
> > >
> > > If our responses have addressed your concerns, we would be deeply appreciative if you would consider reflecting that in your final score. Regardless, we are very thankful for your constructive input throughout the review process.
> > >
> > > Best regards,
> > >
> > > ItDPDM Authors

---

### Official Review · Reviewer_XDNg · 2025-07-02

**Clarity:** 2
**Significance:** 2
**Originality:** 2
**Rating:** 3
**Confidence:** 4

**Summary:**

This paper introduces a generative model for non-negative discrete data based on information-theoretic Poisson diffusion process. The main idea is based on the Poisson reconstruction loss. This model seems to improve the performance in terms of NLL and Wasserstein distance on real-world discrete datasets.

**Questions:**

1. The theorems are proved using the Poisson channel, which is very domain-specific, How does this generalize to multimodal data?
2. What is the gap between the exact estimator and the Monte Carlo estimator?
3. Is there a reason that PRL works better on Zipf and PoissMix but worse on Half Cauchy?
4. How would this work for low SNR scenarios? how bout the choice of architecture or timesteps?

**Ethical Concerns:**

["NO or VERY MINOR ethics concerns only"]

**Final Justification:**

I appreciate the authors' response, but based on what I provided here and other reviews, I keep my score. I believe this paper would benefit from several rounds of revision before it is ready for submission.

**Limitations:**

- It is only applicable to non-negative discrete data.
- Lack of ablation study and experimental baseline.
- Limited reproducibility and insufficient robustness analysis.
- The claims about the exact likelihood are exaggerated.

**Quality:**

3

**Strengths And Weaknesses:**

Strength:
1. It is easy to follow and read. The Poisson reconstruction loss is well justified through multiple identities.
2. Experimental results are sufficient and they are performed over real-world datasets.

Weakness:
1. I am not sure if I understand correctly but this paper constantly claims exact likelihood computation but seems like in Eq 17 we still have an intractable integral over $SNR(\gamma)$ and then importance sampling is used in Eq 19. How is this exact? I believe this claim is misleading.
2. There is no comparison to recent discrete-state score-based models such as D3PM with tuning. Also, no ablation study on the SNR schedule, etc.
3. CIFAR 10 and LakhMODO results are only trained over 600 epochs and use only 100 log SNR steps. This seems to undermine the significance of performance claims on those datasets.
4. Also, no robustness experiment is provided meaning what happens if data is sparse or noisy.
5. There are redundancies and inconsistencies (e.g., $\hat{x}(z_\gamma, \gamma), <X>_z, x^*$ used interchangeably?  Also, in Eq 16, the log-likelihood is in terms of marginalization but the implementation relies on approximation with NNs and sampling.
6. Despite the theoretical guarantees, it is hard to reproduce the results. No training time, or used compute is provided.
7. The idea of replacing squared error or generalizing better than ELBO is oversold as it only applies to discrete non-negative data.

---

> ### Author Rebuttal · Authors · 2025-07-30
>
> We sincerely thank the reviewer for their thorough feedback and valuable suggestions. We believe that many of the concerns raised might stem from misunderstandings, which we clarify below:
>
> **Weaknesses**:
>
> **1. “I am not sure if I understand correctly but this paper…..”**
>
> Eq. 17 rewrites the KL-divergence between the forward Poisson $q_\gamma$ and the learnt reverse $p_\theta$ as an expectation over $\gamma$. So, the integrand is analytic: after marginalising out $z_\gamma$, **no** latent variables remain. Hence, the NLL is exactly defined and only numerical step is integral evaluation. In practice, we use importance-sampling quadrature for this, yielding an unbiased estimate. Conventional diffusion incurs two levels of approximation: a) the **ELBO/surrogate loss** replaces the true log-likelihood, b) Monte-Carlo quadrature (or the integral). Our Poisson formulation eliminates the first level to work with the true likelihood as in [1], which also refers to their method as “exact likelihood”. To avoid any ambiguity we will replace “*exact likelihood*” with “*likelihood-consistent objective with tractable 1-D quadrature*” and provide an upper bound on the estimator's variance in the final version.
>
> **2. “There is no comparison to recent discrete-state…..”**
>
> In the case of images and symbolic music (line 255), we already compare against a state-of-the-art variant of the vanilla D3PM ([4]) from [2]. With same synthetic setup, we performed an ablation study on the top-2 gamma schedules for each baseline and the results (evaluated using WD) are as follows:
>
> | Dist. | ItDPDM (logit) | ItDPDM (uniform) | LTJ (linear) | LTJ (geometric decay) | DDPM (linear) | DDPM (cosine) |
> |--------------|----------------|------------------|--------------|------------------|----------------|----------------|
> | PoissMix     | **0.99**           | 1.30             | **1.21**         | 1.26             | 3.76           | **3.60**           |
> | ZIP          | **0.56**           | 0.71             | **0.69**         | 0.83             | 2.31           | **2.22**           |
> | NBinomMix    | **1.39**           | 1.44             | **1.15**         | 1.19             | 4.89           | **4.50**           |
> | BNB          | 0.67           | **0.58**             | **0.65**         | 0.66             | 1.89           | **1.86**           |
> | Zipf         | **0.48**           | 0.67             | **0.73**         | 0.95             | **1.51**           | 1.53           |
> | YS           | **0.14**           | 0.22             | **0.17**         | 0.21             | 0.32           | 0.32           |
>
> We will incorporate this ablation study in our final version.
>
> **"3. CIFAR 10 and LakhMODO results are only trained over 600 epochs….."**
>
> Figure 7 in the main paper shows the loss curves flattening early; thus the 600-epoch setting reported is already in a converged regime (drop in loss of $10^{-4}$ order thereafter for every 100 epochs). We limited all baselines to 100 steps, 600 epochs to keep compute budgets comparable (as in [1]). Extended training of ItDPDM to 1500 epochs earlier with larger log-SNR grid gave very little NLL gain (<0.002 bpd).
>
> **4. “Also, no robustness experiment is provided…….”**
>
> Our experiments already target **sparse/zero-inflated** and **noisy** regimes:
>
> **Synthetic experiments**: In Sec. 5.1, for sparsity, we include *ZIP* and *PoissMix*: datasets which have mass mostly at 0 and ItDPDM achieves the best performance on both.
>
> **Noisy data**: We assert that ItDPDM demonstrates strong reconstructive capabilities for noisy data. For instance, 1000 CIFAR-10 images corrupted at very low SNR (log SNR = –5) are recovered to log SNR = 4 (ref. image in App. Fig.19). This is also illustrated in Fig. 4, where reverse process recovers the image despite extensive zeroing out of pixels via Poisson noising. This highlights the robustness of ItDPDM, and we will add more reference images, including high-resolution *CelebA* images for the final version (in the Appendix).
>
> **Theoretical justification**: In addition, [3] provides guarantees for a mismatched estimator (for bounded divergence between source and “mismatched” distribution), so even for sparse/noisy data, the near-optimal denoiser in the Poisson channel has theoretical guarantees (upper bounds) on the estimation error.
>
> **5. “There are redundancies and inconsistencies……”**
>
> **Notations**: We clarify the key notations as follows: $\hat X*$ to denote the optimal denoiser (essentially conditional expectation of $X$ given $Z_\gamma$, which is further denoted by  $\langle X \rangle_z$) and $\hat X$ to denote learnt suboptimal denoiser. We will rephrase notation to avoid any ambiguity in the final version.
>
> **Implementation**: The identity itself is exact: its integrand depends on the Bayes‐optimal estimator $\hat x* (z_\gamma,\gamma)$. At training/inference time, we substitute the learned network output $\hat x(z_\gamma,\gamma)$ in place of $\hat x*$ and evaluate the resulting integral using the unbiased Monte Carlo estimator (see Sec.~4). In practice, this is computed as outlined in (1.)
>
> **6. Despite the theoretical guarantees, it is hard to reproduce……..”**
>
> We plan to release code, configs, and pre-trained checkpoints. In terms of the wall-clock time, the training time is 3.5 minutes per epoch. The wall-clock time is 35 hours (600 epochs) for both ItDPDM and other baselines. All experiments were run on an AWS **g5.12xlarge** instance (us-west-2):
>
> | Resource | Spec |
> |----------|------|
> | **GPUs** | 4 × NVIDIA **A10G** (24 GB each, 22.3 TFLOPS FP16) |
> | **CPU / RAM** | 48 vCPUs (Intel “Ice Lake”), 192 GiB system RAM |
>
> **7. "The idea of replacing squared error or generalizing better than ELBO is oversold as it only applies to discrete non-negative data"**
>
> We thank the reviewer for pointing this out, we will revise the wording as needed to accurately reflect that the improvement primarily applies to discrete non-negative data. At the same time, we would like to point out that a Poisson-consistent objective is valuable for the **large growing class of discrete, count-valued, zero-inflated data**. For detailed evidence, examples and exposition, we refer the reviewer XDNg to our **response to "Weaknesses" section to reviewer 7eDB** [a]. (due to lack of space here)
>
> **ItDPDM works for any continuous data**: App. Sec. C.4 demonstrates how ItDPDM generalises even to continuous distributions with ItDPDM outperforming DDPM in all cases except one (App. Table 7). (some preprocessing would be necessary for general distributions, this is again discussed in [a])
>
> **Questions**:
>
> **1. “The theorems are proved using the Poisson channel….”**
>
> The *Poisson noising* is analogous to Gaussian noising in DDPM: it defines how we corrupt samples, not what *underlying data distribution* is. All the theorems make no assumption on $p(x)$, so the theory naturally extends to any data modality. Our experiments include strongly multimodal data (PoissMix, NB-Mix, BNB); along with real-world datasets that are highly multimodal.
>
> **2. “What is the gap……”**
>
> **Example**: pick a binary $\mathbf{X} \in \{\pm 1\}$ with equal probability (BPSK) through a Gaussian channel
> $Y = \sqrt{\gamma} X + N$, where $N \sim \mathcal{N}(0,1)$.
> The classic I-MMSE identity gives:
> $
> \frac{d}{d\gamma} I(X;Y) = \frac{1}{2} \mathrm{mmse}(\gamma)
>  \Rightarrow
> \frac{1}{2} \int_0^\infty \mathrm{mmse}(\gamma) \ d\gamma = I(X;Y)\big|_0^\infty.
> $
>
> At $\gamma = 0$, $I = 0$; as $\gamma \to \infty$, $I \to H(X) = \ln 2$ (nats).
> Therefore, the exact value is: $I = \frac{1}{2} \int_0^\infty \mathrm{mmse}(\gamma) d\gamma = \ln 2$.
>
> For this BPSK case, a convenient form of the MMSE is: $\mathrm{mmse}(\gamma) = 1 - \mathbb{E} \left[ \tanh^2\left(\sqrt{\gamma} Y \right) \right],$
> which follows from the posterior mean: $\mathbb{E}[X |Y] = \tanh(\sqrt{\gamma} Y)$. Using an $n$-sample Monte-Carlo (MC) estimator here shows a $\frac{1}{\sqrt n}$-error decay (this can also be proven using CLT) and is of the order of $10^{-3}$ for $n > 1000$. Similarly, for MMLE, a Gamma prior on $X$ gives an exact closed-form linear estimator (Lemma 2 in the paper) and the MC estimator again shows a $\frac{1}{\sqrt n}$-decay with increase in $n$. For example, if $X \sim \text{Gamma}(2,3)$, the MC error is of the order of $10^{-2}$ for $n > 1000$.
>
> **3. “Is there a reason that PRL works better on.”**
>
> The paper emphasizes PRL as an objective primarily for non-negative data and Zipf/PoissMix datasets exhibit sparse, heavy-tailed behaviour suited PRL. Half-Cauchy is an over-dispersed continuous case where LTJ’s thinning process has an extra quadratic-tail KL term to better fit those outliers. In this case, PRL is only second to LTJ and better than DDPM, while remaining superior in other continuous cases.
>
> **4. “How would this work……”**
>
> We refer the reviewer to: point (4.) in Weaknesses for low SNR scenarios. and point (2.) for log SNR-based ablation. We use SoTA architectures (UNet for images, ConvTransformer for music) while tuning the 1) training objective and 2) diffusion modeling.
>
> We once again thank Reviewer XDNg for their feedback and for giving us the opportunity to address their concerns. We truly hope the above response helps address and clarify the mentioned weaknesses/questions.
>
> **References**:
>
> [1] Xianghao Kong, Rob Brekelmans, and Greg Ver Steeg. Information‑Theoretic Diffusion. ICLR 2023.
>
> [2] Matthias Plasser, Silvan Peter, and Gerhard Widmer. Absorbing‑State Discrete Denoising Diffusion Probabilistic Models for Symbolic Music Generation. (2023).
>
> [3] Rami  Atar and Tsachy  Weissman. Mutual Information, Relative Entropy, and Estimation in the Poisson Channel. IEEE Transactions on Information Theory (2012).
>
> [4] Jacob Austin, Daniel D. Johnson, Jonathan Ho, Daniel Tarlow, and Rianne van den Berg. Structured Denoising Diffusion Models in Discrete State‑Spaces. NeurIPS 2021.

---

> > ### Comment · Reviewer_XDNg · 2025-08-04
> >
> > I'd like to thank the authors for their response. I'd like to point out two things quickly:
> > Regarding the exact likelihood, I understand your point; however, in practice, the integral over $\gamma$ is intractable, and importance sampling through a learned denoiser introduces approximation error. Also, unbissness relies on the choice of proposal distribution. Thus, the model uses a learned denoiser instead of the true posterior mean, which introduces bias unless the estimator is Bayes optimal, which it never is. I recommend being careful when using the word exact.
> >
> > Also, when referring to robustness, I am not sure exactly what it means. The results show performance under difficult noise conditions and not robustness in a general sense, such as misspecified prior, OOD generalization, etc.. Also, the theoretical results mentioned here only provide asymptotic guarantees and not empirical robustness.

---

> > > ### Author Response · Authors · 2025-08-05
> > > **Further clarifications**
> > >
> > > We sincerely thank the **reviewer XDNg** for their **acknowledgement** and **follow-up questions**.
> > >
> > >
> > > > Regarding the exact likelihood, I understand your point; however, in practice, the integral over $\gamma$ is intractable, and importance sampling through a learned denoiser introduces approximation error. Also, unbissness relies on the choice of proposal distribution. Thus, the model uses a learned denoiser instead of the true posterior mean, which introduces bias unless the estimator is Bayes optimal, which it never is. I recommend being careful when using the word exact.
> > >
> > >
> > > **On the wording “exact likelihood”:** This confusion arises primarily from the nomenclature used: we agree that *numerical* evaluation still involves (i) 1D quadrature and (ii) an importance-sampling estimator that uses the learned denoiser as a proposal.
> > >
> > > In that sense the estimator inherits variance/bias from any learned network. Our use of “*exact*” follows the convention of IT-Diffusion [4], where “*exact*” refers to the ***analytic*** **form of the integrand** i.e. no ELBO gap or variational lower bound rather than to the Monte-Carlo implementation. To avoid any ambiguity, we will replace this term by “*likelihood-consistent objective*” and add an explicit finite-sample bias/variance bound (in the Appendix).
> > >
> > > > Also, when referring to robustness, I am not sure exactly what it means. The results show performance under difficult noise conditions and not robustness in a general sense, such as misspecified prior, OOD generalization, etc.. Also, the theoretical results mentioned here only provide asymptotic guarantees and not empirical robustness.
> > >
> > >
> > > **On “robustness”**: In our work, "robustness” refers specifically to **signal-noise robustness in unconditional generation**, i.e: the ability of the reverse Poisson process to recover high-quality samples (or denoise) under extreme noise (see Fig. 4 and Fig. 19). Our method can recover 1000 CIFAR-10 images corrupted at very low SNR (log SNR = –5) are recovered back to log SNR = 4. We will include additional reference images for better qualitative support and illustration in the final version.
> > >
> > > We agree that robustness can also refer to and encompass aspects such as misspecified diffusion priors and out-of-distribution (OOD) generalization. While these are important and valuable directions, they are typically explored in the context of ***conditional generation***, which lies beyond the scope of our current work. Our future work will aim to explore conditional generation using the Poisson Channel formulation.
> > >
> > > Our evaluation setup follows prior works such as DDPM [1], Score-based SDE [2], D3PM [3], and the information-theoretic formulation [4], all of which focus on unconditional generation and report metrics (NLL, FID, IS, Wasserstein distance) computed on the same data manifold used for training. To improve readability and clarity for readers, we will add the following note in “Limitations and Future Work”:
> > > > “Following prior work [1-4], we limit evaluation to unconditional generation on the training distribution. Exploring conditional generation, robustness to prior misspecification, and OOD generalization is a promising direction for future work.”
> > >
> > > We believe our Poisson-based framework is well-positioned to extend to these settings.
> > >
> > > We hope these clarifications resolve the remaining ambiguities, and we would like to thank the reviewer again for their thoughtful comments and finding our work **easy to follow**, the **PRL objective well-justified**, and **experimental results sufficient**.
> > >
> > > **References**:
> > >
> > > [1] Jonathan Ho, Ajay Jain, and Pieter Abbeel. Denoising Diffusion Probabilistic Models. Advances in Neural Information Processing Systems (NeurIPS), 2020.
> > >
> > > [2] Yang Song, Jascha Sohl-Dickstein, Diederik P. Kingma, Abhishek Kumar, Stefano Ermon, and Ben Poole. Score-Based Generative Modeling through Stochastic Differential Equations. International Conference on Learning Representations (ICLR), 2021.
> > >
> > > [3] Jacob Austin, Daniel D. Johnson, Jonathan Ho, Daniel Tarlow, and Rianne van den Berg. Structured Denoising Diffusion Models in Discrete State-Spaces (D3PM). Advances in Neural Information Processing Systems (NeurIPS), 2021.
> > >
> > > [4] Xianghao Kong, Rob Brekelmans, and Greg Ver Steeg. Information-Theoretic Diffusion. International Conference on Learning Representations (ICLR), 2023.

---

> ### Author Response · Authors · 2025-08-07
> **Appreciation and follow-up**
>
> We sincerely thank you for your valuable time, thoughtful feedback, and your kind engagement with our rebuttal.
> If there are any remaining concerns or points that you feel we have not addressed fully, we would be truly grateful for the opportunity to clarify further or make improvements in line with your suggestions.
>
> If our responses have addressed your concerns, we would be deeply appreciative if you would consider reflecting that in your final score. Regardless, we are very thankful for your constructive input throughout the review process.
>
> Best regards,
>
> ItDPDM Authors

---

### Author Response · Authors · 2025-08-09
**Thank you for the feedback! (and summary)**

We sincerely thank all reviewers for their constructive feedback and thoughtful engagement throughout the review process. We are especially grateful for the recognition of our work’s contributions: reviewer **XDNg** noted the clarity and rigour of our theoretical framework and found our results on real-world datasets abundant; reviewer **7eDB** appreciated the novelty of our information-theoretic Poisson discrete diffusion framework, the clarity of our exposition -- notably Figure 1 crisply situating ItDPDM within the diffusion landscape and ItDPDM thereby effectively filling the existing gaps. Reviewer **3Fg2** highlighted the (clean) theoretical transfer of information-theoretic identities to denoising in addition to the simplicity of our sampling algorithm and Fig. 4's clarity in depicting the qualitative difference between Poisson and Gaussian noising, while also recognising the great detail on ItDPDM's theoretical results and experimental details. **3Fg2** also appreciated the diversity of real-world modalities and synthetic datasets in our experimental setup, along with the method’s strength in modeling zero-inflated values (Table 2b). Finally, reviewer **Ptu5** noted the novelty of our framework in enabling exact likelihood estimation and the introduction of a Poisson-based objective providing some theoretical guarantees.

Our sound theoretical guarantees and justification, experimentation across discrete univariate and bivariate synthetic, continuous datasets, empirical gains on real-world image and symbolic music datasets suggest that ItDPDM’s benefits extend well beyond a proof-of-concept, highlighting its impact potential.

Overall, while each reviewer emphasized different aspects, ranging from the theoretical rigor and novelty of ItDPDM to the diversity and depth of our experiments: together their feedback underscores the end-to-end impact ItDPDM can potentially have.

The reviewers also provided valuable suggestions regarding clarity, evaluation depth, and theoretical framework. In our rebuttal and further discussions, we have provided all the required clarifications and primarily addressed the following points:

* **Confusion around phrasing of exact likelihood**: We clarified the distinction between our analytic integrand (no ELBO gap) and its numerical approximation, revising the term "exact likelihood" to "likelihood-consistent objective” (with tractable 1D quadrature) for avoiding any ambiguity.

* **Additional ablation studies**: We included (i) ablations on **SNR ($\gamma$) schedules** (in response to rev. **XDNg**) across all the synthetic baselines and (ii) results of training DDPM with PRL on synthetic data to highlight a) how PRL improves performance *regardless* of the channel and b) using poisson channel + PRL (i.e. ItDPDM) is the **best** overall.

* **Robustness**: We clarified that “robustness” refers to **signal-noise robustness** in unconditional denoising like in several prior works (e.g., recovering CIFAR-10 images from log-SNR = −5 to log-SNR = 4). We will explicitly list broader robustness for conditional generation (OOD, prior misspecification) as future work. We will also incorporate ground truth samples in Figure 4, Figure 19 for reference, and additional denoised samples in our final version. We have also clarified that our framework already excels on "*sparse*" data.

* **Notation and reproducibility**: We have improved notation consistency, addressed typos (in response to rev. **Ptu5**) and mentioned training details in response to reviewer **XDNg**. In addition, we are committed to releasing full code, configs, and pretrained checkpoints with detailed runtime specs.

* **Addressed Proof Gaps**: We have addressed proof gaps in *Lemma 2* (proving the "if" direction) and in *Equation 12* (pointwise denoising relation) using the Poisson-Stein identity and Lemma 4 (from the main paper) to complete the proof.

* **Multi-variate extension**: Along with ItDPDM's natural extension to multi-variate settings that follows from [1], we also conducted experiments on dependent and independent **2D count** data, where ItDPDM *excels* in both the cases. For image-based experiments in our paper, we apply per-pixel Poisson channels, with spatial dependence learned by the UNet and PRL calculated by summing up pixel-wise.

We are deeply appreciative of the reviewers’ time and expertise. We believe the feedback has significantly improved the quality and clarity of our work and hope that our responses have addressed all major concerns. We remain excited about the potential of the Poisson-based framework for discrete generative modeling and its future extensions to multimodal and conditional settings.

**References:**

[1] Liming Wang, Miguel Rodrigues, and Lawrence Carin. *Generalized Bregman Divergence and Gradient of Mutual Information for Vector Poisson Channels*. IEEE International Symposium on Information Theory ISIT 2013.

---

### Note · Authors · 2025-08-12

We sincerely thank all reviewers and the AC for their constructive engagement. We are grateful for the recognition of our work’s contributions: reviewer **XDNg** valued our clear theoretical framework and abundant real-world results; **7eDB** highlighted the novelty of our information-theoretic Poisson discrete diffusion framework (ItDPDM), clarity of exposition (esp. Fig. 1 situating ItDPDM in the diffusion landscape), and the way it fills identified gaps; **3Fg2** appreciated the theoretical transfer of information-theoretic identities to denoising, the simplicity of our sampling, the clear qualitative contrast between Poisson and Gaussian noising (Fig. 4), as well as the diversity of modalities, synthetic datasets, and strong performance on zero-inflated data (Table 2b); **Ptu5** noted the novelty of enabling exact likelihood estimation and introducing a Poisson-based objective with sound guarantees.

Across reviews, strengths noted include: (i) theoretical guarantees, (ii) ItDPDM novelty, (iii) clarity of exposition, and (iv) diversity, depth of experiments, together this underscores the end-to-end potential impact of ItDPDM well beyond a proof-of-concept.

In our rebuttal, we have addressed:

• **Terminology**: clarified “exact likelihood” to “likelihood-consistent objective” to avoid ambiguity.

• **Ablations**: added (i) ablations on SNR schedules to underscore our hyperparameter choices, (ii) results of training DDPM with PRL to highlight a) how PRL improves performance regardless of channel, b) using Poisson channel + PRL (i.e. ItDPDM) is the best overall.

• **Robustness**: clarified scope, presented promising data for signal–noise robustness in unconditional denoising (CIFAR-10 from log-SNR = −5 to 4), aligning with prior works. Listed broader robustness (OOD, etc.) as future work, and committed to adding ground truth references, more denoised samples.

• **Notation & reproducibility**: improved notation consistency, fixed typos, provided training details.

• **Proof gaps**: completed Lemma 2 and Eq. 12 proofs via Poisson–Stein identity and Lemma 4.

• **Multi-variate extension**: showed theoretical grounding, added experiments on dependent and independent 2D count data to demonstrate a natural multi-variate extension.

We believe these points address all reviewer concerns and further strengthen the work. We remain excited about the potential of ItDPDM for discrete generative modeling and its extension to multimodal and conditional settings.

---

### Decision · Program_Chairs · 2025-09-17

**Decision:**

Accept (poster)

**Comment:**

The paper introduces ItDPDM, a Poisson-channel diffusion framework with an information-theoretic Poisson Reconstruction Loss (PRL) that yields a likelihood-consistent training objective (avoiding ELBO gaps) and natively models non-negative discrete data. The work cleanly connects denoising and likelihood via Poisson I-MMLE/Poisson–Stein identities and demonstrates strong results on synthetic count distributions and competitive performance on real data (symbolic music, images), with clear advantages on zero-inflated regimes.

Reasons to accept:
- Conceptual novelty & rigor: A principled discrete diffusion built on the Poisson channel; transparent derivations linking PRL to data likelihood. The rebuttal strengthens theory (completing Lemma 2, clarifying Eq. 12 via Poisson–Stein) and aligns terminology (“likelihood-consistent objective”).
- Empirical substance: Broad synthetic suite covering heavy tails/zero inflation where ItDPDM reliably improves NLL/WD; real-data evidence on music/images; denoising robustness across extreme SNR; ablations (SNR schedules, DDPM+PRL) showing PRL helps beyond the Poisson channel. Added 2D/dependent toy studies and clarified per-pixel image modeling.
- Significance: Provides a native treatment for discrete, count-valued modalities (music, photon-limited imaging, event data, genomics counts), an area underserved by Gaussian diffusions; the method is simple to sample and compatible with standard backbones.

Weaknesses (camera-ready suggestions):
- Report seeds/SDs consistently and release code/checkpoints/configs.
- Keep “exact likelihood” toned down; include the promised finite-sample bias/variance discussion.
- Expand multivariate/correlated evaluations and add clearer qualitative comparisons (with ground truth) and a short note on OOD robustness scope.

The paper offers a well-motivated, theoretically grounded alternative for discrete generative modeling with promising empirical evidence and clear potential impact.